# Equivariant Machine Learning on Graphs with Nonlinear Spectral Filters

**Ya-Wei Eileen Lin**[†]    **Ronen Talmon**[†]    **Ron Levie**[‡]

[†]Viterbi Faculty of Electrical and Computer Engineering, Technion
[‡]Faculty of Mathematics, Technion

## Abstract

Equivariant machine learning is an approach for designing deep learning models that respect the symmetries of the problem, with the aim of reducing model complexity and improving generalization. In this paper, we focus on an extension of shift equivariance, which is the basis of convolution networks on images, to general graphs. Unlike images, graphs do not have a natural notion of domain translation. Therefore, we consider the graph functional shifts as the symmetry group: the unitary operators that commute with the graph shift operator. Notably, such symmetries operate in the signal space rather than directly in the spatial space. We remark that each linear filter layer of a standard spectral graph neural network (GNN) commutes with graph functional shifts, but the activation function breaks this symmetry. Instead, we propose nonlinear spectral filters (NLSFs) that are fully equivariant to graph functional shifts and show that they have universal approximation properties. The proposed NLSFs are based on a new form of spectral domain that is transferable between graphs. We demonstrate the superior performance of NLSFs over existing spectral GNNs in node and graph classification benchmarks.

## 1 Introduction

In many fields, such as chemistry [37], biology [36, 3], social science [9], and computer graphics [97], data can be described by graphs. In recent years, there has been a tremendous interest in the development of machine learning models for graph-structured data [98, 13]. This young field, often called *graph machine learning (graph-ML)* or *graph representation learning*, has made significant contributions to the applied sciences, e.g., in protein folding, molecular design, and drug discovery [45, 2, 70, 89], and has impacted the industry with applications in social media, recommendation systems, traffic prediction, computer graphics, and natural language processing, among others.

Geometric Deep Learning (GDL) [13] is a design philosophy for machine learning models where the model is constructed to inherently respect symmetries present in the data, aiming to reduce model complexity and enhance generalization. By incorporating knowledge of these symmetries into the model, it avoids the need to expend parameters and data to learn them. This inherent respect for symmetries is automatically generalized to test data, thereby improving generalization [6, 76, 25]. For instance, convolutional neural networks (CNNs) respect the translation symmetries of 2D images, with weight sharing due to these symmetries contributing significantly to their success [54]. Respecting node re-indexing in a scalable manner revolutionized machine learning on graphs [13, 12] and has placed GNNs as a main general-purpose tool for processing graph-structured data. Moreover, within GNNs, respecting the 3D Euclidean symmetries of the laws of physics (rotations, reflections, and translations) led to state-of-the-art performance in molecule processing [24].

**Our Contribution.** In this paper, we focus on GDL for graph-ML. We consider extensions of shift symmetries from images to general graphs. Since graphs do not have a natural notion of domain

38th Conference on Neural Information Processing Systems (NeurIPS 2024).

translation, as opposed to images, we propose considering functional translations instead. We model the group of translations on graphs as the group of all unitary operators on signals that commute with the graph shift operator. Such unitary operators are called *graph functional shifts*. Note that each linear filter layer of a standard spectral GNN commutes with graph functional shifts, but the activation function breaks this symmetry. Instead, we propose *non-linear spectral filters (NLSFs)* that are fully equivariant to graph functional shifts and have universal approximation properties.

In Sec. 3, we introduce our NLSFs based on new notions of *analysis* and *synthesis* that map signals between their node-space representations and spectral representations. Our transforms are related to standard graph Fourier and inverse Fourier transforms but differ from them in one important aspect. One key property of our analysis transform is that it is independent of a specific choice of Laplacian eigenvectors. Hence, our spectral representations are transferable between graphs. In comparison, standard graph Fourier transforms are based on an arbitrary choice of eigenvectors, and therefore, the standard frequency domain is not transferable. To achieve transferability, standard graph Fourier methods resort to linear filter operations based on functional calculus. Since we do not have this limitation, we can operate on the frequency coefficients with arbitrary nonlinear functions such as multilayer perceptrons. In Sec. 4, we present theoretical results of our NLSFs, including the universal approximation and expressivity properties. In Sec. 5, we demonstrate the efficacy of our NLSFs in node and graph classification benchmarks, where our method outperforms existing spectral GNNs.

## 2 Background

**Notation.** For $N \in \mathbb{N}$, we denote $[N] = \{1, \ldots, N\}$. We denote matrices by boldface uppercase letter $\mathbf{B}$, vectors (assumed to be columns) by lowercase boldface $\mathbf{b}$, and the entries of matrices and vectors are denoted with the same letter in lower case, e.g., $\mathbf{B} = (b_{i,j})_{i,j \in [N]}$. Let $G = ([N], \mathcal{E}, \mathbf{A}, \mathbf{X})$ be an undirected graph with a node set $[N]$, an edge set $\mathcal{E} \subset [N] \times [N]$, an adjacency matrix $\mathbf{A} \in \mathbb{R}^{N \times N}$ representing the edge weights, and a node feature matrix (also called a signal) $\mathbf{X} \in \mathbb{R}^{N \times d}$ containing $d$-dimensional node attributes. Let $\mathbf{D}$ be the diagonal degree matrix of $G$, where the diagonal element $d_{i,i}$ is the degree of node $i$. Denote by $\mathbf{\Delta}$ any normal graph shift operator (GSO). For example, $\mathbf{\Delta}$ could be the combinatorial graph Laplacian or the normalized graph Laplacian given by $\mathbf{L} = \mathbf{D} - \mathbf{A}$ and $\mathbf{N} = \mathbf{D}^{-\frac{1}{2}} \mathbf{L} \mathbf{D}^{-\frac{1}{2}}$, respectively. Let $\mathbf{\Delta} = \mathbf{V} \mathbf{\Lambda} \mathbf{V}^\top$ be the eigendecomposition of $\mathbf{\Delta}$, where $\mathbf{V}$ is the eigenvector matrix and $\mathbf{\Lambda} = \mathrm{diag}(\lambda_1, \ldots, \lambda_N)$ is the diagonal matrix with eigenvalues $(\lambda_i)_{i=1}^N$ ordered by $|\lambda_1| \leq \ldots \leq |\lambda_N|$. An eigenspace is the span of all eigenvectors corresponding to the same eigenvalue. Let $\mathbf{P}_i = \mathbf{P}_{\mathbf{\Delta};i}$ denote the projection upon the $i$-th eigenspace of $\mathbf{\Delta}$ in increasing order of $|\lambda|$. We denote the Euclidean norm by $\|\mathbf{X}\|_2$. We define the *channel-wise signal norm* $\|\mathbf{X}\|_{\mathrm{sig}}$ of a feature matrix $\mathbf{X} \in \mathbb{R}^{N \times d}$ as the vector $\|\mathbf{X}\|_{\mathrm{sig}} = (\|\mathbf{X}_{:,j}\|_2)_{j=1}^d \in \mathbb{R}^d$, where $\mathbf{X}_{:,j}$ is the $j$-th column on $\mathbf{X}$ and $0 \leq a \leq 1$. We abbreviate multilayer perceptrons by MLP.

### 2.1 Linear Graph Signal Processing

Spectral GNNs define convolution operators on graphs via the spectral domain. Given a self-adjoint *graph shift operator (GSO)* $\mathbf{\Delta}$, e.g., a graph Laplacian, the Fourier modes of the graph are defined to be the eigenvectors $\{\mathbf{v}_i\}_{i=1}^N$ of $\mathbf{\Delta}$ and the eigenvalues $\{\lambda_i\}_{i=1}^N$ are the frequencies. A spectral filter is defined to directly satisfy the "convolution theorem" [11] for graphs. Namely, given a signal $\mathbf{X} \in \mathbb{R}^{N \times d}$ and a function $\mathbf{Q} : \mathbb{R} \to \mathbb{R}^{d' \times d}$, the operator $\mathbf{Q}(\mathbf{\Delta}) : \mathbb{R}^{N \times d} \to \mathbb{R}^{N \times d'}$ defined by

$$\mathbf{Q}(\mathbf{\Delta})\mathbf{X} := \sum_{i=1}^N \mathbf{v}_i \mathbf{v}_i^\top \mathbf{X} \mathbf{Q}(\lambda_i)^\top, \tag{1}$$

is called a filter. Here, $d'$ is the number of output channels. Spectral GNNs, e.g., [21, 49, 61, 5], are graph convolutional networks where convolutions are via Eq. (1), with a trainable function $\mathbf{Q}$ at each layer, and a nonlinear activation function.

### 2.2 Equivariant GNNs

Equivariance describes the ability of functions $f$ to respect symmetries. It is expressed as $f(H_\kappa x) = H_\kappa f(x)$, where $\mathcal{K} \ni \kappa \mapsto H_\kappa$ is an action of a symmetry group $\mathcal{K}$ on the domain of $f$. GNNs [83, 14], including spectral GNNs [22, 49] and subgraph GNNs [31, 4], are inherently permutation equivariant

w.r.t. the ordering of nodes. This means that the network's operations are unaffected by the specific arrangement of nodes, a property stemming from passive symmetries [95] where transformations are applied to both the graph signals and the graph domain. This permutation equivariance is often compared to the translation equivariance in CNNs [56, 57], which involves active symmetries [44]. The key difference between the two symmetries lies in the domain: while CNNs operate on a fixed domain with signals transforming within it, graphs lack a natural notion of domain translation. To address this, we consider graph functional shifts as the symmetry group, defined by unitary operators that commute with the graph shift operator. This perspective allows for our NLSFs to be interpreted as an extension of active symmetry within the graph context, bridging the gap between the passive and active symmetries inherent to GNNs and CNNs, respectively.

## 3 Nonlinear Spectral Graph Filters

In this section, we present new concepts of analysis and synthesis under which the spectral domain is transferrable between graphs. Following these concepts, we introduce new GNNs that are equivariant to *functional symmetries* – symmetries of the Hilbert space of signals rather than symmetries in the domain of definition of the signal [64].

### 3.1 Translation Equivariance of CNNs and GNNs

For motivation, we start with the grid graph $R$ with node set $[M]^2$ and circular adjacency $\mathbf{B}$, we define the translation operator $\mathbf{T}_{m,n}$ by $[m, n]$ as

$$\mathbf{T}_{m,n} \in \mathbb{R}^{M^2 \times M^2}; \quad (\mathbf{T}_{m,n}\mathbf{x})_{i,j} = \mathbf{x}_{l,k} \quad \text{where} \quad l = (i-m) \bmod M \quad \text{and} \quad k = (j-n) \bmod M.$$

Note that any $\mathbf{T}_{m,n}$ is a unitary operator that commutes with the grid Laplacian $\mathbf{\Delta}_R$, i.e., $\mathbf{T}_{m,n}\mathbf{\Delta}_R = \mathbf{\Delta}_R \mathbf{T}_{m,n}$, and therefore it belongs to the group of all unitary operators $\mathcal{U}_R$ that commute with the grid Laplacian $\mathbf{\Delta}_R$. In fact, the space of isotropic convolution operators (with $90^o$ rotation and reflection symmetric filters) can be seen as the space of all normal operators[1] that commute with unitary operators from $\mathcal{U}_R$ [18]. Applying a non-linearity after the convolution retains this equivariance, and hence, we can build multi-layer CNNs that commute with $\mathcal{U}_R$. By the universal approximation theorem [19, 34, 58], this allows us to approximate any continuous function that commutes with $\mathcal{U}_R$.

Note that such translation equivariance cannot be extended to general graphs. Achieving equivariance to graph functional shifts through linear spectral convolutional layers $\mathbf{Q}(\mathbf{\Delta})$ is straightforward, since these layers commute with the space of all unitary operators $\mathcal{U}_{\mathbf{\Delta}}$ that commute with $\mathbf{\Delta}$. However, introducing non-linearity $\rho(\mathbf{Q}(\mathbf{\Delta})\mathbf{X})$ breaks the symmetry. That is, there exists $\mathbf{U} \in \mathcal{U}_{\mathbf{\Delta}}$ such that

$$\rho\left(\mathbf{Q}(\mathbf{\Delta})\mathbf{U}\mathbf{X}\right) \neq \mathbf{U}\rho\left(\mathbf{Q}(\mathbf{\Delta})\mathbf{X}\right),$$

where $\rho$ is any non-linear activation function, e.g., ReLU, Sigmoid, etc.

This means that multi-layer spectral GNNs do not commute with $\mathcal{U}_G$, and are hence not appropriate as approximators of general continuous functions that commute with $\mathcal{U}_G$ (see App. A for an example illustrating how non-linear activation functions break the functional symmetry). Instead, we propose in this paper a multi-layer GNN that is fully equivariant to $\mathcal{U}_G$, which we show to be universal: it can approximate any continuous graph-signal function (w.r.t. some metric) commuting with $\mathcal{U}_G$.

### 3.2 Graph Functional Symmetries and Their Relaxations

We define the symmetry group of graph functional shifts as follows.

**Definition 1** (Graph Functional Shifts). *The space of* graph functional shifts *is the unitary subgroup* $\mathcal{U}_{\mathbf{\Delta}}$*, where a unitary matrix* $\mathbf{U}$ *is in* $\mathcal{U}_{\mathbf{\Delta}}$ *iff it commutes with the GSO* $\mathbf{\Delta}$*, namely,* $\mathbf{U}\mathbf{\Delta} = \mathbf{\Delta}\mathbf{U}$*.*

It is important to note that functional shifts, in general, are not induced from node permutations. Instead, functional shifts are related to the notion of functional maps [73] used in shape correspondence and are general unitary operators that are not permutation matrices in general. The value of the functionally translated signal at a given node can be a *mixture* of the content of the original signal at many different nodes. For example, the functional shift can be a combination of shifts of different frequencies at different speeds. See App. B for illustrations and examples of functional translations.

---

[1]The operator $\mathbf{B}$ is normal iff $\mathbf{B}^*\mathbf{B} = \mathbf{B}\mathbf{B}^*$. Equivalently, iff $\mathbf{B}$ has an orthogonal eigendecomposition.

A fundamental challenge with the symmetry group in Def. 1 is its lack of transferability between different graphs. Hence, we propose to relax this symmetry group. Let $g_1, \ldots, g_S : \mathbb{R} \to \mathbb{R}$ be the indicator functions of the intervals $\{[l_s, l_{s+1}]\}_{s=1}^S$, which constitute a partition of the frequency band $[l_1, l_S] \subset \mathbb{R}$. The operators $g_j(\boldsymbol{\Delta})$, interpreted via functional calculus Eq. (1), are projections of the signal space upon band-limited signals. Namely, $g_j(\boldsymbol{\Delta}) = \sum_{i:\lambda_i \in [l_j, l_{j+1}]} \mathbf{v}_i \mathbf{v}_i^\top$. In our work, we consider filters $g_j$ that are supported on the dyadic sub-bands $\left[\lambda_N r^{S-j+1}, \lambda_N r^{S-j}\right]$, where $0 < r < 1$ is the decay rate. See Fig. 5 in App. F for an illustrated example. Note that for $j = 1$, the sub-band falls in $[0, \lambda_N r^{S-1}]$. The total band $[0, l_S]$ is $[0, \lambda_N]$.

**Definition 2** (Relaxed Functional Shifts). *The space of* relaxed functional shifts *with respect to the filter bank* $\{g_j\}_{j=1}^K$ *(of indicators) is the unitary subgroup* $\mathcal{U}_{\boldsymbol{\Delta}}^g$, *where a unitary matrix* $\mathbf{U}$ *is in* $\mathcal{U}_{\boldsymbol{\Delta}}^g$ *iff it commutes with* $g_j(\boldsymbol{\Delta})$ *for all* $j$, *namely,* $\mathbf{U}g_j(\boldsymbol{\Delta}) = g_j(\boldsymbol{\Delta})\mathbf{U}$.

Similarly, we can relax functional shifts by restricting to the leading eigenspaces.

**Definition 3** (Leading Functional Shifts). *The space of* leading functional shifts *is the unitary subgroup* $\mathcal{U}_{\boldsymbol{\Delta}}^J$, *where a unitary* $\mathbf{U}$ *is in* $\mathcal{U}_{\boldsymbol{\Delta}}^J$ *iff it commutes with the eigenspace projections* $\{\mathbf{P}_j\}_{j=1}^J$.

### 3.3 Analysis and Synthesis

We use the terminology of analysis and synthesis, as in signal processing [68], to describe transformations of signals between their graph and spectral representations. Here, we consider two settings: the eigenspace projections case and the filter bank (of indicators) case. The definition of the frequency domain depends on a given signal $\mathbf{X} \in \mathbb{R}^{N \times d}$, where its projections to the eigenspaces of $\boldsymbol{\Delta}$ are taken as the Fourier modes. Spectral coefficients are modeled as matrices $\mathbf{R}$ that mix the Fourier modes, allowing to synthesize signals of general dimensions.

**Spectral Index Case.** We first define *analysis and synthesis using the spectral index* up to frequency $J$. Let $\mathbf{P}_{J+1} = \mathbf{I} - \sum_{j=1}^J \mathbf{P}_j$ be the orthogonal complement to the first $J$ eigenprojections. Let $\mathbf{X} \in \mathbb{R}^{N \times d}$ be the graph signal and $\mathbf{R} \in \mathbb{R}^{(J+1)d \times (J+1)\widetilde{d}}$ the spectral coefficients to be synthesized, where $\widetilde{d}$ represents number of output channels. The analysis and synthesis are defined respectively by

$$\mathcal{A}^{\text{ind}}(\boldsymbol{\Delta}, \mathbf{X}) = \left(\|\mathbf{P}_i \mathbf{X}\|_{\text{sig}}\right)_{i=1}^{J+1} \in \mathbb{R}^{(J+1)d} \text{ and } \mathcal{S}^{\text{ind}}(\mathbf{R}; \boldsymbol{\Delta}, \mathbf{X}) = \left[\frac{\mathbf{P}_j \mathbf{X}}{\|\mathbf{P}_j \mathbf{X}\|_{\text{sig}}^a + e}\right]_{j=1}^{J+1} \mathbf{R}, \quad (2)$$

where $0 \leq a \leq 1$ and $0 < e \ll 1$ are parameters that promote stability, and the power $\|\mathbf{P}_j \mathbf{X}\|_{\text{sig}}^a$ as well as the division in Eq. (2) are element-wise operations on each entry. Here, $\left[\mathbf{P}_j \mathbf{X} / (\|\mathbf{P}_j \mathbf{X}\|_{\text{sig}}^a + e)\right]_{j=1}^{J+1} \in \mathbb{R}^{N \times (J+1)d}$ denotes concatenation. We remark that the orthogonal complement in the $(J+1)$-th filter alleviates the loss of information due to projecting to the low-frequency bands, and therefore, the full spectral range of the signal can be captured. This is particularly important for heterophilic graphs, which rely on high-frequency components for accurate label representation. The term *index* stems from the fact that eigenvalues are treated according to their index when defining the projections $\mathbf{P}_j$. Note that the synthesis here differs from classic signal processing as it depends on a given signal on the graph. When treating $\boldsymbol{\Delta}$ and $\mathbf{X}$ as fixed, this synthesis operation is denoted by $\mathcal{S}_{\boldsymbol{\Delta}, \mathbf{X}}^{\text{ind}}(\mathbf{R}) := \mathcal{S}^{\text{ind}}(\mathbf{R}; \boldsymbol{\Delta}, \mathbf{X})$. We similarly denote $\mathcal{A}_{\boldsymbol{\Delta}}^{\text{ind}}(\mathbf{X}) := \mathcal{A}^{\text{ind}}(\boldsymbol{\Delta}, \mathbf{X})$.

**Filter Bank Case.** Similarly, we define the *analysis and synthesis in the filter bank* up to band $g_K$ as follows. Let $g_{K+1}(\boldsymbol{\Delta}) = \mathbf{I} - \sum_{j=1}^K g_j(\boldsymbol{\Delta})$ denote the orthogonal complement to the first $K$ bands. Let $\mathbf{X} \in \mathbb{R}^{N \times d}$ be the graph signal, and let $\mathbf{R} \in \mathbb{R}^{(K+1)d \times (K+1)\widetilde{d}}$ represent the spectral coefficients to be synthesized, where $\widetilde{d}$ refers to the general dimension. The analysis and synthesis in the filter bank case are defined by

$$\mathcal{A}^{\text{val}}(\boldsymbol{\Delta}, \mathbf{X}) = \left(\|g_i(\boldsymbol{\Delta})\mathbf{X}\|_{\text{sig}}\right)_{i=1}^{K+1} \in \mathbb{R}^{(K+1)d} \text{ and } \mathcal{S}^{\text{val}}(\mathbf{R}; \boldsymbol{\Delta}, \mathbf{X}) = \left[\frac{g_j(\boldsymbol{\Delta})\mathbf{X}}{\|g_j(\boldsymbol{\Delta})\mathbf{X}\|_{\text{sig}}^a + e}\right]_{j=1}^{K+1} \mathbf{R},$$
$$(3)$$

respectively, where $a, e$ are as before. Here, $\left[g_j(\boldsymbol{\Delta})\mathbf{X} / (\|g_j(\boldsymbol{\Delta})\mathbf{X}\|_{\text{sig}}^a + e)\right]_{j=1}^{K+1} \in \mathbb{R}^{N \times (K+1)d}$. The term *value* refers to how eigenvalues are used based on their magnitude when defining the projections $g_j(\boldsymbol{\Delta})$. As before, we denote $\mathcal{S}_{\boldsymbol{\Delta}, \mathbf{X}}^{\text{val}}$ and $\mathcal{A}_{\boldsymbol{\Delta}}^{\text{val}}$.

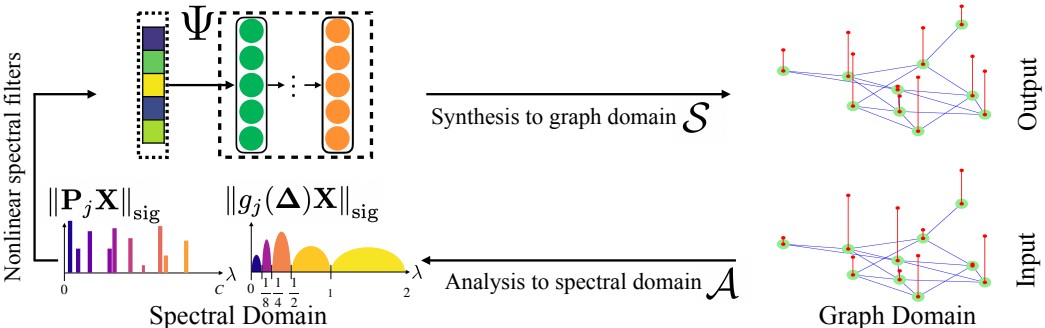

Figure 1: Illustration of nonlinear spectral filters for equivariant machine learning on graphs. Given a graph $G$, the node features $\mathbf{X}$ are projected onto eigenspaces (analysis $\mathcal{A}$). The function $\Psi$ map a sequence of frequency coefficients to a sequence of frequency coefficients. The coefficients are synthesized to the graph domain using the using $\mathcal{S}$.

In App. C.1, we present a special case of diagonal synthesis where $\widetilde{d} = d$. In App. D.3, we show that the diagonal synthesis is stably invertible.

## 3.4 Definitions of Nonlinear Spectral Filters

We introduce three novel types of *non-linear spectral filters (NLSF)*: Node-level NLSFs, Graph-level NLSFs, and Pooling-NLSFs. Fig. 1 illustrates our NLSFs for equivariant machine learning on graphs.

**Node-level NLSFs.** To be able to transfer NLSFs between different graphs and signals, one key property of NLSF is that they do not depend on the specific basis chosen in each eigenspace. This independence is facilitated by the synthesis process, which relies on the input signal $\mathbf{X}$. Following the spectral index and filter bank cases in Sec. 3.3, we define the Index NLSFs and Value NLSFs by

$$\Theta_{\text{ind}}(\boldsymbol{\Delta}, \mathbf{X}) = \mathcal{S}_{\boldsymbol{\Delta},\mathbf{X}}^{(\text{ind})}(\Psi_{\text{ind}}(\mathcal{A}_{\boldsymbol{\Delta}}^{(\text{ind})}(\mathbf{X}))) = \left[\frac{\mathbf{P}_j\mathbf{X}}{\|\mathbf{P}_j\mathbf{X}\|_{\text{sig}}^a + e}\right]_{j=1}^{J+1} \left[\Psi_{\text{ind}}\left(\|\mathbf{P}_i\mathbf{X}\|_{\text{sig}}\right)\right]_{i=1}^{J+1}, \qquad (4)$$

$$\Theta_{\text{val}}(\boldsymbol{\Delta}, \mathbf{X}) = \mathcal{S}_{\boldsymbol{\Delta},\mathbf{X}}^{(\text{val})}(\Psi_{\text{val}}(\mathcal{A}_{\boldsymbol{\Delta}}^{(\text{val})}(\mathbf{X}))) = \left[\frac{g_j(\boldsymbol{\Delta})\mathbf{X}}{\|g_j(\boldsymbol{\Delta})\mathbf{X}\|_{\text{sig}}^a + e}\right]_{j=1}^{K+1} \left[\Psi_{\text{val}}\left(\|g_i(\boldsymbol{\Delta})\mathbf{X}\|_{\text{sig}}\right)\right]_{i=1}^{K+1},$$

$$\tag{5}$$

where $\Psi_{\text{ind}} : \mathbb{R}^{(J+1)d} \to \mathbb{R}^{(J+1)d \times (J+1)\widetilde{d}}$ and $\Psi_{\text{val}} : \mathbb{R}^{(K+1)d} \to \mathbb{R}^{(K+1)d \times (K+1)\widetilde{d}}$ are called *nonlinear frequency responses*, and $\widetilde{d}$ is the output dimension. To adjust the feature output dimension, we apply an MLP with shared weights to all nodes after the NLSF. In the case when $\widetilde{d} = d$ and the filters operator diagonally (i.e., the product and division are element-wise in synthesis), we refer to it as diag-NLSF. See App. C for more details.

**Graph-level NLSFs.** We first introduce the Graph-level NLSFs that are fully spectral, where the NLSFs map a sequence of frequency coefficients to an output vector. Specifically, the Index-based and Value-based Graph-level NLSFs are given by

$$\Phi_{\text{ind}}(\boldsymbol{\Delta}, \mathbf{X}) = \widehat{\Psi}_{\text{ind}}\left(\|\mathbf{P}_i\mathbf{X}\|_{\text{sig}}\right) \quad \text{and} \quad \Phi_{\text{val}}(\boldsymbol{\Delta}, \mathbf{X}) = \widehat{\Psi}_{\text{val}}\left(\|g_i(\boldsymbol{\Delta})\mathbf{X}\|_{\text{sig}}\right), \qquad (6)$$

where $\widehat{\Psi}_{\text{ind}} : \mathbb{R}^{(J+1)d} \to \mathbb{R}^{d'}$, $\widehat{\Psi}_{\text{val}} : \mathbb{R}^{(K+1)d} \to \mathbb{R}^{d'}$, and $d'$ is the output dimension.

**Pooling-NLSFs.** We introduce another type of graph-level NLSFs by first representing each graph in a Node-level NLSFs as in Eq. (4) and Eq. (5). The final graph representation is obtained by applying a nonlinear activation function followed by a readout function to these node-level representations. We consider four commonly used pooling methods, including mean, sum, max, and $L_p$-norm pooling, as the readout function for each graph. We apply an MLP after readout function to obtain a $d'$-dimensional graph-level representation. We term these graph-level NLSFs as *Pooling-NLSFs*.

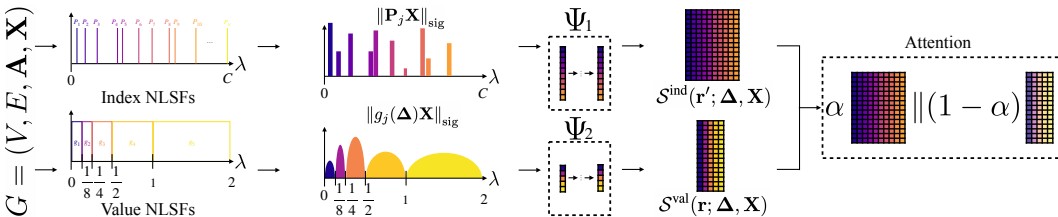

Figure 2: Illustration of Laplacian attention NLSFs. An attention mechanism is applied to both Index NLSFs and Value NLSFs, enabling the adaptive selection of the most appropriate parameterization.

## 3.5 Laplacian Attention NLSFs

To understand which Laplacian and parameterization (index v/s value) of the NLSF are preferable in different settings, we follow the random geometric graph analysis outlined in [74]. Specifically, we consider a setting where random geometric graphs are sampled from a metric-probability space $\mathcal{S}$. In such a case, the graph Laplacian approximates continuous Laplacians on the metric spaces under some conditions. We aim for our NLSFs to produce approximately the same outcome for any two graphs sampled from the same underlying metric space $\mathcal{S}$, ensuring that the NLSF is *transferable*. In App. D.5, we show that if the nodes of the graph are sampled uniformly from $\mathcal{S}$, then using the graph Laplacian $\mathbf{L}$ in Index NLSFs yields a transferable method. Conversely, if the nodes of the graph are sampled non-uniformly, and any two balls of the same radius in $\mathcal{S}$ have the same volume, then utilizing the normalized graph Laplacian $\mathbf{N}$ in Value NLSFs is a transferable method. Given that graphs may fall between these two boundary cases, we present an architecture that chooses between the Index NLSFs with respect to $\mathbf{L}$ and Value NLSFs with respect to $\mathbf{N}$, as illustrated in Fig. 2. While the above theoretical setting may not be appropriate as a model for every real-life graph dataset, it suggests that index NLSF may be more appropriate with $\mathbf{L}$, value NLSFs with $\mathbf{N}$, and different graphs are more appropriately analyzed by different balances between these two cases.

In the Laplacian attention architecture, a soft attention mechanism is employed to dynamically choose between the two parameterizations, given by

$$\mathtt{att}\left(\Theta_{\text{ind}}(\mathbf{L}, \mathbf{X}), \Theta_{\text{val}}(\mathbf{N}, \mathbf{X})\right) = \alpha\Theta_{\text{ind}}(\mathbf{L}, \mathbf{x}) \| (1 - \alpha)\Theta_{\text{val}}(\mathbf{N}, \mathbf{X}),$$

where $0 \leq \alpha \leq 1$ is obtained using a softmax function to normalize the scores into attention weights, balancing each NLSFs' contribution.

## 4 Theoretical Properties of Nonlinear Spectral Filters

We present the desired theoretical properties of our NLSFs at the node-level and graph-level.

### 4.1 Complexity of NLSFs

NLSFs are implemented by computing the eigenvectors of the GSO. Most existing spectral GNNs avoid direct eigendecomposition due to its perceived inefficiency. Instead, they use filters implemented by applying polynomials [49, 22] or rational functions [61, 5] to the GSO in the spatial domain. However, power iteration-based eigendecomposition algorithms, e.g., variants of the Lanczos method, can be highly efficient [80, 55]. For matrices with $E$ non-zero entries, the computational complexity of one iteration for finding $J$ eigenvectors corresponding to the smallest or largest eigenvalues (called *leading eigenvectors*) is $O(JE)$. In practice, these eigendecomposition algorithms converge quickly due to their super-exponential convergence rate, often requiring only a few iterations, which makes them as efficient as message passing networks of signals with $\sqrt{J}$ channels.

This makes NLSFs applicable to node-level tasks on large sparse graphs, as demonstrated empirically in App. F.5, since they rely solely on the leading eigenvectors. In Sec. 4.4, we show that using the leading eigenvectors can approximate GSOs well in the context of learning on graphs. Note that we can precompute the spectral projections of the signal before training. For node-level tasks, such as semi-supervised node classification, the leading eigenvectors only need to be pre-computed once, with a complexity of $O(JE)$. This During the *learning phase*, each step of the architecture search and hyperparameter optimization takes $O(NJd)$ complexity for analysis and synthesis, and $O(J^2 d^2)$

for the MLP in the spectral domain, which is faster than the complexity $O(Ed^2)$ of message passing or standard spectral methods if $NJ < Ed$. Empirical studies on runtime analysis are in App. F.

For dense matrices, the computational complexity of a full eigendecomposition is $O(N^b)$ per iteration, where $N^b$ is the complexity of matrix multiplication. This is practical for graph-level tasks on relatively small and dense graphs, which is typical for many graph classification datasets. In these cases, the eigendecomposition of all graphs in the dataset can be performed as a pre-computation step, significantly reducing the complexity during the learning phase.

## 4.2   Equivariance of Node-level NLSFs

We demonstrate the node-level equivariance of our NLSFs, ensuring that our method respects the functional shift symmetries. The proof is given in App. D.1.

**Proposition 1.** *Index NLSFs in Eq.* (4) *are equivariant to the graph functional shifts* $\mathcal{U}_{\boldsymbol{\Delta}}$, *and Value NLSFs in Eq.* (5) *are equivariant to the relaxed graph functional shifts* $\mathcal{U}_{\boldsymbol{\Delta}}^g$.

## 4.3   Universal Approximation and Expressivity

In this subsection, we discuss the approximation power of NLSFs.

**Node-Level Universal Approximation.** We begin with a setting where a graph is given as a fixed domain, and the data distribution consists of multiple signals defined on this graph. An example of this setup is a spatiotemporal graph [17], e.g., traffic networks, where a fixed sensor system defines a graph and the different signals represent the sensor readings collected at different times.

In App. D.2.1, we prove the following lemma, which shows that linear NLSFs exhaust the space of linear operators that commute with graph functional shifts.

**Lemma 1.** *A linear operator* $\mathbb{R}^{N \times d} \to \mathbb{R}^{N \times d}$ *commutes with* $\mathcal{U}_{\boldsymbol{\Delta}}^J$ *(resp.* $\mathcal{U}_{\boldsymbol{\Delta}}^g$*) iff it is a NLSF based on a linear function* $\Psi$ *in Eq.* (4) *(resp. Eq.* (5)*).*

Lemma 1 shows a close relationship between functions that commute with functional shifts and those defined in the spectral domain. This motivates the following construction of a pseudo-metric on $\mathbb{R}^{N \times d}$. In the case of relaxed functional shifts, we define the standard Euclidean metric $\mathrm{dist}_{\mathrm{E}}$ in the spectral domain $\mathbb{R}^{(K+1) \times d}$. We pull back the Euclidean metric to the spatial domain to define a signal pseudo-metric. Namely, for two signals $\mathbf{X}$ and $\mathbf{X}'$, their distance is defined by

$$\mathrm{dist}_{\boldsymbol{\Delta}}\big(\mathbf{X}, \mathbf{X}'\big) := \mathrm{dist}_{\mathrm{E}}\big(\mathcal{A}(\boldsymbol{\Delta}, \mathbf{X}), \mathcal{A}(\boldsymbol{\Delta}', \mathbf{X}')\big).$$

This pseudo metric can be made into a metric by considering each equivalence class of signals with zero distance as a single point in the space. As MLPs $\Psi$ can approximate any continuous function $\mathbb{R}^{(K+1) \times d} \to \mathbb{R}^{(K+1) \times d}$ (the universal approximation theorem [19, 34, 58]), node-level NLSFs can approximate any continuous function that maps (equivalence classes of) signals to (equivalence classes of) signals. For details, see App. D.2.2. A similar analysis applies to hard functional shifts.

**Graph-Level Universal Approximation.** The above analysis also motivates the construction of a graph-signal metric for graph-level tasks. For graphs with $d$-channel signals, we consider again the standard Euclidean metric $\mathrm{dist}_{\mathrm{E}}$ in the spectral domain $\mathbb{R}^{(K+1) \times d}$. We define the distance between any two graphs with GSOs and signals $(\boldsymbol{\Delta}, \mathbf{X})$ and $(\boldsymbol{\Delta}', \mathbf{X}')$ to be

$$\mathrm{dist}\big((\boldsymbol{\Delta}, \mathbf{X}), (\boldsymbol{\Delta}', \mathbf{X}')\big) := \mathrm{dist}_{\mathrm{E}}\big(\mathcal{A}(\boldsymbol{\Delta}, \mathbf{X}), \mathcal{A}(\boldsymbol{\Delta}', \mathbf{X}')\big).$$

This definition can be extended into a metric by considering the space $\mathcal{G}$ of equivalence classes of graph-signals with distance 0. As before, this distance inherits the universal approximation properties of standard MLPs. Namely, any continuous function $\mathcal{G} \to \mathbb{R}^{d'}$ with respect to $\mathrm{dist}$ can be approximated by NLSFs based on MLPs. Additional details are in App. D.2.3.

**Graph-Level Expressivity of Pooling-NLSFs.** In App. D.2.4, we show that Pooling-NLSFs are more expressive than graph-level NLSF when any $L_p$ norm is used in Eq. (4) and Eq. (5) with $p \neq 2$, both in the definition of the NLSF and as the pooling method. Specifically, for every graph-level NLSF, there is a Pooling-NLSF that coincides with it. Additionally, there are graph signals $(\boldsymbol{\Delta}, \mathbf{X})$ and $(\boldsymbol{\Delta}', \mathbf{X}')$ for which a Pooling-NLSF can attain different values, whereas any graph-level NLSF must attain the same value. Hence, Pooling-NLSFs have improved discriminative power compared to graph-level NLSFs. Indeed, as shown in Tab. 3, Pooling-NLSFs outperform Graph-level NLSFs in

Table 1: Semi-supervised node classification accuracy.

|  | Cora | Citeseer | Pubmed | Chameleon | Squirrel | Actor |
|---|---|---|---|---|---|---|
| GCN | $81.92_{\pm 0.9}$ | $70.73_{\pm 1.1}$ | $80.14_{\pm 0.6}$ | $43.64_{\pm 1.9}$ | $33.26_{\pm 0.8}$ | $27.63_{\pm 1.7}$ |
| GAT | $83.64_{\pm 0.7}$ | $71.32_{\pm 1.3}$ | $79.45_{\pm 0.7}$ | $42.19_{\pm 1.3}$ | $28.21_{\pm 0.9}$ | $29.46_{\pm 0.9}$ |
| SAGE | $74.01_{\pm 2.1}$ | $66.40_{\pm 1.2}$ | $79.91_{\pm 0.9}$ | $41.92_{\pm 0.7}$ | $27.64_{\pm 2.1}$ | $30.85_{\pm 1.8}$ |
| ChebNet | $79.72_{\pm 1.1}$ | $70.48_{\pm 1.0}$ | $76.47_{\pm 1.5}$ | $44.95_{\pm 1.2}$ | $33.82_{\pm 0.8}$ | $27.42_{\pm 2.3}$ |
| ChebNetII | $83.95_{\pm 0.8}$ | $71.76_{\pm 1.2}$ | $81.38_{\pm 1.3}$ | $46.37_{\pm 3.1}$ | $34.40_{\pm 1.1}$ | $33.48_{\pm 1.2}$ |
| CayleyNet | $81.76_{\pm 1.9}$ | $68.32_{\pm 2.3}$ | $77.48_{\pm 2.1}$ | $38.29_{\pm 3.2}$ | $26.53_{\pm 3.3}$ | $30.62_{\pm 2.8}$ |
| APPNP | $83.19_{\pm 0.8}$ | $71.93_{\pm 0.8}$ | $\mathbf{82.69}_{\pm 1.4}$ | $37.43_{\pm 1.9}$ | $25.68_{\pm 1.3}$ | $\mathbf{35.98}_{\pm 1.3}$ |
| GPRGNN | $82.82_{\pm 1.3}$ | $70.28_{\pm 1.4}$ | $81.31_{\pm 2.6}$ | $39.27_{\pm 2.3}$ | $26.09_{\pm 1.3}$ | $31.47_{\pm 1.6}$ |
| ARMA | $81.64_{\pm 1.2}$ | $69.91_{\pm 1.6}$ | $79.24_{\pm 0.5}$ | $39.40_{\pm 1.8}$ | $27.42_{\pm 0.7}$ | $30.42_{\pm 2.6}$ |
| JacobiConv | $\underline{84.12}_{\pm 0.7}$ | $72.59_{\pm 1.4}$ | $82.05_{\pm 1.9}$ | $49.66_{\pm 1.6}$ | $33.65_{\pm 0.8}$ | $34.61_{\pm 0.7}$ |
| BernNet | $82.96_{\pm 1.1}$ | $71.25_{\pm 1.0}$ | $81.07_{\pm 1.6}$ | $42.65_{\pm 3.4}$ | $31.68_{\pm 1.5}$ | $33.92_{\pm 0.8}$ |
| Specformer | $82.27_{\pm 0.7}$ | $\underline{73.45}_{\pm 1.4}$ | $81.62_{\pm 1.0}$ | $\underline{49.79}_{\pm 1.2}$ | $\underline{38.24}_{\pm 0.9}$ | $34.12_{\pm 0.6}$ |
| OptBasisGNN | $81.97_{\pm 1.2}$ | $70.46_{\pm 1.6}$ | $80.38_{\pm 0.9}$ | $47.12_{\pm 2.4}$ | $37.66_{\pm 1.1}$ | $34.84_{\pm 1.3}$ |
| att-Node-level NLSFs | $\mathbf{85.37}_{\pm 1.8}$ | $\mathbf{75.41}_{\pm 0.8}$ | $\underline{82.22}_{\pm 1.2}$ | $\mathbf{50.58}_{\pm 1.3}$ | $\mathbf{38.39}_{\pm 0.9}$ | $\underline{35.13}_{\pm 1.0}$ |

practice, which can be attributed to their increased expressivity. We refer to App. D.2.5 for additional discussion on graph-level expressivity.

## 4.4 Uniform Approximation of GSOs by Their Leading Eigenvectors

Since NLSFs on large graphs are based on the leading eigenvectors of $\mathbf{\Delta}$, we justify its low-rank approximation in the following. While approximating matrices with low-rank matrices might lead to a high error in the spectral and Frobenius norms, we show that such an approximation entails a uniformly small error in the cut norm. We define and interpret the cut norm in App. D.4.1, and explain why it is a natural graph similarity measure for graph machine learning.

The following theorem is a corollary of the Constructive Weak Regularity Lemma presented in [29]. Its proof is presented in App. D.4.

**Theorem 1.** *Let $\mathbf{M}$ be a symmetric matrix with entries bounded by $|m_{i,j}| \leq \alpha$, and let $J \in \mathbb{N}$. Suppose $m$ is sampled uniformly from $[J]$, and let $R \geq 1$ s.t. $J/R \in \mathbb{N}$. Let $\phi_1, \ldots, \phi_m$ be the leading eigenvectors of $\mathbf{M}$, with eigenvalues $\mu_1, \ldots, \mu_m$ ordered by their magnitudes $|\mu_1| \geq \ldots \geq |\mu_m|$. Define $\mathbf{C} = \sum_{k=1}^{m} \mu_k \phi_k \phi_k^\top$. Then, with probability $1 - \frac{1}{R}$ (w.r.t. the choice of $m$),*

$$\|\mathbf{M} - \mathbf{C}\|_\square < \frac{3\alpha}{2}\sqrt{\frac{R}{J}}.$$

Note that the bound in Thm. 1 is uniformly small, independently of $\mathbf{M}$ and its dimension $N$. This theorem justifies using the leading eigenvectors when working with the adjacency matrix as the GSO. For a justification when working with other GSOs see App. D.4.

## 5 Experimental Results

We evaluate the NLSFs on node and graph classification tasks. Additional implementation details are in App. E, and additional experiments, including runtime analysis and ablation studies, are in App. F.

### 5.1 Semi-Supervised Node Classification

We first demonstrate the main advantage of the proposed Node-level NLSFs over existing GNNs with convolution design on semi-supervised node classification tasks. We test three citation networks [84, 100]: Cora, Citeseer, and Pubmed. In addition, we explore three heterophilic graphs: Chameleon, Squirrel, and Actor [79, 90]. For more comprehensive descriptions of these datasets, see App. E.

We compare the Node-level NLSFs using Laplacian attention with existing spectral GNNs for node-level predictions, including GCN [49], ChebNet [22], ChebNetII [41], CayleyNet [61], APPNP [50], GPRGNN [16], ARMA [5], JacobiConv [96], BernNet [42], Specformer [7], and OptBasisGNN [38]. We also consider GAT [93] and SAGE [39]. For datasets splitting on citation graphs (Cora, Citeseer, and Pubmed), we apply the standard splits following [100], using 20 nodes per class for training,

500 nodes for validation, and 1000 nodes for testing. For heterophilic graphs (Chameleon, Squirrel, and Actor), we use the sparse splitting as in [16, 41], allocating 2.5% of samples for training, 2.5% for validation, and 95% for testing. We measure the classification quality by computing the average classification accuracy with a 95% confidence interval over 10 random splits. We utilize the source code released by the authors for the baseline algorithms and optimize their hyperparameters using Optuna [1]. Each model's hyperparameters are fine-tuned to achieve the highest possible accuracy. Detailed hyperparameter settings are provided in App. E.

Tab. 1 presents the node classification accuracy of our NLSFs using Laplacian attention and the various competing baselines. We see that `att`-Node-level NLSFs outperform the competing models on the Cora, Citeseer, and Chameleon datasets. Notably, it shows remarkable performance on the densely connected Squirrel graph, outperforming the baselines by a large margin. This can be explained by the sparse version in Eq. (21) of Thm. 1, which shows that the denser the graphs is, the better its rank-$J$ approximation. For the Pubmed and Actor datasets, `att`-Node-level NLSFs yield the second-best results, which are comparable to the best results obtained by APPNP.

## 5.2 Node Classification on Filtered Chameleon and Squirrel Datasets in Dense Split Setting

Recently, the work in [77] identified the presence of many duplicate nodes across the train, validation, and test splits in the dense split setting of the Chameleon and Squirrel [75]. This results in train-test data leakage, causing GNNs to inadvertently fit the test splits during training, thereby making performance results on Chameleon and Squirrel less reliable. To further validate the performance of our Node-level NLSFs

Table 2: Node classification performance on original and filtered Chameleon and Squirrel datasets.

| | Chameleon | | Squirrel | |
|---|---|---|---|---|
| | Original | Filtered | Original | Filtered |
| ResNet+SGC | 49.93±2.3 | 41.01±4.5 | 34.36±1.2 | 38.36±2.0 |
| ResNet+adj | 71.07±2.2 | 38.67±3.9 | 65.46±1.6 | 38.37±2.0 |
| GCN | 50.18±3.3 | 40.89±4.1 | 39.06±1.5 | 39.47±1.5 |
| GPRGNN | 47.26±1.7 | 39.93±3.3 | 33.39±2.1 | 38.95±2.0 |
| FSGNN | 77.85±0.5 | 40.61±3.0 | **68.93**±1.7 | 35.92±1.3 |
| GloGNN | 70.04±2.1 | 25.90±3.6 | 61.21±2.0 | 35.11±1.2 |
| FAGCN | 64.23±2.0 | 41.90±2.7 | 47.63±1.9 | 41.08±2.3 |
| `att`-Node-level NLSFs | **79.84**±1.2 | **42.53**±1.5 | 68.17±1.9 | **42.66**±1.7 |

on these datasets in the dense split setting, we use both the original and filtered versions of Chameleon and Squirrel, which do not contain duplicate nodes, as suggested in [77]. We use the same random splits as in [77], dividing the datasets into 48% for training, 32% for validation, and 20% for testing.

Tab. 2 presents the classification performance comparison between the original and filtered Chameleon and Squirrel. The baseline results are taken from [77], and we include the following competitive models: ResNet+SGC [77], ResNet+adj [77], GCN [49], GPRGNN [16], FSGNN [69], GloGNN [62], and FAGCN [8]. The detailed comparisons are in App. F. We see in the table that the `att`-Node-level NLSFs consistently outperform the competing baselines on both the original and filtered datasets. `att`-Node-level NLSFs demonstrate less sensitivity to node duplicates and exhibit stronger generalization ability, further validating the reliability of the Chameleon and Squirrel datasets in the dense split setting. We note that compared to the dense split setting, the sparse split setting in Tab. 1 is more challenging, resulting in lower classification performance. A similar trend of significant performance difference between the two settings of Chameleon and Squirrel is also observed in [41].

## 5.3 Graph Classification

We further illustrate the power of NLSFs on eight graph classification benchmarks [47]. Specifically, we consider five bioinformatics datasets [10, 53, 86]: MUTAG, PTC, NCI1, ENZYMES, and PROTEINS, where MUTAG, PTC, and NCI1 are characterized by discrete node features, while ENZYMES and PROTEINS have continuous node features. Additionally, we examine three social network datasets: IMDB-B, IMDB-M, and COLLAB. The unattributed graphs are augmented by adding node degree features following [26]. For more details of these datasets, see App. E.

In graph classification tasks, a readout function is used to summarize node representations for each graph. The final graph-level representation is obtained by aggregating these node-level summaries and is then fed into an MLP with a (log)softmax layer to perform the graph classification task. We compare our NLSFs with two kernel-based approaches: GK [87] and WL [86], as well as nine GNNs: GCN [49], GAT [93], SAGE [39], ChebNet [22], ChebNetII [41], CayleyNet [61], APPNP [50], GPRGNN [16], and ARMA [5]. Additionally, we consider the hierarchical graph pooling model

Table 3: Graph classification accuracy.

| | MUTAG | PTC | ENZYMES | PROTEINS | NCI1 | IMDB-B | IMDB-M | COLLAB |
|---|---|---|---|---|---|---|---|---|
| GK | $76.38_{\pm2.9}$ | $52.13_{\pm1.2}$ | $30.07_{\pm6.1}$ | $72.33_{\pm4.5}$ | $62.62_{\pm2.2}$ | $67.63_{\pm1.0}$ | $44.19_{\pm0.9}$ | $72.32_{\pm3.7}$ |
| WL | $78.04_{\pm1.8}$ | $52.44_{\pm1.3}$ | $51.29_{\pm5.9}$ | $76.20_{\pm4.1}$ | $76.45_{\pm2.3}$ | $71.42_{\pm3.2}$ | $46.63_{\pm1.3}$ | $76.23_{\pm2.2}$ |
| GCN | $81.63_{\pm3.1}$ | $60.22_{\pm1.9}$ | $43.66_{\pm3.4}$ | $75.17_{\pm3.7}$ | $76.29_{\pm1.8}$ | $72.96_{\pm1.3}$ | $50.28_{\pm3.2}$ | $79.98_{\pm1.9}$ |
| GAT | $83.17_{\pm4.4}$ | $62.31_{\pm1.4}$ | $39.83_{\pm3.7}$ | $74.72_{\pm4.1}$ | $74.01_{\pm4.3}$ | $70.62_{\pm2.5}$ | $45.67_{\pm2.7}$ | $74.27_{\pm2.5}$ |
| SAGE | $75.18_{\pm4.7}$ | $61.33_{\pm1.1}$ | $37.99_{\pm3.7}$ | $74.01_{\pm4.3}$ | $74.90_{\pm1.7}$ | $68.38_{\pm2.6}$ | $46.94_{\pm2.9}$ | $73.61_{\pm2.1}$ |
| DiffPool | $82.40_{\pm1.4}$ | $56.43_{\pm2.9}$ | $60.13_{\pm3.2}$ | $79.47_{\pm3.1}$ | $77.18_{\pm0.7}$ | $71.09_{\pm1.6}$ | $50.43_{\pm1.5}$ | $80.16_{\pm1.8}$ |
| ChebNet | $82.15_{\pm1.6}$ | $64.06_{\pm1.2}$ | $50.42_{\pm1.4}$ | $74.28_{\pm0.9}$ | $76.98_{\pm0.7}$ | $73.14_{\pm1.1}$ | $49.82_{\pm1.6}$ | $77.40_{\pm1.6}$ |
| ChebNetII | $84.17_{\pm3.1}$ | $70.03_{\pm2.8}$ | $64.29_{\pm2.9}$ | $78.31_{\pm4.1}$ | $\mathbf{81.14}_{\pm3.6}$ | $\mathbf{77.09}_{\pm3.9}$ | $52.69_{\pm2.7}$ | $80.06_{\pm3.3}$ |
| CayleyNet | $83.06_{\pm4.2}$ | $62.73_{\pm5.1}$ | $42.28_{\pm5.3}$ | $74.12_{\pm4.7}$ | $77.21_{\pm4.5}$ | $71.45_{\pm4.8}$ | $51.89_{\pm3.9}$ | $76.33_{\pm5.8}$ |
| APPNP | $\underline{84.45}_{\pm4.4}$ | $65.26_{\pm6.9}$ | $49.68_{\pm3.9}$ | $77.26_{\pm4.8}$ | $73.24_{\pm7.3}$ | $72.94_{\pm2.3}$ | $44.36_{\pm1.9}$ | $73.85_{\pm3.3}$ |
| GPRGNN | $80.26_{\pm2.0}$ | $58.41_{\pm1.4}$ | $45.29_{\pm1.7}$ | $73.90_{\pm2.5}$ | $73.12_{\pm4.0}$ | $69.17_{\pm2.6}$ | $47.07_{\pm2.8}$ | $77.93_{\pm1.9}$ |
| ARMA | $83.21_{\pm2.7}$ | $69.23_{\pm2.2}$ | $61.21_{\pm3.4}$ | $76.62_{\pm3.6}$ | $79.51_{\pm2.9}$ | $73.27_{\pm2.7}$ | $\underline{53.60}_{\pm1.9}$ | $78.34_{\pm1.1}$ |
| att-Graph-level NLSFs | $84.13_{\pm1.5}$ | $68.17_{\pm1.0}$ | $\underline{65.94}_{\pm1.6}$ | $\underline{82.69}_{\pm1.9}$ | $80.51_{\pm1.2}$ | $74.26_{\pm1.8}$ | $52.49_{\pm0.7}$ | $79.06_{\pm1.2}$ |
| att-Pooling-NLSFs | $\mathbf{86.89}_{\pm1.2}$ | $\mathbf{71.02}_{\pm1.3}$ | $\mathbf{69.94}_{\pm1.0}$ | $\mathbf{84.89}_{\pm0.9}$ | $\underline{80.95}_{\pm1.4}$ | $\underline{76.78}_{\pm1.9}$ | $\mathbf{55.28}_{\pm1.7}$ | $\mathbf{82.19}_{\pm1.3}$ |

DiffPool [101]. For dataset splitting, we apply the random split following [94, 101, 67, 104], using 80% for training, 10% for validation, and 10% for testing. This random splitting process is repeated 10 times, and we report the average performance along with the standard deviation. For the baseline algorithms, we use the source code released by the authors and optimize their hyperparameters using Optuna [1]. We fine-tune the hyperparameters of each model to achieve the highest possible accuracy. Detailed hyperparameter settings for both the baselines and our method are provided in App. E.

Tab. 3 presents the graph classification accuracy. Notably, the att-Graph-level NLSFs (i.e., NLSFs without the synthesis and pooling process) perform the second best on the ENZYMES and PROTEINS. Additionally, att-Pooling-NLSFs consistently outperform att-Graph-level NLSFs, indicating that the node features learned in our Node-level NLSFs representation are more expressive, corroborating our theoretical findings in Sec. 4.3. Furthermore, our att-Pooling-NLSFs consistently outperform all baselines on the MUTAG, PTC, ENZYMES, PROTEINS, IMDB-M, and COLLAB datasets. For NCI1 and IMDB-B, att-Pooling-NLSFs rank second and are comparable to ChebNetII.

## 6 Summary

We presented an approach for defining non-linear filters in the spectral domain of graphs in a transferable and equivariant way. Transferability between different graphs is achieved by using the input signal as a basis for the synthesis operator, making the NLSF depend only on the eigenspaces of the GSO and not on an arbitrary choice of the eigenvectors. While different graph-signals may be of different sizes, the spectral domain is a fixed Euclidean space independent of the size and topology of the graph. Hence, our spectral approach represents graphs as vectors. We note that standard spectral methods do not have this property since the coefficients of the signal in the frequency domain depend on an arbitrary choice of eigenvectors, while our representation depends only on the eigenspaces. We analyzed the universal approximation and expressivity power of NLSFs through metrics that are pulled back from this Euclidean vector space. From a geometric point of view, our NLSFs are motivated by respecting graph functional shift symmetries, making them related to Euclidean CNNs.

**Limitation and Future Work.** One limitation of NLSFs is that, when deployed on large graphs, they only depend on the leading eigenvectors of the GSO and their orthogonal complement. However, important information can also lie within any other band. In future work, we plan to explore NLSFs that are sensitive to eigenvalues that lie within a set of bands of interest, which can be adaptive to the graph.

## Acknowledgments

The work of YEL and RT was supported by the European Union's Horizon 2020 research and innovation programme under grant agreement No. 802735-ERC-DIFFOP. The work of RL was supported by the Israel Science Foundation grant No. 1937/23, and the United States - Israel Binational Science Foundation (NSF-BSF) grant No. 2024660.

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

# A  Non-linear Activation Functions Break the Functional Symmetry

We offer a construction for complex-valued signals and first order derivative as the graph Laplacian, which is easy to extend to the real case and second order derivative. Consider a circular 1D grid with 100 nodes and the first order central difference as the Laplacian. Here, the Laplacian eigenvectors are the standard Fourier modes. In this case, one can see that a graph functional shift is any operator that is diagonal in the frequency domain and multiplies each frequency by any complex number with the unit norm. Consider the nonlinearity that takes the real part of the signal and then applies ReLU, which we denote in short by ReLU. Consider a graph signal $x = (e^{i\pi 10n/100} + e^{i\pi 20n/100})_{n=0}^{99}$. We consider a graph functional shift $S$ that shifts frequencies 10 and 20 at a distance of 5, and every other frequency is not shifted. Namely, frequency 10 is multiplied by $e^{-i\pi 50/100} = -i$, frequency 20 by $e^{-i\pi = -i\pi 100/100} = -1$, and every other frequency is multiplied by 1. Consider also the classical shift $D$ that translates the whole signal by 5 uniformly. Since $x$ consists only of the frequencies 10 and 20, it is easy to see that $Sx = Dx$. Hence, $\text{ReLU}(Sx) = \text{ReLU}(Dx) = D(\text{ReLU}(x))$. Conversely, if we apply $\text{ReLU}(x)$ and only then shift, note that $\text{ReLU}(x)$ consists of many frequencies in addition to 10 and 20. For example, by nonnegativity of ReLU, $\text{ReLU}(x)$ has a nonzero DC component (zeroth frequency). Now, $S(\text{ReLU}(x))$ only translates the 10 and 20 frequencies, so we have $S(\text{ReLU}(x)) \neq D(\text{ReLU}(x)) = \text{ReLU}(S(x))$.

# B  Illustrating Functional Translations

Here we present an additional discussion and illustrations on the new notions of symmetry.

## B.1  Functional Translation of a Gaussian Signal with Different Speeds at Different Frequencies

To illustrate the concepts of relaxed symmetry and functional translation, we present a toy example involving the classical translation and a functional translation of a Gaussian signal on a 2D grid with a standard deviation of 1.

The classical translation involves moving the entire signal uniformly across the grid. This uniform movement can be represented as $I'(x, y) = I(x - t_x, y - t_y)$, where $t_x$ and $t_y$ are the translation amounts in the $x$ and $y$ directions, respectively, and $x - t_x$ and $y - t_y$ are circular translations, namely, subtractions modulo the size of the circular grid. For simplicity, we consider $t_x = t_y = t$. For instance, if $t = 5$, the Gaussian is shifted by 5 units in both the $x$ and $y$ directions. Fig. 3 top-row shows the classical translation for $t = 0$, $t = 5$, $t = 10$, and $t = 15$. We see that every part of the signal shifts at the same rate and direction, preserving the overall shape of the Gaussian signal. Note that classical translation is equivalent to modulation of the frequency domain, i.e. $\widehat{I'}(u, v) = \widehat{I}(u, v)e^{-i2\pi(ut_x + vt_y)}$, where $\widehat{I}(u, v)$ is the Fourier transform of $I(x, y)$. Therefore, we can view the classical translation as a specific type of functional translation.

Next, we illustrate a functional translation that is based on a frequency-dependent movement. Specifically, the Gaussian signal is decomposed into low and high-frequency components based on two indicator band-pass filters. In our example, the translation parameter $t$ is different for the low and high-frequency components: low frequencies are shifted by $t_{low}$ while high frequencies are shifted by $t_{high}$. This functional translation is defined via modulations in the frequency domain given by $\widehat{I'}_{low}(u, v) = \widehat{I}_{low}(u, v)e^{-i2\pi(ut_{x,low} + vt_{y,low})}$ for low-frequency components and $\widehat{I'}_{high}(u, v) = \widehat{I}_{high}(u, v)e^{-i2\pi(ut_{x,high} + vt_{y,high})}$ for high-frequency components. The combined functionally translated signal in the frequency domain is then $\widehat{I'} = (\widehat{I'}_{low}, \widehat{I'}_{high})$. Fig. 3 bottom-row demonstrates the functional translation for $t = 0$, $t = 5$, $t = 10$, and $t = 15$. We observe that the low-frequency components (smooth parts) of the signal move at one speed, while high-frequency components move at another. This demonstrates that functional symmetries are typically more rich than domain symmetries.

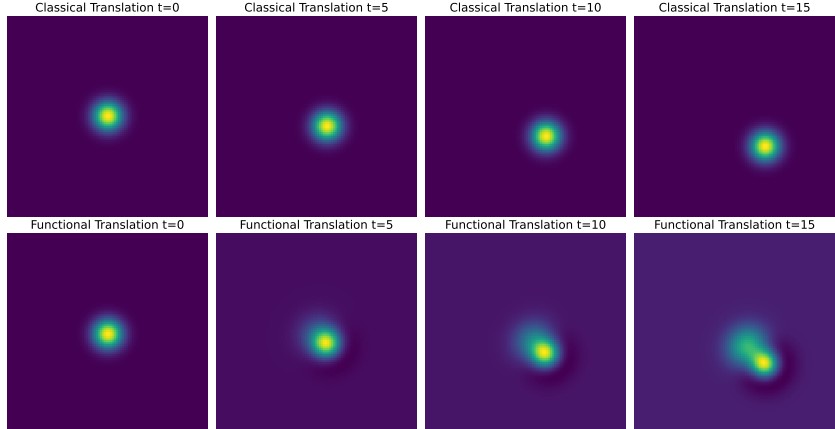

Figure 3: Comparison of classical and functional translation of a Gaussian signal. Top Row (Classical Translation): The Gaussian signal moves uniformly across the grid without changing shape. Bottom Row (Functional Translation): The Gaussian signal translates as low-frequency components move at different speeds than high-frequency components, demonstrating relaxed symmetry.

## B.2 Robust Graph Functional Shifts in Image Translation and MNIST Classification on Perturbed Grids

We present another toy example to illustrate that functional translations are more stable than hard symmetries of the graph, namely, graph automorphisms (isomorphisms from the graph to itself). Automorphisms are another analog to translations on general graphs, which competes with our notion of functional shifts. Consider as an example the standard 2D circular grid (discretizing the torus). The automorphisms of the grid include all translations. However, this notion of hard symmetry is very sensitive to graph perturbations. If one adds or deletes even one edge in the graph, the automorphism group becomes much smaller and does not contain the translations anymore.

In contrast, we claim that functional shifts are not so sensitive to graph perturbations. To demonstrate this empirically, we conducted the following toy experiment. We add a Gaussian noise to the edge weights of the 2D grid to create a perturbed graph. Given a standard domain shift, we optimize the coefficients of a functional shift so it is as close as possible to the classical domain shift of the clean grid in Frobenius error. Fig. 4 presents an example of classically and functionally shifted image of the digit 5. The original digit (left), a classical domain translation of the clean grid (middle), and a functional translation constructed from the perturbed graph (right) to match the classical translation. We see that standard translations can be approximated by a graph functional shift on a perturbed grid.

We further demonstrate the robustness to graph perturbations of NLSFs trained on MNIST classification. The experimental setup follows previous work in [22, 71, 61]. We compare the NLSF on the clean grid (i.e., without Gaussian noise) to the NLSF on the perturbed grid. Tab. 4 presents the classification accuracy of the NLSF. We see that the classification performance is almost unaffected by the perturbation, indicating that NLSFs on the perturbed grid can roughly express CNN-like operations (translation equivariant functions).

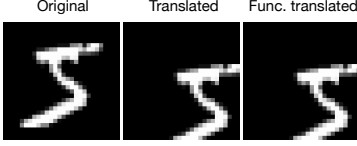

Figure 4: Approximate a standard translation by functional translation on a perturbed graph.

|  | MNIST | Perturbed MNIST |
|---|---|---|
| Ours | 99.19 | 99.16 |

Table 4: Classification accuracy on the MNIST and perturbed MNIST using NLSFs.

## C  Special Cases of NLSFs

We present two special cases of NLSFs as follows.

### C.1  Diagonal NLSFs

In Sec. 3, we introduced the NLSFs with the output dimension $\widetilde{d}$, which is a tunable hyperparameter. Here, we present a special case when $\widetilde{d} = d$ such that the multiplication and division in synthesis are operated diagonally. Specifically, the diagonal analysis and synthesis in the spectral index case and in the filter bank case are defined respectively by

$$\mathcal{A}^{\text{ind, diag}}(\mathbf{\Delta}, \mathbf{X}) = \left( \|\mathbf{P}_j \mathbf{X}\|_{\text{sig}} \right)_{j=1}^{J+1} \in \mathbb{R}^{(J+1)\times d} \quad \text{and} \tag{7}$$

$$\mathcal{A}^{\text{val, diag}}(\mathbf{\Delta}, \mathbf{X}) = \left( \|g_j(\mathbf{\Delta})\mathbf{X}\|_{\text{sig}} \right)_{j=1}^{K+1} \in \mathbb{R}^{(K+1)\times d} \tag{8}$$

and

$$\mathcal{S}^{\text{ind, diag}}(\mathbf{R}; \mathbf{\Delta}, \mathbf{X}) = \sum_{j=1}^{J+1} \mathbf{r}_j \odot \frac{\mathbf{P}_j \mathbf{X}}{\|\mathbf{P}_j \mathbf{X}\|_{\text{sig}}^a + e} \quad \text{and} \tag{9}$$

$$\mathcal{S}^{\text{val, diag}}(\mathbf{R}; \mathbf{\Delta}, \mathbf{X}) = \sum_{j=1}^{K+1} \mathbf{r}_j \odot \frac{g_j(\mathbf{\Delta})\mathbf{X}}{\|g_j(\mathbf{\Delta})\mathbf{X}\|_{\text{sig}}^a + e}, \tag{10}$$

where the product and division are element-wise along the channel direction. That is, $\mathbf{r}_j \odot \frac{\mathbf{P}_j \mathbf{X}}{\|\mathbf{P}_j \mathbf{X}\|_{\text{sig}}^a + e} = \left[ r_{j1} \frac{\mathbf{P}_j \mathbf{X}_{:,1}}{\|\mathbf{P}_j \mathbf{X}_{:,1}\|_2^a + e}, \dots, r_{jd} \frac{\mathbf{P}_j \mathbf{X}_{:,d}}{\|\mathbf{P}_j \mathbf{X}_{:,d}\|_2^a + e} \right]$ in the spectral index case (resp. $\mathbf{r}_j \odot \frac{g_j(\mathbf{\Delta})\mathbf{X}}{\|g_j(\mathbf{\Delta})\mathbf{X}\|_{\text{sig}}^a + e} = \left[ r_{j1} \frac{g_j(\mathbf{\Delta})\mathbf{X}_{:,1}}{\|g_j(\mathbf{\Delta})\mathbf{X}_{:,1}\|_2^a + e}, \dots, r_{jd} \frac{g_j(\mathbf{\Delta})\mathbf{X}_{:,d}}{\|g_j(\mathbf{\Delta})\mathbf{X}_{:,d}\|_2^a + e} \right]$ in the filter bank case). Here, $\mathbf{R} = (\mathbf{r}_j)_{j=1}^{J+1} \in \mathbb{R}^{(J+1)\times d}$ in the spectral index case (resp. $\mathbf{R} = (\mathbf{r}_j)_{j=1}^{K+1} \in \mathbb{R}^{(K+1)\times d}$ in the filter bank case) are the spectral coefficients to be synthesized and $a, e$ are as before.

For Node-level diag-NLSFs, we define the Index diag-NLSFs and Value diag-NLSFs as follows

$$\Theta_{\text{ind, diag}}(\mathbf{\Delta}, \mathbf{X}) = \mathcal{S}_{\mathbf{\Delta},\mathbf{X}}^{(\text{ind, diag})}(\Psi(\mathcal{A}_{\mathbf{\Delta}}^{(\text{ind})}(\mathbf{X}))) = \sum_{j=1}^{J+1} \left[ \Psi \left( \|\mathbf{P}_i \mathbf{X}\|_{\text{sig}} \right)_{i=1}^{J+1} \right]_j \frac{\mathbf{P}_j \mathbf{X}}{\|\mathbf{P}_j \mathbf{X}\|_{\text{sig}}^a + e}, \tag{11}$$

$$\Theta_{\text{val, diag}}(\mathbf{\Delta}, \mathbf{X}) = \mathcal{S}_{\mathbf{\Delta},\mathbf{X}}^{(\text{val, diag})}(\Psi(\mathcal{A}_{\mathbf{\Delta}}^{(\text{val})}(\mathbf{X}))) = \sum_{j=1}^{K+1} \left[ \Psi \left( \|g_i(\mathbf{\Delta})\mathbf{X}\|_{\text{sig}} \right)_{i=1}^{K+1} \right]_j \frac{g_j(\mathbf{\Delta})\mathbf{X}}{\|g_j(\mathbf{\Delta})\mathbf{X}\|_{\text{sig}}^a + e}, \tag{12}$$

where $\Psi : \mathbb{R}^{(J+1)d} \to \mathbb{R}^{(J+1)d}$ in $\Theta_{\text{ind, diag}}$ and $\Psi : \mathbb{R}^{(K+1)d} \to \mathbb{R}^{(K+1)d}$ in $\Theta_{\text{val, diag}}$. To adjust the feature output dimension at each node, we apply an MLP with shared weights to all nodes after the NLSF. We present the empirical study of diag-NLSFs in App. F.7.

### C.2  Leading NLSFs

In Sec. 3.3, we introduce the NLSFs with the orthogonal complement. Specifically, the $(J+1)$-th filter in the Index NLSFs is given by $\mathbf{P}_{J+1} = \mathbf{I} - \sum_{j=1}^{J} \mathbf{P}_j$, and the $(K+1)$-th filter in Vale NLSFs is defined as $g_{K+1}(\mathbf{\Delta}) = \mathbf{I} - \sum_{j=1}^{K} g_j(\mathbf{\Delta})$. To explore the effects of the orthogonal complement, we focus on the leading NLSFs in both the Index NLSFs and Vale NLSFs, considering only the first $J$ and $K$ filters without including their orthogonal complements. In this case, the analysis and synthesis in the spectral index case and in the filter bank case are respectively given by

$$\mathcal{A}^{\text{ind, lead}}(\mathbf{R}; \mathbf{\Delta}, \mathbf{X}) = \left( \|\mathbf{P}_i \mathbf{X}\|_{\text{sig}} \right)_{i=1}^{J} \in \mathbb{R}^{Jd} \quad \text{and}$$

$$\mathcal{A}^{\text{val, lead}}(\mathbf{R}; \mathbf{\Delta}, \mathbf{X}) = \left( \|g_i(\mathbf{\Delta})\mathbf{X}\|_{\text{sig}} \right)_{i=1}^{K} \in \mathbb{R}^{Kd}, \tag{13}$$

and

$$\mathcal{S}^{\text{ind, lead}}(\mathbf{R}; \boldsymbol{\Delta}, \mathbf{X}) = \left[ \frac{\mathbf{P}_j \mathbf{X}}{\|\mathbf{P}_j \mathbf{X}\|_{\text{sig}}^a + e} \right]_{j=1}^J \mathbf{R} \text{ and}$$

$$\mathcal{S}^{\text{val, lead}}(\mathbf{R}; \boldsymbol{\Delta}, \mathbf{X}) = \left[ \frac{g_j(\boldsymbol{\Delta})\mathbf{X}}{\|g_j(\boldsymbol{\Delta})\mathbf{X}\|_{\text{sig}}^a + e} \right]_{j=1}^K \mathbf{R},$$

(14)

where $\mathbf{P}_j$ and $g_j(\boldsymbol{\Delta})$ are defined as in Sec. 3.2. The lead-NLSFs are then defined by

$$\Theta_{\text{ind, lead}}(\boldsymbol{\Delta}, \mathbf{X}) = \mathcal{S}_{\boldsymbol{\Delta}, \mathbf{X}}^{(\text{ind, lead})}(\Psi_{\text{ind}}(\mathcal{A}_{\boldsymbol{\Delta}}^{(\text{ind})}(\mathbf{X})))$$

$$= \left[ \frac{\mathbf{P}_j \mathbf{X}}{\|\mathbf{P}_j \mathbf{X}\|_{\text{sig}}^a + e} \right]_{j=1}^J \left[ \Psi_{\text{ind}} \left( \|\mathbf{P}_i \mathbf{X}\|_{\text{sig}} \right) \right]_{i=1}^J ,$$

(15)

$$\Theta_{\text{val, ind}}(\boldsymbol{\Delta}, \mathbf{X}) = \mathcal{S}_{\boldsymbol{\Delta}, \mathbf{X}}^{(\text{val, ind})}(\Psi_{\text{val}}(\mathcal{A}_{\boldsymbol{\Delta}}^{(\text{val})}(\mathbf{X})))$$

$$= \left[ \frac{g_j(\boldsymbol{\Delta})\mathbf{X}}{\|g_j(\boldsymbol{\Delta})\mathbf{X}\|_{\text{sig}}^a + e} \right]_{j=1}^K \left[ \Psi_{\text{val}} \left( \|g_i(\boldsymbol{\Delta})\mathbf{X}\|_{\text{sig}} \right) \right]_{i=1}^K ,$$

(16)

where $\Psi_{\text{ind}} : \mathbb{R}^{Jd} \to \mathbb{R}^{Jd \times J\widetilde{d}}$, $\Psi_{\text{val}} : \mathbb{R}^{Kd} \to \mathbb{R}^{Kd \times K\widetilde{d}}$, and $\widetilde{d}$ is the output dimension. To adjust the feature output dimension, we apply an MLP with shared weights to all nodes after the NLSF.

Similar to App. C.1, when considering $\widetilde{d} = d$ such that the multiplication and division in synthesis are operated diagonally, we have the lead-diag-NLSFs defined by

$$\Theta_{\text{ind, lead, diag}}(\boldsymbol{\Delta}, \mathbf{X}) = \mathcal{S}_{\boldsymbol{\Delta}, \mathbf{X}}^{(\text{ind, lead, diag})}(\Psi(\mathcal{A}_{\boldsymbol{\Delta}}^{(\text{ind})}(\mathbf{X})))$$

$$= \sum_{j=1}^J \left[ \Psi \left( \|\mathbf{P}_i \mathbf{X}\|_{\text{sig}} \right)_{i=1}^J \right]_j \frac{\mathbf{P}_j \mathbf{X}}{\|\mathbf{P}_j \mathbf{X}\|_{\text{sig}}^a + e},$$

(17)

$$\Theta_{\text{val, lead, diag}}(\boldsymbol{\Delta}, \mathbf{X}) = \mathcal{S}_{\boldsymbol{\Delta}, \mathbf{X}}^{(\text{val, lead, diag})}(\Psi(\mathcal{A}_{\boldsymbol{\Delta}}^{(\text{val})}(\mathbf{X})))$$

$$= \sum_{j=1}^K \left[ \Psi \left( \|g_i(\boldsymbol{\Delta})\mathbf{X}\|_{\text{sig}} \right)_{i=1}^K \right]_j \frac{g_j(\boldsymbol{\Delta})\mathbf{X}}{\|g_j(\boldsymbol{\Delta})\mathbf{X}\|_{\text{sig}}^a + e},$$

(18)

where $\Psi : \mathbb{R}^{Jd} \to \mathbb{R}^{Jd}$ in $\Theta_{\text{ind, lead, diag}}$ and $\Psi : \mathbb{R}^{Kd} \to \mathbb{R}^{Kd}$ in $\Theta_{\text{val, lead, diag}}$. We present the empirical study of lead-NLSFs and lead-diag-NLSFs in App. F.7.

# D Theoretical Analysis

We note that the numbering of the statements corresponds to the numbering used in the paper, and we have also included several separately numbered propositions and lemmas that are used in supporting the proofs presented. We restate the claim of each statement for convenience.

## D.1 Equivariance of NLSFs

We start with two simple lemmas that characterize the graph functional shifts. The lemmas directly follow the fact that two normal operators commute iff the projections upon their eigenspaces commute.

**Projection to Eigenspaces Case.** Note that

$$\mathbb{R}^N = \left( \prod_{j=1}^{J+1} \mathbf{P}_j \mathbb{R}^N \right),$$

where $\prod$ denotes direct products of linear spaces and the $(J+1)$-th filter is the orthogonal complement, given by $\mathbf{P}_{J+1} = \mathbf{I} - \sum_{j=1}^J \mathbf{P}_j$. Denote by $\oplus$ the direct sum of operators.

**Lemma D.1.** $\mathbf{U}$ *is in* $\mathcal{U}_{\boldsymbol{\Delta}}^J$ *iff it has the form* $\mathbf{U} = \left( \bigoplus_{j=1}^{J+1} \mathbf{U}_j \right)$, *where* $\mathbf{U}_j$ *is a unitary operator in* $\mathbf{P}_j \mathbb{R}^N$.

*Proof.* Let $\mathbf{U} \in \mathcal{U}_{\boldsymbol{\Delta}}^J$. Since $\mathbf{U}$ commute with $\mathbf{P}_j$ for $j = 1, \dots, J$, and with $\mathbf{I}$, it also commutes with $\mathbf{P}_{J+1} = \mathbf{I} - \sum_{j=1}^{J} \mathbf{P}_j$. Therefore, we can write

$$\mathbf{U} = \sum_{j=1}^{J+1} \mathbf{P}_j \mathbf{U} = \sum_{j=1}^{J+1} \mathbf{P}_j^2 \mathbf{U} = \sum_{j=1}^{J+1} \mathbf{P}_j \mathbf{U} \mathbf{P}_j.$$

Now, when restricting $\mathbf{P}_j \mathbf{U} \mathbf{P}_j$ to an operator in $\mathbf{P}_j \mathbb{R}^N$, it is unitary. Indeed, for every $\mathbf{v} = \mathbf{P}_j \mathbf{v} \in \mathbf{P}_j \mathbb{R}^N$ and $\mathbf{u} = \mathbf{P}_j \mathbf{u} \in \mathbf{P}_j \mathbb{R}^N$, since $\mathbf{U}$ is unitary and $\mathbf{P}_j$ is self-adjoint and satisfies $\mathbf{P}_j^2 = \mathbf{P}_j$, we have

$$\langle \mathbf{P}_j \mathbf{U} \mathbf{P}_j \mathbf{v}, \mathbf{u} \rangle = \langle \mathbf{v}, \mathbf{P}_j \mathbf{U}^{-1} \mathbf{P}_j \mathbf{u} \rangle,$$

and $\mathbf{P}_j \mathbf{U}^{-1} \mathbf{P}_j$ is the inverse of $\mathbf{P}_j \mathbf{U} \mathbf{P}_j$ in $\mathbf{P}_j \mathbb{R}^N$, since for every $\mathbf{v} \in \mathbf{P}_j \mathbb{R}^N$

$$\mathbf{P}_j \mathbf{U}^{-1} \mathbf{P}_j \mathbf{P}_j \mathbf{U} \mathbf{P}_j \mathbf{v} = \mathbf{P}_j \mathbf{U}^{-1} \mathbf{U} \mathbf{P}_j \mathbf{v} = \mathbf{P}_j \mathbf{v} = \mathbf{v},$$

and similarly $\mathbf{P}_j \mathbf{U} \mathbf{P}_j \mathbf{P}_j \mathbf{U}^{-1} \mathbf{P}_j \mathbf{v} = \mathbf{v}$. Here, an invertible normal operator commutes with an orthogonal projection if and only if its inverse does.

The other direction is trivial. $\qquad\square$

**Projection to Bands Case.** Note that

$$\mathbb{R}^N = \left( \prod_{j=1}^{K+1} g_j(\boldsymbol{\Delta}) \mathbb{R}^N \right),$$

where the $(K+1)$-th filter is the orthogonal complement, given by $g_{K+1}(\boldsymbol{\Delta}) = \mathbf{I} - \sum_{j=1}^{K} g_j(\boldsymbol{\Delta})$.

**Lemma D.2.** $\mathbf{U}$ *is in* $\mathcal{U}_{\boldsymbol{\Delta}}^g$ *iff it has the form* $\mathbf{U} = \left( \bigoplus_{j=1}^{K+1} \mathbf{U}_j \right)$, *where* $\mathbf{U}_j$ *is a unitary operator in* $g_j(\boldsymbol{\Delta}) \mathbb{R}^N$.

The proof is analogous to the proof of Lemma D.1.

### D.1.1 Proof of Propositions 1

**Proposition 1.** *Index NLSFs in Eq. (4) are equivariant to the graph functional shifts* $\mathcal{U}_{\boldsymbol{\Delta}}$, *and Value NLSFs in Eq. (5) are equivariant to the relaxed graph functional shifts* $\mathcal{U}_{\boldsymbol{\Delta}}^g$.

*Proof.* We start with Index-NLSFs. Consider the Index NLSF defined as in Eq. (4)

$$\Theta_{\mathrm{ind}}(\boldsymbol{\Delta}, \mathbf{X}) = \left[ \frac{\mathbf{P}_j \mathbf{X}}{\|\mathbf{P}_j \mathbf{X}\|_{\mathrm{sig}}^a + e} \right]_{j=1}^{J+1} \left[ \Psi_{\mathrm{ind}} \left( \|\mathbf{P}_i \mathbf{X}\|_{\mathrm{sig}} \right) \right]_{i=1}^{J+1}.$$

We need to show that for any unitary operator $\mathbf{U} \in \mathcal{U}_{\boldsymbol{\Delta}}$,

$$\Theta_{\mathrm{ind}}(\boldsymbol{\Delta}, \mathbf{U} \mathbf{X}) = \mathbf{U} \Theta_{\mathrm{ind}}(\boldsymbol{\Delta}, \mathbf{X}).$$

Consider $\mathbf{U} \in \mathcal{U}_{\boldsymbol{\Delta}}$ and apply it to the graph signal $\mathbf{X}$. The Index NLSF with the transformed input is given by

$$\Theta_{\mathrm{ind}}(\boldsymbol{\Delta}, \mathbf{U} \mathbf{X}) = \left[ \frac{\mathbf{P}_j \mathbf{U} \mathbf{X}}{\|\mathbf{P}_j \mathbf{U} \mathbf{X}\|_{\mathrm{sig}}^a + e} \right]_{j=1}^{J+1} \left[ \Psi_{\mathrm{ind}} \left( \|\mathbf{P}_i \mathbf{U} \mathbf{X}\|_{\mathrm{sig}} \right) \right]_{i=1}^{J+1}.$$

Since $\mathbf{U} \in \mathcal{U}_{\boldsymbol{\Delta}}$, it commutes with $\mathbf{P}_j$ for all $j$, i.e., $\mathbf{U} \mathbf{P}_j = \mathbf{P}_j \mathbf{U}$. Using this commutation property, we can rewrite the norm and the projections as

$$\|\mathbf{P}_j \mathbf{U} \mathbf{X}\|_{\mathrm{sig}} = \|\mathbf{U} \mathbf{P}_j \mathbf{X}\|_{\mathrm{sig}}.$$

In addition, since $\mathbf{U}$ is a unitary matrix, it preserves the norm. Therefore, we have

$$\|\mathbf{U} \mathbf{P}_j \mathbf{X}\|_{\mathrm{sig}} = \|\mathbf{P}_j \mathbf{X}\|_{\mathrm{sig}}.$$

Substituting these expressions back into the definition of the Index NLSFs gives

$$\Theta_{\text{ind}}(\boldsymbol{\Delta}, \mathbf{U}\mathbf{X}) = \left[ \frac{\mathbf{U}\mathbf{P}_j\mathbf{X}}{\|\mathbf{P}_j\mathbf{X}\|_{\text{sig}}^a + e} \right]_{j=1}^{J+1} \left[ \Psi_{\text{ind}} \left( \|\mathbf{P}_i\mathbf{X}\|_{\text{sig}} \right) \right]_{i=1}^{J+1}.$$

Notice that $\mathbf{U}$ appears linearly in the numerator of the fraction. Hence, we can factor it out by

$$\Theta_{\text{ind}}(\boldsymbol{\Delta}, \mathbf{U}\mathbf{X}) = \mathbf{U} \left[ \frac{\mathbf{P}_j\mathbf{X}}{\|\mathbf{P}_j\mathbf{X}\|_{\text{sig}}^a + e} \right]_{j=1}^{J+1} \left[ \Psi_{\text{ind}} \left( \|\mathbf{P}_i\mathbf{X}\|_{\text{sig}} \right) \right]_{i=1}^{J+1}.$$

The expression inside the summation is exactly the original Index NLSFs applied to $\mathbf{X}$, so

$$\Theta_{\text{ind}}(\boldsymbol{\Delta}, \mathbf{U}\mathbf{X}) = \mathbf{U}\Theta_{\text{ind}}(\boldsymbol{\Delta}, \mathbf{X}).$$

Therefore, we have shown that applying the unitary operator $\mathbf{U}$ to the graph signal $\mathbf{X}$ results in the Index NLSFs being transformed by the same unitary operator $\mathbf{U}$, proving the equivariance property.

The proof for value-NLSF follows the same steps. $\qquad\square$

## D.2 Expressivity and Universal Approximation

In this section, we focus on value parameterized NLSFs. The analysis for index-NLSF is equivalent.

### D.2.1 Proof of Lemma 1 - Linear Node-level NLSF Exhaust the Linear Operators that Commute with Functional Shifts

We next show that node-level linear NLSFs exhaust the space of linear operators that commute with graph functional shifts (on the fixed graph).

**Lemma 1.** *A linear operator $\mathbb{R}^{N \times d} \to \mathbb{R}^{N \times d}$ commute with $\mathcal{U}_{\boldsymbol{\Delta}}^J$ (resp. $\mathcal{U}_{\boldsymbol{\Delta}}^g$) iff it is a NLSF based on a linear function $\Psi$ in Eq. (4) (resp. Eq. (5)).*

*Proof.* For simplicity, we restrict the analysis to the case of 1D signals ($d = 1$). The extension to a general dimension $d$ is natural.

By Lemma D.2, the unitary operators $\mathbf{U}$ in $\mathcal{U}_{\boldsymbol{\Delta}}^g$ are exhausted by the operators of the form

$$\mathbf{U} = \left( \bigoplus_{j=1}^{K+1} \mathbf{U}_j \right),$$

where $\mathbf{U}_j$ is any unitary operator in $g_j(\boldsymbol{\Delta})\mathbb{R}^N$. Hence, since the unitary representation $\mathbf{T} \mapsto \mathbf{T}$ of the group of unitary operators in $\mathbb{R}^m$ (for any $m \in \mathbb{N}$) is irreducible, by Schur's lemma [85] any linear operator $\mathbf{B}$ that commutes with all operators of $\mathcal{U}_{\boldsymbol{\Delta}}^g$ must be a scalar times the identity when projected to $g_j(\boldsymbol{\Delta})\mathbb{R}^N$. Namely, $\mathbf{B}$ has the form

$$\mathbf{B} = \left( \bigoplus_{j=1}^{K+1} (b_j\mathbf{I}_j) \right),$$

where $\mathbf{I}_j$ is the identity operator in $g_j(\boldsymbol{\Delta})\mathbb{R}^N$. This means that linear node-level NLSFs exhaust the space of linear operators that commute with $\mathcal{U}_{\boldsymbol{\Delta}}^g$.

The case of hard graph functional shifts is treated similarly. $\qquad\square$

### D.2.2 Node-Level Universal Approximation

The above analysis motivates the following construction of a metric on $\mathbb{R}^N$. Given the filter bank $\{g_j\}_{j=1}^{K+1}$, on graph with $d$-dimensional signals, we define the standard Euclidean metric $\text{dist}_{\text{E}}$ in the spectral domain $\mathbb{R}^{(K+1) \times d}$. We pull back the Euclidean metric to the spatial domain to define a graph-signal metric. Namely, for two signals $\mathbf{X}$ and $\mathbf{X}'$, their distance is defined to be

$$\text{dist}_{\boldsymbol{\Delta}}\big(\mathbf{X}, \mathbf{X}'\big) := \text{dist}_{\text{E}}\big(\mathcal{A}(\boldsymbol{\Delta}, \mathbf{X}), \mathcal{A}(\boldsymbol{\Delta}', \mathbf{X}')\big).$$

This defines a pseudometric on the space of graph-signals. To obtain a metric, we consider each equivalence class of signals with zero distance as a single point. Namely, we define the space $\mathcal{N}_{\boldsymbol{\Delta}} := \mathbb{R}^N/\mathrm{dist}_{\boldsymbol{\Delta}}$ to be the space of signals with 1D features modulo $\mathrm{dist}$. In $\mathbf{N}_{\boldsymbol{\Delta}}$, the pseudometric $\mathrm{dist}$ becomes a metric, and $\mathcal{A}_{\boldsymbol{\Delta}}/\mathrm{dist}_{\boldsymbol{\Delta}}$ an isometry of metric spaces[2].

Now, since MLPs $\Psi$ can approximate any continuous function $\mathbb{R}^{(K+1)\times d} \to \mathbb{R}^{(K+1)\times d'}$ (by the universal approximation theorem), and by the fact that $\mathcal{A}_{\boldsymbol{\Delta}}/\mathrm{dist}_{\boldsymbol{\Delta}}$ is an isometry, node-level NLSFs based on MLPs in the spectral domain can approximate any continuous function from signals with $d$ features to signal with $d'$ features $(\mathcal{N}_{\boldsymbol{\Delta}})^d \to (\mathcal{N}_{\boldsymbol{\Delta}})^{d'}$.

### D.2.3  Graph-Level Universal approximation

The above analysis also motivates the construction of a graph-signal metric for graph-level tasks. For graphs with $d$-channel signals, we define the standard Euclidean metric $\mathrm{dist}_{\mathrm{E}}$ in the spectral domain $\mathbb{R}^{(K+1)\times d}$. We pull back the Euclidean metric to the spatial domain to define a graph-signal metric. Namely, for two graphs with Laplacians and signals $(\boldsymbol{\Delta}, \mathbf{X})$ and $(\boldsymbol{\Delta}', \mathbf{X}')$, their distance is defined to be

$$\mathrm{dist}\big((\boldsymbol{\Delta}, \mathbf{X}), (\boldsymbol{\Delta}', \mathbf{X}')\big) := \mathrm{dist}_{\mathrm{E}}\big(\mathcal{A}(\boldsymbol{\Delta}, \mathbf{X}), \mathcal{A}(\boldsymbol{\Delta}', \mathbf{X}')\big)$$

This defines a pseudometric on the space of graph-signals. To obtain a metric, we consider equivalence classes of graph-signals with zero distance as a single point. Namely, we define the space $\mathcal{G}$ to be the space of graph-signal modulo $\mathrm{dist}$. In $\mathcal{G}$, the function $\mathrm{dist}$ becomes a metric, and $\mathcal{A}$ an isometry.

By the isometry property, $\mathcal{G}$ inherits any approximation property from $\mathbb{R}^{(K+1)\times d}$. For example, since MLPs can approximate any continuous function $\mathbb{R}^{(K+1)\times d} \to \mathbb{R}^{d'}$, the space of NLSFs based on MLPs $\Psi$ has a universality property: any continuous function $\mathcal{G} \to \mathbb{R}^{d'}$ with respect to $\mathrm{dist}$ can be approximated by a NLSF based on an MLP.

### D.2.4  Graph-Level Expressivity of Pooling-NLSFs

We now show that pooling-NLSF are more expressive than graph-level NLSF if the norm in Eq. (4) and Eq. (5) is $L_p$ with $p \neq 2$.

First, we show that there are graph-signals that graph-level-NLSFs cannot separate and Pooling-NLSFs can. Consider an index NLSF with norm $L_1$ normalize by $1/N$. Namely, for $\mathbf{a} \in \mathbb{R}^N$,

$$\|\mathbf{a}\|_1 = \frac{1}{N}\sum_{n=1}^{N}|a_n|.$$

The general case is similar.

For the two graphs, take the graph Laplacian

$$\begin{pmatrix} 1 & -1 \\ -1 & 1 \end{pmatrix}$$

with eigenvalues $0, 1$, and corresponding eigenvectors $(1, 1)$ and $(1, -1)$. Take the graph Laplacian

$$\begin{pmatrix} 1 & -1 & 0 \\ -1 & 2 & -1 \\ 0 & -1 & 1 \end{pmatrix}$$

with eigenvalues $0, 1, 3$. The first two eigenvectors are $(1, 1, 1)$ and $(1, 0, -1)$ respectively.

Consider an Index-NLSF based on two eigenprojections $\mathbf{P}_1, \mathbf{P}_2$. As the signal of the first graph take $(1, 1) + (1, -1)$, and for the second graph take $(1, 1, 1) + \frac{3}{2}(1, 0, -1)$. Both graph-signals have the same spectral coefficients $(1, 1)$, so graph-level NLSF cannot separate them. Suppose that the NLSF is

$$\Theta_{\mathrm{val}}(\boldsymbol{\Delta}, \mathbf{X}) = (\|\mathbf{P}_1\mathbf{X}\|_{\mathrm{sig}}^a + e)\frac{\mathbf{P}_1\mathbf{X}}{\|\mathbf{P}_1\mathbf{X}\|_{\mathrm{sig}}^a + e} + (\|\mathbf{P}_2\mathbf{X}\|_{\mathrm{sig}}^a + e)\frac{\mathbf{P}_2\mathbf{X}}{\|\mathbf{P}_2\mathbf{X}\|_{\mathrm{sig}}^a + e}.$$

---

[2]$\mathcal{A}_{\boldsymbol{\Delta}}/\mathrm{dist}_{\boldsymbol{\Delta}}$ operates on an equivalence class $[\mathbf{X}]$ of signals by applying $\mathcal{A}_{\boldsymbol{\Delta}}$ on an arbitrary element $\mathbf{Y}$ of $[\mathbf{X}]$. The output of $\mathcal{A}_{\boldsymbol{\Delta}}$ on $\mathbf{Y}$ does not depend on the specific representative $\mathbf{Y}$, but only on $[\mathbf{X}]$.

The outcome of the corresponding Pooling NLSF on the two graphs is

$$1 = \|(1+1, 1-1)\|_1 \neq \|(1+3/2, 1, 1-3/2)\|_1 = \frac{4}{3}.$$

Hence, Pooling-NLSFs separate these inputs, while graph-level-NLSFs do not.

Next, we show that Pooling-NLSFs are at least as expressive as graph-level NLSFs. In particular, any graph-level NLSF can be expressed as a pooling NLSF.

Let $\Psi$ be a graph-level NLSF. Define the node-level NLSF that chooses one spectral index $j$ with a nonzero value (e.g., the band with the largest coefficient) and projects upon it the value $\Psi(\boldsymbol{\Delta}, \mathbf{X})(\|\mathbf{P}_j\mathbf{X}\|_{\mathrm{sig}}^a + e)/\|\mathbf{P}_j X\|_{\mathrm{sig}}$. Hence, before pooling, the NLSF gives

$$\Theta(\boldsymbol{\Delta}, \mathbf{X}) = \Psi(\boldsymbol{\Delta}, \mathbf{X})(\|\mathbf{P}_j\mathbf{X}\|_{\mathrm{sig}}^a + e)\frac{\mathbf{P}_j\mathbf{X}}{\|\mathbf{P}_j\mathbf{X}\|_{\mathrm{sig}}^a + e}/\|\mathbf{P}_j X\|_{\mathrm{sig}},$$

where $j$ depends on the spectral coefficients (e.g., it is the index of the largest spectral coefficient). Hence, after pooling, the Pooling NLSF returns $\Psi(\boldsymbol{\Delta}, \mathbf{X})$, which coincides with the output of the graph-level NLSF.

Now, let us show that for $p = 2$, they have the same expressivity. For any NLSF,

$$\Theta(\boldsymbol{\Delta}, \mathbf{X}) = \left\| \sum_{j=1}^{J} \left[ \Psi\left( \|\mathbf{P}_i\mathbf{X}\|_{\mathrm{sig}} \right)_{i=1}^{J} \right]_j \frac{\mathbf{P}_j\mathbf{X}}{\|\mathbf{P}_j\mathbf{X}\|_{\mathrm{sig}}^a + e} \right\|_{\mathrm{sig}}^2$$

$$= \sum_{j=1}^{J} \left[ \Psi\left( \|\mathbf{P}_i\mathbf{X}\|_{\mathrm{sig}} \right)_{i=1}^{J} \right]_j^2 \frac{\|\mathbf{P}_j\mathbf{X}\|_{\mathrm{sig}}^2}{\|\mathbf{P}_j\mathbf{X}\|_{\mathrm{sig}}^a + e}.$$

This is a generic fully spectral NLSF.

### D.2.5 Discussion on Graph-Level Expressivity of NLSFs

We offer additional discussion on the expressivity of graph-level NLSFs. Note that graph-level NLSFs are bounded by the expressive power of MPNNs with random positional encodings. For example, in [81], the authors showed that random features improve GNN expressivity, distinguishing certain structures that deterministic GNNs cannot. The Lanczos algorithm for computing the leading $J$ eigenvectors can be viewed as a message-passing algorithm with a randomly initialized $J$-channel signal.

The example in App. D.2.4 can be written as a standard linear graph spectral filter, followed by the pooling (readout) function. This shows that graph-level NLSFs (without synthesis and pooling) are not more expressive than standard spectral GNNs with pooling. In the other direction, there are no graph-signals that can be separated by a graph-level NLSF but not by a spectral GNN. Given two graph-signals that an NLSF can separate, we can build a specific standard spectral GNN that gives the same output as the NLSF specifically for the two given graph-signals. However, several considerations should be noted:

1. the feature dimension of this GNN would is as large as the combined number of eigenvalues of the two graphs times the number of features,

2. this GNN is designed to fit these two specific graphs, and will not work for other graphs, and,

3. if implemented via polynomial filters, the polynomial order would have to be the combined number of eigenvalues of the two graphs, which makes it very unstable.

Let us explain the architecture next. Given the two graphs $(G_1, f_1)$ with $N_1$ vertices and $(G_2, f_2)$ with $N_2$ vertices, with node features of dimension $D$, define band-pass filters that separate the combined set of eigenvalues of the two given graphs. Concatenate these filters to build a single multi-channel filter. Namely, this filter maps the signal to the sequence of band-pass filtered signal – each band at a different channel, for a total of $D(N_1 + N_2)$ channels. If the band-pass filters are based on polynomials, each polynomial would be chosen to be zero on all eigenvalues except for one. Apply $L_2$-norm pooling on each channel to obtain a single feature of dimension $D(N_1 + N_2)$. This gives the sequence of norms of the signal at the bands, where, for the two given graphs, the two

pooled signals of the two graphs are supported on disjoint sets of indices. Hence, the linear spectral filter contains the frequency information of the NLSF. Then, one can apply the MLP of the NLSF to the channels corresponding to the first graph, and to the channels corresponding to the second graph. This means that the linear spectral GNN gives the exact same outputs on $(G_1, f_1)$ and $(G_2, f_2)$ as the NLSF.

Regarding pooling-NLSF, we do not offer an answer to the question whether standard spectral GNNs are more powerful (assuming an unlimited budget) than pooling-NLSFs, or vice-versa. More practically meaningful questions would be:

1. Compare the expressivity of standard spectral GNNs to NLSFs for a given budget of parameters.

2. How many graph-signal can a single spectral GNN separate, vs a single NLSF, of the same budget.

We leave these questions as open problems for future research. We note that, in practice, NLSFs with the same budget as standard spectral GNNs perform better.

### D.3 Stable Invertibility of Synthesis

We present the analysis for diagonal synthesis defined from App. C.1. In the fixed graph setting, given any 1D signal $\mathbf{X}$ such that $g_j(\mathbf{\Delta})\mathbf{X} \neq 0$ for every $j \in [K+1]$, we show that the synthesis operator $\mathcal{S}_{\mathbf{\Delta},\mathbf{X}}^{\text{val, diag}}$ is stably invertible.

Since we consider 1D signal $\mathbf{X}$, the diagonal synthesis in filter bank case in Eq. (10) can be written as

$$\mathcal{S}_{\mathbf{\Delta},\mathbf{X}}^{\text{val, diag}} = \mathbf{H}\mathbf{r}',$$

where

$$\mathbb{R}^{N \times (K+1)} \ni \mathbf{H} = \left[ \frac{g_1(\mathbf{\Delta})\mathbf{X}}{\|g_1(\mathbf{\Delta})\mathbf{X}\|_2^a + e}, \dots, \frac{g_{K+1}(\mathbf{\Delta})\mathbf{X}}{\|g_{K+1}(\mathbf{\Delta})\mathbf{X}\|_2^a + e} \right],$$

$\frac{g_j(\mathbf{\Delta})\mathbf{X}}{\|g_j(\mathbf{\Delta})\mathbf{X}\|_2^a + e} \in \mathbb{R}^N$, and $\mathbf{r}' = [r_1, r_2, \dots, r_{K+1}] \in \mathbb{R}^{K+1}$.

Since $(g_{j_1}(\mathbf{\Delta})\mathbf{X})^\top (g_{j_2}(\mathbf{\Delta})\mathbf{X}) = 0$ for all $j_1, j_2 \in [K+1]$, the matrix $\mathbf{H}^\top \mathbf{H}$ is a diagonal matrix with entries $\left\| \frac{g_1(\mathbf{\Delta})\mathbf{X}}{\|g_1(\mathbf{\Delta})\mathbf{X}\|_2^a + e} \right\|_2^2$. Therefore, the singular values of $\mathbf{H}$ are

$$\sigma_j = \frac{\|g_1(\mathbf{\Delta})\mathbf{X}\|_2}{\|g_1(\mathbf{\Delta})\mathbf{X}\|_2^a + e}$$

the right singular vectors are the standard basis elements $\mathbf{e}_j$ in $\mathbb{R}^{K+1}$, and the left singular vectors are

$$\frac{g_j(\mathbf{\Delta})\mathbf{X}}{\|g_j(\mathbf{\Delta})\mathbf{X}\|_2}.$$

for $j \in [K+1]$. Hence, we have

$$\left\| \mathcal{S}_{\mathbf{\Delta},\mathbf{X}}^{\text{val, diag}} \right\|_2 = \max_j \sigma_j$$

and

$$\left\| \left( \mathcal{S}_{\mathbf{\Delta},\mathbf{X}}^{\text{val, diag}} \right)^{-1} \right\|_2 = \frac{1}{\min_j \sigma_j}.$$

Suppose that $a = 1$ and $e = 0$. In this case, $\mathcal{S}_{\mathbf{\Delta},\mathbf{X}}^{\text{val, diag}}$ is an isometry from the spectral domain to a subspace of the signal space $\mathbb{R}^N$. Analysis is the adjoint of synthesis. This analysis can be extended to the index parametrization case for diagonal synthesis, and to higher dimensional signals.

### D.4 Uniform Approximation of Graphs with $J$ Eigenvectors

In this section, we develop the setting under which the low-rank approximation of GSOs with their leading eigenvectors can be interpreted as a uniform approximation (Sec. 4.4).

### D.4.1 Cut Norm

The cut norm of $\mathbf{M} \in \mathbb{R}^{N \times N}$ is defined to be

$$\|\mathbf{M}\|_\square := \frac{1}{N^2} \sup_{S,T \subset [N]} \Big| \sum_{i \in S} \sum_{j \in T} m_{i,j} \Big|. \tag{19}$$

The distance between two matrices in cut norm is defined to be $\|\mathbf{M} - \mathbf{M}'\|_\square$.

The cut norm has the following interpretation, which has precise formulation in terms of the weak regularity lemma [32, 65]. Any pair of (deterministic) graphs are close to each other in cut norm if and only if they can be described as pseudo-random graphs sampled from the same stochastic block model. Hence, the cut norm is a meaningful notion of graph similarity for practical graph machine learning, where graphs are noisy and can represent the same underlying phenomenon even if they have different sizes and topologies. In addition, the distance between non-isomorphic graphons is always positive in cut norm [66]. In this context, the work in [59] showed that GNNs with normalized sum aggregation cannot separate graphs that have zero distance in the cut norm. This means that the cut norm is sufficiently discriminative for practical machine learning on graphs.

### D.4.2 The Constructive Weak Regularity Lemma in Hilbert Spaces

The following lemma, called the *constructive weak regularity lemma in Hilbert spaces*, was proven in [29]. It is an extension of the classical respective result from [65].

We define the Frobenius norm normalized by $1/N$ as

$$\|\mathbf{M}\|_{\mathrm{F}} = \sqrt{\frac{1}{N} \sum_{i,j=1}^{N} |m_{i,j}|^2}.$$

**Lemma D.3.** *[[29]] Let $\{\mathcal{K}_j\}_{j \in \mathbb{N}}$ be a sequence of nonempty subsets of a real Hilbert space $\mathcal{H}$ and let $\delta \geq 0$. Let $J > 0$, let $R \geq 1$ such that $J/R \in \mathbb{N}$, and let $g \in \mathcal{H}$. Let $m$ be randomly uniformly sampled from $[J]$. Then, in probability $1 - \frac{1}{R}$ (with respect to the choice of $m$), any vector of the form*

$$g^* = \sum_{j=1}^{m} \gamma_j f_j \quad \text{such that} \quad \boldsymbol{\gamma} = (\gamma_j)_{j=1}^m \in \mathbb{R}^m \quad \text{and} \quad \mathbf{f} = (f_j)_{j=1}^m \in \mathcal{K}_1 \times \ldots \times \mathcal{K}_m$$

*that gives a close-to-best Hilbert space approximation of $g$ in the sense that*

$$\|g - g^*\| \leq (1 + \delta) \inf_{\boldsymbol{\gamma}, \mathbf{f}} \|g - \sum_{i=1}^m \gamma_i f_i\|, \tag{20}$$

*where the infimum is over $\boldsymbol{\gamma} \in \mathbb{R}^m$ and $\mathbf{f} \in \mathcal{K}_1 \times \ldots \times \mathcal{K}_m$, also satisfies*

$$\forall w \in \mathcal{K}_{m+1}, \quad |\langle w, g - g^* \rangle| \leq \|w\| \|g\| \sqrt{\frac{R}{J} + \delta}.$$

### D.4.3 Proof of Theorem 1

**Theorem 1.** *Let $\mathbf{M}$ be a symmetric matrix with entries bounded by $|m_{i,j}| \leq \alpha$, and let $J \in \mathbb{N}$. Suppose $m$ is sampled uniformly from $[J]$, and let $R \geq 1$ s.t. $J/R \in \mathbb{N}$. Consider $\phi_1, \ldots, \phi_m$ as the leading eigenvectors of $\mathbf{M}$, with eigenvalues $\mu_1, \ldots, \mu_m$ ordered by their magnitudes $|\mu_1| \geq \ldots \geq |\mu_m|$. Define $\mathbf{C} = \sum_{k=1}^m \mu_k \phi_k \phi_k^\top$. Then, with probability $1 - \frac{1}{R}$ (w.r.t. the choice of $m$),*

$$\|\mathbf{M} - \mathbf{C}\|_\square < \frac{3\alpha}{2} \sqrt{\frac{R}{J}}.$$

*Proof.* Let us use Lemma D.3, with $\mathcal{H} = \mathbb{R}^{N \times N}$, and $\mathcal{K}_j = \mathcal{K}$ the set of symmetric rank one matrices of the form $\mathbf{v}\mathbf{v}^\top$ where $\mathbf{v} \in \mathbb{R}^N$ is a column vector. Denote by $\mathcal{Y}_m$ the space of linear combinations of $m$ elements of $\mathcal{K}$, which is the space of symmetric matrices of rank bounded by $m$. For the Hilbert space norm, we take the Frobenius norm. In the setting of the lemma, we take $g = \mathbf{M}$, and $g^* \in \mathcal{Y}_m$, and $\delta = 0$. By the lemma, with probability $1 - 1/R$, any Frobenius minimizer $\mathbf{C}$,

namely, that satisfies $\|\mathbf{M} - \mathbf{C}\|_{\mathrm{F}} = \min_{\mathbf{C}' \in \mathcal{Y}_m} \|\mathbf{M} - \mathbf{C}'\|_{\mathrm{F}}$, also satisfies

$$\langle \mathbf{Y}, \mathbf{M} - \mathbf{C} \rangle \leq \|\mathbf{Y}\|_{\mathrm{F}} \|\mathbf{M}\|_{\mathrm{F}} \sqrt{\frac{R}{J}} \leq \alpha \|\mathbf{Y}\|_{\mathrm{F}} \sqrt{\frac{R}{J}}$$

for every $\mathbf{Y} \in \mathcal{Y}_m$. Hence, for every choice of subset $S, T \subset [N]$, we have

$$\left| \sum_{i \in S} \sum_{j \in T} (m_{i,j} - c_{i,j}) \right|$$

$$= \frac{1}{2} \left| \sum_{i \in S} \sum_{j \in T} (m_{i,j} - c_{i,j}) + \sum_{i \in T} \sum_{j \in S} (m_{i,j} - c_{i,j}) \right|$$

$$= \frac{1}{2} \left| \sum_{i \in S \cup T} \sum_{j \in S \cup T} (m_{i,j} - c_{i,j}) - \sum_{i \in S} \sum_{j \in S} (m_{i,j} - c_{i,j}) - \sum_{i \in T} \sum_{j \in T} (m_{i,j} - c_{i,j}) \right|$$

$$\leq \left\| \frac{1}{2} \left\langle \mathbb{1}_{S \cup T} \mathbb{1}_{S \cup T}^{\top}, \mathbf{M} - \mathbf{C} \right\rangle \right\|_{\mathrm{F}} + \left\| \frac{1}{2} \left\langle \mathbb{1}_S \mathbb{1}_S^{\top}, \mathbf{M} - \mathbf{C} \right\rangle \right\|_{\mathrm{F}} + \left\| \frac{1}{2} \left\langle \mathbb{1}_T \mathbb{1}_T^{\top}, \mathbf{M} - \mathbf{C} \right\rangle \right\|_{\mathrm{F}}$$

$$\leq \frac{3\alpha}{2} \sqrt{\frac{R}{J}},$$

where for a set $S \subset [N]$, denote by $\mathbb{1}_S \in \mathbb{R}^N$ the column vector with 1 at coordinates in $S$ and 0 otherwise.

Hence, we also have

$$\|\mathbf{M} - \mathbf{C}\|_{\square} \leq \frac{3\alpha}{2} \sqrt{\frac{R}{J}}.$$

Lastly, note that by the best rank-$m$ approximation theorem (Eckart–Young–Mirsky Theorem [92, Thm. 5.9]), any Frobenius minimizer $\mathbf{C}$ is the projection upon the $m$ leading eigenvectors of $\mathbf{M}$ (or some choice of these eigenvectors in case of multiplicity higher than 1).

$\square$

Now, one can use the adjacency matrix $\mathbf{A}$ as $\mathbf{M}$ in Thm. 1. When working with sparse matrices of $E \ll N^2$ edges, to achieve a meaningful scaling of cut distance, we re-normalize the cut norm and define

$$\|\mathbf{M}\|_{\square}^{(E)} = \frac{N^2}{E} \|\mathbf{M}\|_{\square}.$$

With this norm, Thm. 1 gives

$$\|\mathbf{M} - \mathbf{C}\|_{\square}^{(E)} < \frac{3\alpha N^2}{2E} \sqrt{\frac{R}{J}}. \tag{21}$$

While this bound is not uniformly small in $N$, it is still independent of $\mathbf{M}$. In contrast, the error bounds for spectral and Frobenius norms do depend on the specific properties of $\mathbf{M}$.

Now, if we want to apply Thm. 1 to other GSOs $\mathbf{\Delta}$, we need to make some assumptions. Note that when the GSO is a Laplacian, we take as the leading eigenvectors the ones corresponding to the smallest eigenvalues, not the largest ones. To make the theorem applicable, we need to reorder the eigenvectors of $\mathbf{\Delta}$. This can be achieved by applying a decreasing function $h$ to $\mathbf{\Delta}$, such as $h(\mathbf{\Delta}) = \mathbf{\Delta}^{-1}$. The role of $h$ is to amplify the eigenspaces of $\mathbf{\Delta}$ in which most of the energy of signals interest (the ones that often appear in the dataset) is concentrated. Under the assumption that $h(\mathbf{\Delta})$ has entries bounded by some not-too-high $\alpha > 0$, one can now justify approximating GSOs by low-rank approximations based on the smallest eigenvectors.

## D.5   NLSFs on Random Geometric Graphs

In this section we consider the $L_2[N]$ norm normalized by $1/N^{1/2}$, namely,

$$\|\mathbf{v}\|_2 = \sqrt{\frac{1}{N}\sum_{n=1}^{N}|v_n|^2}.$$

We follow a similar analysis to [74], mostly skipping the rigorous Monte-Carlo error rate proofs, as these are standard.

Let $\mathcal{S}$ be a compact metric space with metric d and with a Borel probability measure, that we formally call the *standard measure* $dx$. Let $r > 0$, and denote by $B_r(x)$ the ball of radius $r$ about $x \in \mathcal{S}$. Let $V(x)$ be the volume of $B_r(x)$ with respect to the standard measure (note that $V(x)$ need not be constant). We consider an underlying Laplacian on the space $\mathcal{S}$ defined on signals $f : \mathcal{S} \to \mathbb{R}$ by

$$\mathcal{L}f(x) = C\int_{B_r(x)}(f(x) - f(y))dy, \tag{22}$$

where we assume $C = 1$ in the analysis below without loss of generality. In case the integral in Eq. (22) is normalized by $1/r^2V(x)$, this Laplacian was called $\rho$-Laplacian in [15], which we denote by $\mathcal{L}_r$. Such a Laplacian is related to Korevaar-Schoen type energies [52], in which the limit case of the radius $r$ going to 0 is considered. It was shown in [15] that the $\rho$-Laplacian is self-adjoint with spectrum supported inside some interval $[0, Q]$, for some $Q > 0$, where for some $0 < R < Q$, the part of the spectrum in $[0, R) \cup (R, Q]$ is discrete (consisting of isolated eigenvalues with finite multiplicities). The intuition behind this result is that, for the $\rho$-Laplacian $\mathcal{L}_r$, we have

$$\mathcal{L}_r f = \frac{1}{r^2}f - \frac{1}{V(x)r^2}\int_{B_r(x)}f(y)dy. \tag{23}$$

The first term in the right-hand-side of Eq. (23) is a scaled version of the identity operator, and the second term is a compact self-adjoint integral operator, and hence has a discrete spectrum with accumulation point at 0, or no accumulation point. After showing that the sum of these two operators is self-adjoint, we end up with only one accumulation point of the spectrum of $\mathcal{L}_r$.

We show below that $\mathcal{L}$ is self-adjoint under the assumption that $V(x)$ is bounded from above. In this case it must also be positive semi-definite (the proof is equivalent to the positivity of the combinatorial graph Laplacian). Consider the decomposition

$$\mathcal{L}f(x) = CV(x)f(x) - C\int_{B_r(x)}f(y)dy. \tag{24}$$

The second term of Eq. (24) is a compact integral operator, and hence only has an accumulation point at 0, or has no accumulation point. The first term is a multiplication operator by the real-valued function $V(x)$, so it is self-adjoint and its spectrum is the closure of $\{V(x) \mid x \in \mathcal{S}\}$. We suppose that $V(x)$ is bounded from below by some $R' > 0$ and also bounded from above. This shows that $\mathcal{L}$ is bounded and self-adjoint (as a sum of two bounded self-adjoint operators). Under these assumptions, it is reasonable to further assume that the spectrum of $\mathcal{L}$ in the interval $[0, R)$, for some $R > 0$, is discrete, with accumulation point at $R$. This happens, for example, if $V(x) = V$ is constant.

We now assume that the signal $f$ consists only of frequencies in $[0, R)$. This means that we can project $\mathcal{L}$ upon this part of the spectrum, giving a self-adjoint operator with discrete spectrum, and accumulation point of the eigenvalues only at $R$.

Let $w : \mathcal{S} \to \mathbb{R}$ be a measurable weight function with $\int_{\mathcal{S}}w(x)dx = 1$. Suppose that $w(x)$ is continuous and varies slowly over $\mathcal{S}$. Namely, we assume that $w(x) \approx w(y)$ for every $y \in B_r(x)$. To generate a random geomertic graph of $N$ nodes, we sample $N$ points $\mathbf{x} = \{x_n\}_{n=1}^{N}$ independently from the weighted probability measure $w(x)dx$. We connect node $x_j$ to $x_j$ by an edge if and only if $d(x_i, x_j) \leq r$ to obtain the adjacency matrix $\mathbf{A}$. Let $\mathbf{L}$ be the combinatorial Laplacian with respect to $\mathbf{A}$, and $\mathbf{N}$ the normalized symmetric Laplacian. The number of samples inside the ball of radius $r$ around $x$ is approximately $w(x)V(x)N$. Hence, the degree of the node $x_n$ is approximately $w(x_n)V(x_n)N$.

We consider the leading $J$ eigenvectors the ones corresponding to the smallest eigenvalues of $\mathcal{L}$ in the interval $[0, R)$, where $J \ll N$. We order the eigenvalues in increasing order. Let $\mathcal{PW}(J)$ be the space spanned by the $J$ leading eigenvectors of $\mathcal{L}$, called the *Paley-Wiener* space of $\mathcal{L}$. Let $p_J$ be the

projection upon the Paley-Wiener space of $\mathcal{L}$. Let $R_N : L^2(\mathcal{S}) \to L^2[N]$ be the sampling operator, defined by $(R_N f)_j = f(x_j)$.

Let $f : \mathcal{S} \to \mathbb{R}$ be a bounded signal over the metric space and suppose that $f \in \mathcal{PW}(J)$. Denote $\mathbf{f} = (f(x_n))_{n=1}^N = R_N f$. By Monte Carlo theory, we have

$$(\mathbf{Lf})_j \approx w(x_j)N \int_{B_r(x_j)} (f(x) - f(y))dy = w(x_j)N\mathcal{L}f(x_j),$$

So, in case the sampling is uniform. i.e., $w(x) = 1$, we have

$$\mathbf{L} \approx N\mathcal{L},$$

pointwise. To make this precise, let us recall Hoeffding's Inequality [43].

**Theorem D.1** (Hoeffding's Inequality [43]). *Let $Y_1, \ldots, Y_N$ be independent random variables such that $a \leq Y_n \leq b$ almost surely. Then, for every $k > 0$,*

$$\mathbb{P}\left( \left| \frac{1}{N} \sum_{n=1}^N (Y_n - \mathbb{E}[Y_n]) \right| \geq k \right) \leq 2 \exp\left( -\frac{2k^2 N}{(b-a)^2} \right).$$

Using Hoeffding's Inequality, one can now show that there is an event of probability more than $1 - p$ in which for every $j \in [N]$, the error between $(\frac{1}{N}\mathbf{L}R_N f)_j$ and $\mathcal{L}f$ satisfies

$$\left\| R_N \mathcal{L}f - \frac{1}{N}\mathbf{L}R_N f \right\| = \mathcal{O}\left( J\sqrt{\frac{\log(1/p) + \log(N) + \log(2)}{N}} \right).$$

The fact that different graphs of different sizes $N$ approximate $\mathcal{L}$ with different scaling factors means that $\mathbf{L}$ is not value transferable. Let us show that $\mathbf{L}$ is index transferable.

We now evoke the transferability theorem – a slight reformulation of Section 3 of Theorem 4 in [60].

**Theorem D.2.** *Let $\mathcal{L}$ be a compact operator on $L^2(\mathcal{S})$ and $\mathbf{L}$ an operator on $\mathbb{C}^N$. Let $J \in \mathbb{N}$ and let $R_N$ be the sampling operator defined above. Let $\mathcal{PW}(J)$ be the space spanned by the $J$ leading eigenvectors of $\mathcal{L}$, and let $f \in \mathcal{PW}(J)$. Let $g : \mathbb{R} \to \mathbb{R}$ be Lipschitz with Lipschitz constant $D$. Then,*

$$\left\| R_N g(\mathcal{L})f - \frac{1}{N}g(\mathbf{L})R_N f \right\| \leq D\sqrt{J} \left\| R_N \mathcal{L}f - \frac{1}{N}\mathbf{L}R_N f \right\|.$$

For every leading eigenvalue $\lambda_j$ of $\mathcal{L}$ let $\gamma_{i_j}$ be the leading eigenvalue of $\mathcal{L}$ closest to $\lambda$, where, if there are repeated eigenvalues, $i_j$ is chosen arbitrarily from the eigenvalues of $\mathbf{L}$ that best approximate $\lambda_i$. Let

$$\delta = \min\{\alpha, \beta\},$$

where

$$\alpha = \min_{j \in [J]} \min\left\{ \min_{i_j \neq i \in [N]} |\lambda_j - \gamma_i| , \min_{j \neq m \in [J]} |\lambda_j - \lambda_m| , \min_{j \neq m \in [J]} |\gamma_{i_j} - \lambda_m| \right\}$$

and

$$\beta = \min_{i \in [J], n \in [N]} |\gamma_i - \gamma_n|$$

and suppose that $\delta > 0$. For each $j \in [N]$ there exists a Lipschitz continuous function $g_j$ with Lipschitz constant $\delta^{-1}$ such that $g_j(\lambda_j) = g_j(\gamma_{i_j}) = 1$ and $\gamma$ is zero on all other leading eigenvalues of $\mathcal{L}$ and all other eigenvalues of $\mathbf{L}$. Hence,

$$g_j(\mathcal{L})f = p_j^{\mathcal{L}}f, \quad g_j(\mathbf{L})R_N f = p_{i_j}^{\mathbf{L}}\mathbf{f},$$

where $p_j^{\mathcal{L}}$ is the projection upon the space spanned by the $j$-th eigenvector of $\mathcal{L}$, and $p_{i_j}^{\mathbf{L}}$ is the projection upon the space spanned by the $i_j$-th eigenvector of $\mathbf{L}$ .

Now, by the transferability theorem,

$$\left\| R_N p_j^{\mathcal{L}}f - p_{i_j}^{\mathbf{L}}\mathbf{f} \right\| \leq D\sqrt{J} \left\| R_N \mathcal{L}f - \frac{1}{N}\mathbf{Lf} \right\| = \mathcal{O}(\sqrt{J/N}).$$

By induction over $j$, with base $j = 1$, we must have $i_j = j$ for every $j \in [J]$.

Now, note that by standard Monte Carlo theory (evoking Hoeffding's inequality again and intersecting events), we have

$$\left| \left\| R_N p_j^{\mathcal{L}}f \right\|_2^2 - \left\| p_j^{\mathcal{L}}f \right\|_2^2 \right| = \mathcal{O}(J^{-1/2})$$

Table 5: Node classification datasets statistics.

| Dataset | # Nodes | # Classes | # Edges | # Features |
|---------|---------|-----------|---------|------------|
| Cora | 2708 | 7 | 5278 | 1433 |
| Citeseer | 3327 | 6 | 4552 | 3703 |
| Pubmed | 19717 | 5 | 44324 | 500 |
| Chameleon | 2277 | 5 | 31371 | 2325 |
| Squirrel | 5201 | 5 | 198353 | 2089 |
| Actor | 7600 | 5 | 26659 | 932 |

in high probability. Hence, by the fact that

$$\left\| R_N p_j^{\mathcal{L}} f \right\|_2^2 - \left\| p_j^{\mathcal{L}} f \right\|_2^2 = \left( \left\| R_N p_j^{\mathcal{L}} f \right\|_2 - \left\| p_j^{\mathcal{L}} f \right\|_2 \right) \left( \left\| R_N p_j^{\mathcal{L}} f \right\|_2 + \left\| p_j^{\mathcal{L}} f \right\|_2 \right)$$

and $\left\| R_N p_j^{\mathcal{L}} f \right\|_2 + \left\| p_j^{\mathcal{L}} f \right\|_2$ is bounded from below by the constant $\left\| p_j^{\mathcal{L}} f \right\|_2$, we have

$$\left| \left\| R_N p_j^{\mathcal{L}} f \right\|_2 - \left\| p_j^{\mathcal{L}} f \right\|_2 \right| = \mathcal{O}(J^{-1/2}).$$

This shows that

$$\left\| p_j^{\mathcal{L}} f \right\|_2 \approx \left\| p_j^{\mathbf{L}} \mathbf{f} \right\|_2,$$

which shows index transferability. Namely, for two graphs of $N$ and $N'$ nodes sampled from $\mathcal{S}$, with corresponding Laplacians $\mathbf{L}$ and $\mathbf{L}'$, by the triangle inequality,

$$\left\| p_j^{\mathbf{L}} \mathbf{f} \right\|_2 \approx \left\| p_j^{\mathbf{L}'} \mathbf{f} \right\|_2.$$

Next, we show value transferability for $\mathbf{N}$. Here,

$$\mathbf{N} f(x_j) \approx \frac{1}{V(x_j)} \mathcal{L} f(x_j).$$

Hence, if $V(x) = V$ is constant, we have $\mathbf{N} \approx \mathcal{L}$ up to a constant that does not depend on $N$. In this case, by a similar analysis to the above, $\mathbf{N}$ is value transferable. We note that $\mathbf{N}$ is also index transferable, but value transferability is guaranteed in a more general case, where we need not assume a separable spectrum.

## E Additional Details on Experimental Study

We describe the experimental setups and additional details of our experiments in Sec. 5. The experiments are performed on NVIDIA DGX A100.

### E.1 Semi-Supervised Node Classification

We provide a detailed overview of the experimental settings for semi-supervised node classification tasks, along with the validated hyperparameters used in our benchmarks.

**Datasets.** We consider six datasets in node classification tasks, including Cora, Citeseer, Pubmed, Chameleon, Squirrel, and Actor. The detailed statistics of the node classification benchmarks are summarized in Tab. 5. For datasets splitting on citation graphs (Cora, Citeseer, and Pubmed), we follow the standard splits from [100], using 20 nodes per class for training, 500 nodes for validation, and 1000 nodes for testing. For heterophilic graphs (Chameleon, Squirrel, and Actor), we adopt the sparse splitting method from [16, 41], allocating 2.5% of samples for training, 2.5% for validation, and 95% for testing. The classification quality is assessed by computing the average classification accuracy across 10 random splits, along with a 95% confidence interval.

**Baselines.** For GCN, GAT, SAGE, ChebNet, ARMA, and APPNP, we use the implementations from the PyTorch Geometric library [28]. For other baselines, we use the implementations released by the respective authors.

Table 6: Graph classification datasets statistics.

| Dataset | # Graphs ($Z$) | # Classes ($C$) | $|N_i|_{\min}$ | $|N_i|_{\max}$ | $|N_i|_{\mathrm{avg}}$ | # Features ($d$) |
|---------|---------|---------|---------|---------|---------|---------|
| MUTAG | 188 | 2 | 10 | 28 | 17.93 | 7 |
| PTC | 344 | 2 | 2 | 64 | 14.29 | 18 |
| ENZYMES | 600 | 6 | 2 | 126 | 32.63 | 18 |
| PROTEINS | 1113 | 2 | 4 | 620 | 39.06 | 29 |
| NCI1 | 4110 | 2 | 3 | 111 | 29.87 | 37 |
| IMDB-B | 1000 | 2 | 12 | 136 | 19.77 | None |
| IMDB-M | 1500 | 3 | 7 | 89 | 13.00 | None |
| COLLAB | 5000 | 3 | 32 | 492 | 74.50 | None |

**Hyperparameters Settings.** The hidden dimension is set to be either 64 or 128 for all models and datasets. We implement our proposed Node-level NLSFs using PyTorch and optimize the model with the Adam optimizer [48]. To determine the optimal dropout probability, we search within the range $[0, 0.9]$ in increments of $0.1$. The learning rate is examined within the set $\{1e^{-1}, 5e^{-2}, 1e^{-2}, 5e^{-3}, 1e^{-3}\}$. We explore weight decay values within the set $\{1e^{-2}, 5e^{-3}, 1e^{-3}, 5e^{-4}, 1e^{-4}, 5e^{-5}, 1e^{-5}, 5e^{-6}, 1e^{-6}, 0.0\}$. Furthermore, the number of layers is varied from 1 to 10. The number of leading eigenvectors $J$ in Index NLSFs is set within $[1, 1e^2]$. The decay rate in the Value NLSFs is determined using dyadic sampling within the set $\{\frac{1}{4}, \frac{1}{3}, \frac{1}{2}, \frac{2}{3}, \frac{3}{4}\}$, the sampling resolution $S$ within $[1, 1e^2]$, and the number of bands in Value NLSFs $K \leq S$. For hyperparameter optimization, we conduct a grid search using Optuna [1] for each dataset. An early stopping criterion is employed during training, stopping the process if the validation loss does not decrease for 200 consecutive epochs.

## E.2 Graph Classification

We provide a comprehensive description of the experimental settings for graph classification tasks and the validated hyperparameters used in our benchmarks. The results are reported in the main paper.

**Problem Setup.** Consider a set of $Z$ graphs $\mathcal{G}_Z = \{G_1, G_2, \ldots, G_Z\}$, where in each graph $G_i = ([N_i], \mathcal{E}_i, \mathbf{A}_i, \mathbf{X}_i)$, we have $N_i$ nodes for each graph $G_i$, $\mathcal{E}_i$ represents the edge set, $\mathbf{A}_i \in \mathbb{R}^{N_i \times N_i}$ denotes the edge weights, and $\mathbf{X}_i \in \mathbb{R}^{N_i \times d}$ represents the node feature matrix with $d$-dimensional node attributes. Let $\mathbf{Y}_Z \in \mathbb{R}^{Z \times C}$ be the label matrix with $C$ classes such that $y_{i,j} = 1$ if the graph $G_i$ belongs to the class $j$, and $y_{i,j} = 0$ otherwise. Given a set of $Z'$ graphs $\mathcal{G}_{Z'} \subset \mathcal{G}$, where $Z' < Z$, with the label information $\mathbf{Y}_{Z'} \in \mathbb{R}^{Z' \times C}$, our goal is to classify the set of unseen graph labels of $\mathcal{G}_Z \setminus \mathcal{G}_{Z'}$.

**Datasets.** We consider eight datasets [72] for graph classification tasks, including five bioinformatics: MUTAG, PTC, NCI1, ENZYMES, and PROTEINS, and three social network datasets: IMDB-B, IMDB-M, and COLLAB. The detailed statistics of the graph classification benchmarks are summarized in Tab. 6. We use the random split from [94, 101, 67, 104], using 80% for training, 10% for validation, and 10% for testing. This process is repeated 10 times, and we report the average performance and standard deviation.

**Baselines.** For GCN, GAT, SAGE, ChebNet, ARMA, APPNP, and DiffPool, we use the implementations from the PyTorch Geometric library [28]. For other baselines, we use the implementations released by the respective authors.

**Hyperparameters Settings.** The dimension of node representations is set to 128 for all methods and datasets. We implement the proposed Pooling-NLSFs and Graph-level NLSFs using PyTorch and optimize the model with the Adam optimizer [48]. A readout function is applied to aggregate the node representations for each graph, utilizing mean, add, max, or RMS poolings. The learning rate and weight decay are searched within $\{1e^{-1}, 1e^{-2}, 1e^{-3}, 1e^{-4}, 1e^{-5}\}$, the pooling ratio within $[0.1, 0.9]$ with step $0.1$, the number of layers within $[1, 10]$ with step 1, the number of leading eigenvectors $J$ in Index NLSFs within $[1, 1e^2]$, the decay rate in the Value NLSFs using dyadic sampling within

Table 7: Semi-supervised node classification accuracy with random split, following the experimental protocol established by [41]. Results marked with * are taken from [41].

| | Cora | Citeseer | Pubmed | Chameleon | Squirrel | Actor |
|---|---|---|---|---|---|---|
| GCN | $79.19_{\pm1.4}$* | $69.71_{\pm1.3}$* | $78.81_{\pm0.8}$* | $38.15_{\pm3.8}$* | $31.18_{\pm1.0}$* | $22.74_{\pm2.3}$* |
| GAT | $80.03_{\pm0.8}$ | $68.16_{\pm0.9}$ | $77.26_{\pm0.5}$ | $34.16_{\pm1.2}$ | $27.40_{\pm1.4}$ | $24.35_{\pm1.7}$ |
| SAGE | $72.68_{\pm1.9}$ | $63.87_{\pm1.2}$ | $77.68_{\pm0.8}$ | $31.77_{\pm1.8}$ | $22.67_{\pm1.8}$ | $25.61_{\pm1.8}$ |
| ChebNet | $78.08_{\pm0.9}$* | $67.87_{\pm1.5}$* | $73.96_{\pm1.7}$* | $37.15_{\pm1.5}$* | $26.55_{\pm0.5}$* | $26.58_{\pm1.9}$* |
| ChebNetII | $\underline{82.42_{\pm0.6}}$* | $\underline{69.89_{\pm1.2}}$* | $79.53_{\pm1.0}$* | $\underline{43.42_{\pm3.5}}$* | $\underline{33.96_{\pm1.2}}$* | $\textbf{30.18}_{\pm0.8}$* |
| CayleyNet | $80.25_{\pm1.4}$ | $66.46_{\pm2.9}$ | $75.42_{\pm3.4}$ | $34.52_{\pm3.1}$ | $24.08_{\pm2.9}$ | $27.42_{\pm3.3}$ |
| APPNP | $82.39_{\pm0.7}$* | $69.79_{\pm0.9}$* | $\textbf{79.97}_{\pm1.6}$* | $32.73_{\pm2.3}$* | $24.50_{\pm0.9}$* | $29.74_{\pm1.0}$* |
| GPRGNN | $82.37_{\pm0.9}$* | $69.22_{\pm1.3}$* | $79.28_{\pm1.3}$* | $33.03_{\pm1.9}$* | $24.36_{\pm1.5}$* | $28.58_{\pm1.0}$* |
| ARMA | $79.14_{\pm1.1}$* | $69.35_{\pm1.4}$* | $78.31_{\pm1.3}$* | $37.42_{\pm1.7}$* | $24.15_{\pm0.9}$* | $27.02_{\pm2.3}$* |
| `att`-Node-level NLSFs | $\textbf{82.94}_{\pm1.1}$ | $\textbf{72.13}_{\pm1.1}$ | $\underline{79.62_{\pm1.2}}$ | $\textbf{46.75}_{\pm1.3}$ | $\textbf{35.17}_{\pm1.6}$ | $\underline{29.96_{\pm0.8}}$ |

$\{\frac{1}{4}, \frac{1}{3}, \frac{1}{2}, \frac{2}{3}, \frac{3}{4}\}$, the sampling resolution $S$ within $[1, 1e^2]$, and the number of the bands in Value NLSFs within $K \leq S$. The graph-level representation is then fed into an MLP with a (log)softmax classifier, using a cross-entropy loss function for predictions over the labels. Specifically, the MLP consists of three fully connected layers with 256, 128, and 64 neurons, respectively, followed by a (log)softmax classifier. We conduct a grid search on the hyperparameters for each dataset using Optuna [1]. An early stopping criterion is employed during training, stopping the process if the validation loss does not decrease for 100 consecutive epochs.

# F    Additional Experimental Results

Here, we present additional experiments on node and graph classification benchmarks, ablation studies, runtime analysis, and uniform sub-bands.

## F.1    Semi-Supervised Node Classification Following [41] Protocol

We present additional experimental results for semi-supervised node classification using random splits, adhering to the protocol established by [41]. The results, summarized in Tab. 7, demonstrate the classification accuracy across six benchmark datasets: Cora, Citeseer, Pubmed, Chameleon, Squirrel, and Actor. Our `att`-Node-level NLSFs achieve the highest accuracy on four out of the six datasets, outperforming other models significantly. On Pubmed, it records a close second-highest accuracy, slightly behind APPNP. Our method achieves a competitive second place in the Actor dataset. `att`-Node-level NLSFs demonstrate substantial improvements, particularly in challenging datasets like Chameleon and Squirrel. The comparison models, including GCN, GAT, SAGE, ChebNet, ChebNetII, CayleyNet, APPNP, GPRGNN, and ARMA, varied in their effectiveness, with ChebNetII and APPNP also showing strong results on several datasets. These findings highlight the efficacy of the `att`-Node-level NLSFs in semi-supervised node classification tasks.

## F.2    Node Classification on Filtered Chameleon and Squirrel Datasets in Dense Split Setting

Following [77], we conduct additional experiments on the original and filtered Chameleon and Squirrel datasets in the dense split setting. We use the same random splits as in [77], dividing the datasets into 48% for training, 32% for validation, and 20% for testing. We compare the Node-level NLSFs using Laplacian attention with GCN [49], SAGE [39], GAT [93], GT [88], $H_2$GCN [103], CPGNN [102], GPRGNN [16], FSGNN [69], GloGNN [62], FAGCN [8], GBKGNN [23], JacobiConv [96], and ResNet [40] with GNN models [77]. The study by [103] demonstrates the benefits of separating ego- and neighbor-embeddings in the GNN aggregation step when dealing with heterophily. Therefore, [77] also adopts this approach for the GNN aggregation step in GAT and GT models, denoted as "sep." The baseline results used for comparison are taken from [77]. Tab. 8 presents the full performance comparison on the original and filtered datasets. Note that Tab. 8 is the same as Tab. 2 but with more baseline methods. `att`-Node-level NLSFs achieve the highest accuracy on both the filtered Chameleon and Squirrel datasets. Additionally, `att`-Node-level NLSFs demonstrate strong performance on the original Chameleon dataset, achieving the highest

Table 8: Full Performance comparison of node classification on original and filtered Chameleon and Squirrel datasets in dense split setting. The baseline results used for comparison are taken from [77].

| | Chameleon | | Squirrel | |
| --- | --- | --- | --- | --- |
| | Original | Filtered | Original | Filtered |
| ResNet | $49.52_{\pm1.7}$ | $36.73_{\pm4.7}$ | $33.88_{\pm1.8}$ | $36.55_{\pm1.8}$ |
| ResNet+SGC | $49.93_{\pm2.3}$ | $41.01_{\pm4.5}$ | $34.36_{\pm1.2}$ | $38.36_{\pm2.0}$ |
| ResNet+adj | $71.07_{\pm2.2}$ | $38.67_{\pm3.9}$ | $65.46_{\pm1.6}$ | $38.37_{\pm2.0}$ |
| GCN | $50.18_{\pm3.3}$ | $40.89_{\pm4.1}$ | $39.06_{\pm1.5}$ | $39.47_{\pm1.5}$ |
| SAGE | $50.18_{\pm1.8}$ | $37.77_{\pm4.1}$ | $35.83_{\pm1.3}$ | $36.09_{\pm2.0}$ |
| GAT | $45.02_{\pm1.8}$ | $39.21_{\pm3.1}$ | $32.21_{\pm1.6}$ | $35.62_{\pm2.1}$ |
| GAT-sep | $50.24_{\pm2.2}$ | $39.26_{\pm2.5}$ | $35.72_{\pm2.0}$ | $35.46_{\pm3.1}$ |
| GT | $44.93_{\pm1.4}$ | $38.87_{\pm3.7}$ | $31.61_{\pm1.1}$ | $36.30_{\pm2.0}$ |
| GT-sep | $50.33_{\pm2.6}$ | $40.31_{\pm3.0}$ | $36.08_{\pm1.6}$ | $36.66_{\pm1.6}$ |
| H$_2$GCN | $46.27_{\pm2.7}$ | $26.75_{\pm3.6}$ | $29.45_{\pm1.7}$ | $35.10_{\pm1.2}$ |
| CPGNN | $48.77_{\pm2.1}$ | $33.00_{\pm3.2}$ | $30.91_{\pm2.0}$ | $30.04_{\pm2.0}$ |
| GPRGNN | $47.26_{\pm1.7}$ | $39.93_{\pm3.3}$ | $33.39_{\pm2.1}$ | $38.95_{\pm2.0}$ |
| FSGNN | $\underline{77.85}_{\pm0.5}$ | $40.61_{\pm3.0}$ | $\mathbf{68.93}_{\pm1.7}$ | $35.92_{\pm1.3}$ |
| GloGNN | $70.04_{\pm2.1}$ | $25.90_{\pm3.6}$ | $61.21_{\pm2.0}$ | $35.11_{\pm1.2}$ |
| FAGCN | $64.23_{\pm2.0}$ | $\underline{41.90}_{\pm2.7}$ | $47.63_{\pm1.9}$ | $\underline{41.08}_{\pm2.3}$ |
| GBKGNN | $51.36_{\pm1.8}$ | $39.61_{\pm2.6}$ | $37.06_{\pm1.2}$ | $35.51_{\pm1.7}$ |
| JacobiConv | $68.33_{\pm1.4}$ | $39.00_{\pm4.2}$ | $46.17_{\pm4.3}$ | $29.71_{\pm1.7}$ |
| att-Node-level NLSFs | $\mathbf{79.42}_{\pm1.6}$ | $\mathbf{42.06}_{\pm1.3}$ | $67.81_{\pm1.4}$ | $\mathbf{42.18}_{\pm1.2}$ |

Table 9: An ablation study investigated the effect of Node-level NLSFs on node classification, comparing the use of Index NLSFs and Value NLSFs. The symbol $(\uparrow)$ denotes an improvement using Laplacian attention.

| | Cora | | Citeseer | | Pubmed | | Chameleon | | Squirrel | | Actor | |
| --- | --- | --- | --- | --- | --- | --- | --- | --- | --- | --- | --- | --- |
| $\Theta_{\mathrm{ind}}(\mathbf{L},\cdot)$ | $82.46_{\pm1.2}$ | | $71.31_{\pm1.0}$ | | $\underline{80.97}_{\pm0.9}$ | | $44.37_{\pm1.8}$ | | $34.12_{\pm1.4}$ | | $29.88_{\pm1.1}$ | |
| $\Theta_{\mathrm{ind}}(\mathbf{N},\cdot)$ | $81.73_{\pm1.4}$ | | $70.25_{\pm0.8}$ | | $79.88_{\pm1.2}$ | | $44.52_{\pm1.7}$ | | $34.26_{\pm1.5}$ | | $29.97_{\pm1.7}$ | |
| $\Theta_{\mathrm{val}}(\mathbf{L},\cdot)$ | $80.25_{\pm1.5}$ | | $70.43_{\pm1.7}$ | | $80.06_{\pm1.6}$ | | $45.52_{\pm1.2}$ | | $35.23_{\pm0.8}$ | | $32.69_{\pm1.9}$ | |
| $\Theta_{\mathrm{val}}(\mathbf{N},\cdot)$ | $81.98_{\pm0.7}$ | | $71.16_{\pm1.3}$ | | $80.33_{\pm1.7}$ | | $\underline{47.91}_{\pm1.4}$ | | $35.78_{\pm0.8}$ | | $\underline{33.53}_{\pm1.4}$ | |
| $\mathrm{att}(\Theta_{\mathrm{ind}}(\mathbf{L},\cdot),\Theta_{\mathrm{val}}(\mathbf{L},\cdot))$ | $82.46_{\pm1.2}$ | | $71.48_{\pm0.8}$ | $(\uparrow)$ | $80.97_{\pm0.9}$ | | $46.71_{\pm2.2}$ | $(\uparrow)$ | $36.92_{\pm1.1}$ | $(\uparrow)$ | $32.69_{\pm1.9}$ | |
| $\mathrm{att}(\Theta_{\mathrm{ind}}(\mathbf{N},\cdot),\Theta_{\mathrm{val}}(\mathbf{N},\cdot))$ | $81.98_{\pm0.7}$ | | $\underline{72.45}_{\pm1.4}$ | $(\uparrow)$ | $80.33_{\pm1.7}$ | | $47.91_{\pm1.4}$ | | $36.87_{\pm0.7}$ | $(\uparrow)$ | $33.53_{\pm1.4}$ | |
| $\mathrm{att}(\Theta_{\mathrm{ind}}(\mathbf{N},\cdot),\Theta_{\mathrm{val}}(\mathbf{L},\cdot))$ | $\underline{82.65}_{\pm1.2}$ | $(\uparrow)$ | $71.26_{\pm1.7}$ | $(\uparrow)$ | $80.56_{\pm1.7}$ | $(\uparrow)$ | $45.98_{\pm2.3}$ | $(\uparrow)$ | $\underline{37.03}_{\pm1.9}$ | $(\uparrow)$ | $33.09_{\pm1.2}$ | $(\uparrow)$ |
| $\mathrm{att}(\Theta_{\mathrm{ind}}(\mathbf{L},\cdot),\Theta_{\mathrm{val}}(\mathbf{N},\cdot))$ | $\mathbf{84.75}_{\pm0.7}$ | $(\uparrow)$ | $\mathbf{73.62}_{\pm1.1}$ | $(\uparrow)$ | $\mathbf{81.93}_{\pm1.0}$ | $(\uparrow)$ | $\mathbf{49.68}_{\pm1.6}$ | $(\uparrow)$ | $\mathbf{38.25}_{\pm0.7}$ | $(\uparrow)$ | $\mathbf{34.72}_{\pm0.9}$ | $(\uparrow)$ |

accuracy and the second-highest accuracy on the original Squirrel dataset. `att`-Node-level NLSFs show less sensitivity to node duplicates and exhibit stronger generalization ability, further validating the reliability of the Chameleon and Squirrel datasets in the dense split setting.

### F.3 Ablation Study on `att`-Node-level NLSFs, `att`-Graph-NLSF, and `att`-Pooling-NLSFs

In Sec. 3.5, we note that using the graph Laplacian $\mathbf{L}$ in Index NLSFs and the normalized graph Laplacian $\mathbf{N}$ in Value NLSFs is transferable. Since real-world graphs often fall between these two boundary cases, we present the Laplacian attention NLSFs that operate between them at both the node-level and graph-level. Indeed, as demonstrated in Sec. 5, the proposed `att`-Node-level NLSFs, `att`-Graph-level NLSFs, and `att`-Pooling-NLSFs outperform existing spectral GNNs.

We conduct an ablation study to evaluate the contribution and effectiveness of different components within the `att`-Node-level NLSFs, `att`-Graph-level-NLSFs, and `att`-Pooling-NLSFs on node and graph classification tasks. Specifically, we compare the Index NLSFs and Value NLSFs using both the graph Laplacian $\mathbf{L}$ and the normalized graph Laplacian $\mathbf{N}$ to understand their individual and Laplacian attention impact on these tasks.

The ablation study of `att`-Node-level NLSPs for node classification is summarized in Tab. 9. We investigate the performance on six node classification benchmarks as in Sec. 5, including Cora, Citeseer, Pubmed, Chameleon, Squirrel, and Actor. Tab. 9 shows that using the graph Laplacian $\mathbf{L}$ in Index NLSFs (denoted as $\Theta_{\mathrm{ind}}(\mathbf{L},\cdot)$) and the normalized graph Laplacian $\mathbf{N}$ in Value NLSFs (denoted as $\Theta_{\mathrm{val}}(\mathbf{N},\cdot)$ ) has superior node classification accuracy compared to using the normalized graph Laplacian $\mathbf{N}$ in Index NLSFs (denoted as $\Theta_{\mathrm{ind}}(\mathbf{N},\cdot)$ ) and the graph Laplacian $\mathbf{L}$ in Value

Table 10: An ablation study investigated the effect of Graph-NLSFs on graph classification, comparing the use of Index NLSFs and Value NLSFs. The symbol (↑) denotes an improvement using Laplacian attention.

| | MUTAG | PTC | ENZYMES | PROTEINS | NCI1 | IMDB-B | IMDB-M | COLLAB |
|---|---|---|---|---|---|---|---|---|
| $\Phi_{\text{ind}}(\mathbf{L},\cdot)$ | $82.14_{\pm1.2}$ | $65.73_{\pm2.4}$ | $62.31_{\pm1.4}$ | $78.22_{\pm1.6}$ | $\mathbf{80.51}_{\pm1.2}$ | $63.27_{\pm1.1}$ | $50.29_{\pm1.4}$ | $70.06_{\pm1.3}$ |
| $\Phi_{\text{ind}}(\mathbf{N},\cdot)$ | $83.27_{\pm0.3}$ | $66.08_{\pm2.1}$ | $60.18_{\pm0.5}$ | $80.31_{\pm1.4}$ | $78.40_{\pm0.6}$ | $64.58_{\pm2.2}$ | $\underline{52.33}_{\pm0.9}$ | $71.88_{\pm1.7}$ |
| $\Phi_{\text{val}}(\mathbf{L},\cdot)$ | $\underline{83.40}_{\pm1.4}$ | $65.67_{\pm1.4}$ | $\mathbf{66.42}_{\pm1.1}$ | $\mathbf{83.36}_{\pm1.4}$ | $76.17_{\pm1.2}$ | $\underline{72.71}_{\pm1.4}$ | $52.13_{\pm0.9}$ | $74.19_{\pm1.0}$ |
| $\Phi_{\text{val}}(\mathbf{N},\cdot)$ | $81.19_{\pm0.3}$ | $66.20_{\pm0.7}$ | $63.32_{\pm1.2}$ | $78.20_{\pm0.9}$ | $78.68_{\pm1.2}$ | $\mathbf{74.26}_{\pm1.8}$ | $\mathbf{52.49}_{\pm0.7}$ | $\mathbf{79.06}_{\pm1.2}$ |
| $\text{att}(\Phi_{\text{ind}}(\mathbf{L},\cdot),\Phi_{\text{val}}(\mathbf{L},\cdot))$ | $\underline{83.40}_{\pm1.4}$ | $66.32_{\pm1.9}$ (↑) | $\mathbf{66.42}_{\pm1.1}$ | $\mathbf{83.36}_{\pm1.4}$ | $\mathbf{80.51}_{\pm1.2}$ | $72.71_{\pm1.4}$ | $52.13_{\pm0.9}$ | $75.14_{\pm1.8}$ (↑) |
| $\text{att}(\Phi_{\text{ind}}(\mathbf{N},\cdot),\Phi_{\text{val}}(\mathbf{N},\cdot))$ | $83.71_{\pm1.7}$ (↑) | $67.14_{\pm1.5}$ (↑) | $65.97_{\pm1.1}$ (↑) | $81.79_{\pm1.2}$ (↑) | $79.83_{\pm0.6}$ (↑) | $\mathbf{74.26}_{\pm1.8}$ | $\mathbf{52.49}_{\pm0.7}$ | $\mathbf{79.06}_{\pm1.2}$ |
| $\text{att}(\Phi_{\text{ind}}(\mathbf{N},\cdot),\Phi_{\text{val}}(\mathbf{L},\cdot))$ | $83.78_{\pm0.8}$ (↑) | $66.20_{\pm0.7}$ | $66.42_{\pm1.1}$ | $83.36_{\pm1.4}$ | $78.40_{\pm0.6}$ | $72.71_{\pm1.4}$ | $52.33_{\pm0.9}$ | $76.22_{\pm0.8}$ (↑) |
| $\text{att}(\Phi_{\text{ind}}(\mathbf{L},\cdot),\Phi_{\text{val}}(\mathbf{N},\cdot))$ | $\mathbf{84.13}_{\pm1.5}$ (↑) | $\mathbf{68.17}_{\pm1.0}$ (↑) | $65.94_{\pm1.6}$ (↑) | $\underline{82.69}_{\pm1.9}$ (↑) | $\mathbf{80.51}_{\pm1.2}$ | $\mathbf{74.26}_{\pm1.8}$ | $\mathbf{52.49}_{\pm0.7}$ | $\mathbf{79.06}_{\pm1.2}$ |

Table 11: An ablation study investigated the effect of Pooling-NLSFs on graph classification, comparing the use of Index NLSFs and Value NLSFs. The symbol (↑) denotes an improvement using Laplacian attention.

| | MUTAG | PTC | ENZYMES | PROTEINS | NCI1 | IMDB-B | IMDB-M | COLLAB |
|---|---|---|---|---|---|---|---|---|
| $\Theta^{\text{P}}_{\text{ind}}(\mathbf{L},\cdot)$ | $86.41_{\pm1.9}$ | $68.76_{\pm0.9}$ | $69.88_{\pm1.3}$ | $84.27_{\pm1.3}$ | $80.33_{\pm0.8}$ | $60.40_{\pm1.3}$ | $51.01_{\pm1.3}$ | $71.28_{\pm0.9}$ |
| $\Theta^{\text{P}}_{\text{ind}}(\mathbf{N},\cdot)$ | $84.52_{\pm0.8}$ | $67.19_{\pm1.2}$ | $66.37_{\pm2.1}$ | $81.12_{\pm1.7}$ | $77.05_{\pm2.2}$ | $62.08_{\pm1.6}$ | $52.48_{\pm1.0}$ | $72.03_{\pm1.2}$ |
| $\Theta^{\text{P}}_{\text{val}}(\mathbf{L},\cdot)$ | $83.64_{\pm0.9}$ | $67.74_{\pm1.3}$ | $65.87_{\pm1.4}$ | $84.18_{\pm1.4}$ | $79.43_{\pm1.4}$ | $71.98_{\pm2.0}$ | $51.85_{\pm1.7}$ | $78.80_{\pm1.1}$ |
| $\Theta^{\text{P}}_{\text{val}}(\mathbf{N},\cdot)$ | $86.58_{\pm1.0}$ | $69.13_{\pm1.1}$ | $68.89_{\pm0.4}$ | $84.80_{\pm1.0}$ | $80.82_{\pm0.8}$ | $76.48_{\pm1.8}$ | $54.12_{\pm1.0}$ | $81.31_{\pm0.7}$ |
| $\text{att}^{\text{P}}(\Theta_{\text{ind}}(\mathbf{L},\cdot),\Theta_{\text{val}}(\mathbf{L},\cdot))$ | $86.41_{\pm1.9}$ | $68.76_{\pm0.9}$ | $69.88_{\pm1.3}$ | $84.27_{\pm1.3}$ | $80.33_{\pm0.8}$ | $72.44_{\pm1.3}$ (↑) | $51.85_{\pm1.7}$ | $78.80_{\pm1.1}$ |
| $\text{att}^{\text{P}}(\Theta_{\text{ind}}(\mathbf{N},\cdot),\Theta_{\text{val}}(\mathbf{N},\cdot))$ | $86.58_{\pm1.0}$ | $69.67_{\pm1.2}$ (↑) | $69.06_{\pm1.1}$ (↑) | $84.80_{\pm1.0}$ | $80.82_{\pm0.8}$ | $76.48_{\pm1.8}$ | $54.37_{\pm1.2}$ (↑) | $81.70_{\pm0.9}$ (↑) |
| $\text{att}^{\text{P}}(\Theta_{\text{ind}}(\mathbf{N},\cdot),\Theta_{\text{val}}(\mathbf{L},\cdot))$ | $84.93_{\pm1.7}$ (↑) | $68.14_{\pm1.6}$ (↑) | $67.26_{\pm1.0}$ (↑) | $84.18_{\pm1.4}$ | $79.43_{\pm1.4}$ | $72.64_{\pm1.2}$ (↑) | $52.48_{\pm1.0}$ | $78.80_{\pm1.1}$ |
| $\text{att}^{\text{P}}(\Theta_{\text{ind}}(\mathbf{L},\cdot),\Theta_{\text{val}}(\mathbf{N},\cdot))$ | $\mathbf{86.89}_{\pm1.2}$ (↑) | $\mathbf{71.02}_{\pm1.3}$ (↑) | $\mathbf{69.94}_{\pm1.0}$ (↑) | $\mathbf{84.89}_{\pm0.9}$ (↑) | $\mathbf{80.95}_{\pm1.4}$ (↑) | $\mathbf{76.78}_{\pm1.9}$ (↑) | $\mathbf{55.28}_{\pm1.7}$ (↑) | $\mathbf{82.19}_{\pm1.3}$ (↑) |

NLSFs (denoted as $\Theta_{\text{val}}(\mathbf{L},\cdot)$). This is in line with our theoretical findings in App. D.5. Moreover, the att-Node-level NLSFs using the Laplacian attention $\text{att}(\Theta_{\text{ind}}(\mathbf{L},\cdot),\Theta_{\text{val}}(\mathbf{N},\cdot))$ yield the highest accuracies across all datasets, corroborating the findings in Sec. 3.5. We also note that without Laplacian attention, the Node-level NLSFs ($\Theta_{\text{ind}}(\mathbf{L},\cdot)$ and $\Theta_{\text{val}}(\mathbf{N},\cdot)$) alone still achieve more effective classification performance compared to existing baselines, as shown in Tab. 1.

Tab. 10 demonstrates the ablation study of Graph-level NLSFs for graph classification tasks. We examine the eight graph datasets as in Sec. 5, including MUTAG, PTC, ENZYMES, PROTEINS, NCI1, IMDB-B, IMDB-M, and COLLAB. Similar to the above, we investigate the Index and Value settings using graph Laplacian $\mathbf{L}$ and normalized graph Laplacian $\mathbf{N}$, including $\Phi_{\text{ind}}(\mathbf{L},\cdot)$, $\Phi_{\text{ind}}(\mathbf{N},\cdot)$, $\Phi_{\text{val}}(\mathbf{L},\cdot)$, and $\Phi_{\text{val}}(\mathbf{N},\cdot)$, along with their variants using Laplacian attention. Here, we see that att-Graph-level NLSFs do not show significant improvement over the standard Graph-level NLSFs. Notably, $\Phi_{\text{val}}(\mathbf{N},\cdot)$ outperforms other models in social network datasets (IMDB-B, IMDB-M, and COLLAB), where the node features are augmented by the node degree. We plan to investigate the limited graph attribution in future work.

The ablation study of att-Pooling-NLSFs for graph classification is reported in Tab. 11. Similar to Tab. 10, we consider eight graph classification benchmarks: MUTAG, PTC, ENZYMES, PROTEINS, NCI1, IMDB-B, IMDB-M, and COLLAB. Tabl. 11 demonstrates that using the graph Laplacian $\mathbf{L}$ in Index Pooling-NLSFs (denoted as $\Theta^{\text{P}}_{\text{ind}}(\mathbf{L},\cdot)$) and the normalized graph Laplacian $\mathbf{N}$ in Value Pooling-NLSFs (denoted as $\Theta^{\text{P}}_{\text{val}}(\mathbf{N},\cdot)$) achieves superior graph classification accuracy compared to using the normalized graph Laplacian $\mathbf{N}$ in Index Pooling-NLSFs (denoted as $\Theta^{\text{P}}_{\text{ind}}(\mathbf{N},\cdot)$) and the graph Laplacian $\mathbf{L}$ in Value Pooling-NLSFs (denoted as $\Theta^{\text{P}}_{\text{val}}(\mathbf{L},\cdot)$). This finding aligns with our theoretical results in App. D.5. Moreover, the att-Pooling-NLSFs using the Laplacian attention $\text{att}^{\text{P}}(\Theta_{\text{ind}}(\mathbf{L},\cdot),\Theta_{\text{val}}(\mathbf{N},\cdot))$ yield the highest accuracies across all datasets, corroborating the findings in Sec. 3.5. In addition, att-Pooling-NLSFs consistently outperform att-Graph-level NLSFs as shown in Tab. 10, indicating that the node features learned in our Node-level NLSFs representation are more expressive. This supports our theoretical findings in Sec. 4.3.

The ablation study demonstrates that the Laplacian attention between the Index and Value NLSFs significantly enhances classification accuracy for both node and graph classification tasks across various datasets, outperforming existing baselines.

Table 12: Average running time per epoch(ms)/average total running time(s).

|  | Cora | Citeseer | Pubmed | Chameleon | Squirrel | Actor |
|---|---|---|---|---|---|---|
| GCN | 8.97/2.1 | 9.1/2.3 | 12.15/3.9 | 11.68/2.7 | 25.52/6.2 | 14.51/3.8 |
| GAT | 13.13/3.3 | 13.87/3.6 | 19.42/6.2 | 14.83/3.4 | 46.13/15.9 | 20.13/4.4 |
| SAGE | 11.72/2.2 | 12.11/2.4 | 25.31/6.1 | 60.61/12.7 | 321.65/72.8 | 25.16/5.4 |
| ChebNet | 21.36/4.9 | 22.51/5.3 | 34.53/13.1 | 42.21/16.0 | 38.21/45.1 | 42.91/9.3 |
| ChebNetII | 20.53/5.9 | 20.61/5.7 | 33.57/12.9 | 39.03/17.3 | 37.29/38.04 | 40.05/9.1 |
| CayleyNet | 401.24/79.3 | 421.69/83.4 | 723.61/252.3 | 848.51/389.4 | 972.53/361.8 | 794.61/289.4 |
| APPNP | 18.31/4.2 | 19.17/4.8 | 19.63/5.9 | 18.56/3.8 | 24.18/4.9 | 15.93/4.6 |
| GPRGNN | 19.07/3.8 | 18.69/4.0 | 19.77/6.3 | 19.31/3.6 | 28.31/5.5 | 17.28/4.8 |
| ARMA | 20.91/5.2 | 19.33/4.9 | 34.27/14.5 | 41.63/19.7 | 39.42/42.7 | 46.22/5.7 |
| att-Node-level NLSFs | 18.22/4.5 | 18.51/4.4 | 20.23/6.1 | 28.51/17.1 | 25.56/5.1 | 17.09/4.6 |

Table 13: Experimental results on large heterophilic graphs. The results for BernNet, ChebNet, ChebNetII, and GPRGNN are taken from [41], while the results for OptBasisGNN are taken from [38]. All other competing results are taken from [63].

|  | Penn94 | Pokec | Genius | Twitch-Gamers | Wiki |
|---|---|---|---|---|---|
| GCN | $82.47_{\pm0.3}$ | $75.45_{\pm0.2}$ | $87.42_{\pm0.4}$ | $62.18_{\pm0.3}$ | OOM |
| LINK | $80.79_{\pm0.5}$ | $80.54_{\pm0.0}$ | $73.56_{\pm0.1}$ | $64.85_{\pm0.2}$ | $57.11_{\pm0.3}$ |
| LINKX | $84.71_{\pm0.5}$ | $82.04_{\pm0.1}$ | $90.77_{\pm0.3}$ | $\mathbf{66.06}_{\pm0.2}$ | $59.80_{\pm0.4}$ |
| GPRGNN | $83.54_{\pm0.3}$ | $80.74_{\pm0.2}$ | $90.15_{\pm0.3}$ | $62.59_{\pm0.4}$ | $58.73_{\pm0.3}$ |
| ChebNet | $82.59_{\pm0.3}$ | $72.71_{\pm0.7}$ | $89.36_{\pm0.3}$ | $62.31_{\pm0.4}$ | OOM |
| ChebNetII | $\underline{84.86}_{\pm0.3}$ | $82.33_{\pm0.3}$ | $90.85_{\pm0.3}$ | $65.03_{\pm0.3}$ | $60.95_{\pm0.4}$ |
| BernNet | $83.26_{\pm0.3}$ | $81.67_{\pm0.2}$ | $90.47_{\pm0.3}$ | $64.27_{\pm0.3}$ | $59.02_{\pm0.3}$ |
| OptBasisGNN | $84.85_{\pm0.4}$ | $\underline{82.83}_{\pm0.0}$ | $\underline{90.83}_{\pm0.1}$ | $65.17_{\pm0.2}$ | $\underline{61.85}_{\pm0.0}$ |
| att-Node-level NLSFs | $\mathbf{85.19}_{\pm0.3}$ | $\mathbf{82.96}_{\pm0.1}$ | $\mathbf{91.24}_{\pm0.1}$ | $\underline{65.97}_{\pm0.2}$ | $\mathbf{62.44}_{\pm0.3}$ |

## F.4 Runtime Analysis

In our NLSFs, the eigendecomposition can be calculated once for each graph and reused in the training process. This step is essential as the cost of the forward pass during model training often surpasses the initial eigendecomposition preprocessing cost. Note that the computation time for eigendecomposition is considerably less than the time needed for model training. For medium and large graphs, the computation time is further reduced when partial eigendecomposition is utilized, making it more efficient than the training times of competing baselines. To evaluate the computational complexity of our NLSFs compared to baseline models, we report the empirical training time in Tab. 12. Our Node-level NLSFs showcase competitive running times, with moderate values per epoch and total running times that are comparable to the most efficient models like GCN, GPRGNN, and APPNP. Notably, Node-level NLSFs are particularly efficient for the Cora, Citeseer, and Pubmed datasets. For the dense Squirrel graph, our Node-level NLSFs exhibit efficient performance with a moderate running time, outperforming several models that struggle with significantly higher times.

## F.5 Scalability to Large-Scale Datasets

To demonstrate the scalability of our method, we conduct additional tests on five large heterophilic graphs: Penn94, Pokec, Genius, Twitch-Gamers, and Wiki datasets from [63]. The experimental setup is in line with previous work by [16, 63, 41]. We use the same hyperparameters for our NLSFs as reported in App. E. Tab. 13 presents the classification accuracy. We see that NLSFs outperform the competing methods on the Penn94, Pokec, Genius, and Wiki datasets. For the Twitch-Gamers dataset, NLSFs yield the second-best results. Our additional experiments show that our method could indeed scale to handle large-scale graphs effectively.

## F.6 Index-by-Index Index-NLSFs and Band-by-Band Value-NLSFs

Our primary objective in this work is to introduce new GNNs that are equivariant to functional symmetries, based on a novel spectral domain that is transferable between graphs. We emphasize

Table 14: Graph classification performance using Diagonal NLSFs, including index-by-index Index-NLSFs and band-by-band Value-NLSFs, along with their variants using Laplacian attention. The symbol (↑) denotes an improvement using Laplacian attention.

| | MUTAG | | PTC | | ENZYMES | | PROTEINS | | NCI1 | IMDB-B | IMDB-M | | COLLAB | |
|---|---|---|---|---|---|---|---|---|---|---|---|---|---|---|
| $\Gamma^P_{ind}(\mathbf{L},\cdot)$ | $82.33_{\pm1.7}$ | | $66.43_{\pm1.8}$ | | $\mathbf{69.69}_{\pm2.4}$ | | $\mathbf{83.47}_{\pm2.2}$ | | $77.43_{\pm1.4}$ | $60.17_{\pm0.8}$ | $50.04_{\pm1.2}$ | | $70.89_{\pm0.9}$ | |
| $\Gamma^P_{ind}(\mathbf{N},\cdot)$ | $82.85_{\pm0.8}$ | | $63.40_{\pm1.7}$ | | $68.06_{\pm2.1}$ | | $82.69_{\pm0.8}$ | | $78.00_{\pm1.4}$ | $62.27_{\pm1.8}$ | $51.74_{\pm1.4}$ | | $70.69_{\pm1.3}$ | |
| $\Gamma^P_{val}(\mathbf{L},\cdot)$ | $82.77_{\pm0.8}$ | | $63.20_{\pm0.9}$ | | $64.19_{\pm2.3}$ | | $\underline{83.39}_{\pm1.7}$ | | $78.26_{\pm1.4}$ | $70.61_{\pm2.2}$ | $51.86_{\pm1.9}$ | | $77.03_{\pm1.5}$ | |
| $\Gamma^P_{val}(\mathbf{N},\cdot)$ | $80.29_{\pm0.8}$ | | $65.02_{\pm1.4}$ | | $65.29_{\pm1.7}$ | | $81.62_{\pm1.2}$ | | $\underline{78.28}_{\pm1.0}$ | $\mathbf{74.33}_{\pm1.7}$ | $52.89_{\pm0.9}$ | | $\underline{80.41}_{\pm0.8}$ | |
| $\mathtt{att}^P(\Gamma_{ind}(\mathbf{L},\cdot),\Gamma_{val}(\mathbf{L},\cdot))$ | $83.19_{\pm1.4}$ | (↑) | $66.43_{\pm1.8}$ | | $\mathbf{69.69}_{\pm2.4}$ | | $\mathbf{83.47}_{\pm2.2}$ | | $78.26_{\pm1.4}$ | $70.61_{\pm2.2}$ | $52.03_{\pm1.4}$ | (↑) | $77.56_{\pm1.3}$ | (↑) |
| $\mathtt{att}^P(\Gamma_{ind}(\mathbf{N},\cdot),\Gamma_{val}(\mathbf{N},\cdot))$ | $83.34_{\pm1.2}$ | (↑) | $66.58_{\pm1.2}$ | (↑) | $68.26_{\pm1.9}$ | (↑) | $82.99_{\pm1.8}$ | (↑) | $\mathbf{78.28}_{\pm1.0}$ | $\mathbf{74.33}_{\pm1.7}$ | $53.14_{\pm0.8}$ | (↑) | $80.41_{\pm0.8}$ | |
| $\mathtt{att}^P(\Gamma_{ind}(\mathbf{N},\cdot),\Gamma_{val}(\mathbf{L},\cdot))$ | $\underline{83.42}_{\pm1.5}$ | (↑) | $64.08_{\pm1.7}$ | (↑) | $68.06_{\pm2.1}$ | | $83.39_{\pm1.7}$ | | $78.26_{\pm1.4}$ | $70.61_{\pm2.2}$ | $52.13_{\pm1.4}$ | (↑) | $78.12_{\pm1.9}$ | (↑) |
| $\mathtt{att}^P(\Gamma_{ind}(\mathbf{L},\cdot),\Gamma_{val}(\mathbf{N},\cdot))$ | $\mathbf{83.85}_{\pm1.4}$ | (↑) | $\mathbf{67.12}_{\pm1.6}$ | (↑) | $\mathbf{69.69}_{\pm2.4}$ | | $\mathbf{83.47}_{\pm2.2}$ | | $\mathbf{78.28}_{\pm1.0}$ | $\mathbf{74.33}_{\pm1.7}$ | $\mathbf{53.91}_{\pm1.6}$ | (↑) | $\mathbf{81.03}_{\pm1.2}$ | (↑) |

Table 15: Semi-supervised node classification accuracy using NLSFs, diag-NLSFs, lead-NLSFs, and lead-diag-NLSFs.

| | Cora | Citeseer | Pubmed | Chameleon | Squirrel | Actor |
|---|---|---|---|---|---|---|
| att-Node-level diag-NLSFs | $85.37_{\pm1.8}$ | $75.41_{\pm0.8}$ | $82.22_{\pm1.2}$ | $50.58_{\pm1.3}$ | $38.39_{\pm0.9}$ | $35.13_{\pm1.0}$ |
| att-Node-level NLSFs | $86.03_{\pm1.2}$ | $\underline{74.87}_{\pm1.2}$ | $83.15_{\pm1.5}$ | $50.12_{\pm1.2}$ | $36.23_{\pm1.6}$ | $35.01_{\pm1.7}$ |
| att-Node-level lead-NLSFs | $85.26_{\pm0.4}$ | $74.16_{\pm1.4}$ | $82.24_{\pm0.9}$ | $49.86_{\pm1.9}$ | $34.21_{\pm1.1}$ | $33.68_{\pm1.1}$ |
| att-Node-level lead-diag-NLSFs | $84.75_{\pm0.7}$ | $73.62_{\pm1.1}$ | $81.93_{\pm1.0}$ | $49.68_{\pm1.6}$ | $38.25_{\pm0.7}$ | $34.72_{\pm0.9}$ |

the unique aspects of our method rather than providing an exhaustive list of operators in this group, which, while important, is a key direction for future research.

Here, we present a new type of NLSFs: index-by-index Index-NLSFs and band-by-band Value-NLSFs. We denote them as follows:

$$\Gamma_{ind}(\mathbf{\Delta},\mathbf{X}) = \sum_{j=1}^{J} \gamma_j \left( \|\mathbf{P}_j\mathbf{X}\|_{sig} \right) \frac{\mathbf{P}_j\mathbf{X}}{\|\mathbf{P}_j\mathbf{X}\|^a_{sig} + e} \text{ and}$$

$$\Gamma_{val}(\mathbf{\Delta},\mathbf{X}) = \sum_{j=1}^{K} \gamma_j \left( \|g_j(\mathbf{\Delta})\mathbf{X}\|_{sig} \right) \frac{g_j(\mathbf{L})\mathbf{X}}{\|g_p(\mathbf{\Delta})\mathbf{X}\|^a_{sig} + e},$$

where $a, e$ are as before in Sec. 3.3. Note that these operators are also equivariant to our group actions.

We investigate index-by-index Index-NLSFs and band-by-band Value-NLSFs in graph classification tasks as described in Sec. 5, including MUTAG, PTC, ENZYMES, PROTEINS, NCI1, IMDB-B, IMDB-M, and COLLAB datasets. Unlike Graph-NLSFs, which are fully spectral and map a sequence of frequency coefficients to an output vector, index-by-index Index-NLSFs and band-by-band Value-NLSFs do not possess such a sequential spectral form. In index-by-index Index-NLSFs and band-by-band Value-NLSFs, the index-by-index (or band-by-band) frequency response is projected back to the graph domain. Consequently, for graph classification tasks, we apply the readout function as defined for Pooling-NLSFs in Sec. 3.4.

Following App. F.3, we examine index-by-index Index-NLSFs and band-by-band Value-NLSFs settings using graph Laplacian $\mathbf{L}$ and normalized graph Laplacian $\mathbf{N}$, including $\Gamma^P_{ind}(\mathbf{L},\cdot)$, $\Gamma^P_{ind}(\mathbf{N},\cdot)$, $\Gamma^P_{val}(\mathbf{L},\cdot)$, and $\Gamma^P_{val}(\mathbf{N},\cdot)$, along with their variants using Laplacian attention, where P denotes the pooling function as in Tab. 11.

Tab. 14 presents the graph classification accuracy using index-by-index Index-NLSFs and band-by-band Value-NLSFs. It shows that incorporating Laplacian attention consistently improves classification performance, aligning with the findings in App. F.3. We note that index-by-index Index-NLSFs and band-by-band Value-NLSFs perform comparably to existing baselines in graph classification benchmarks. However, index-by-index Index-NLSFs and band-by-band Value-NLSFs are generally less effective compared to Pooling-NLSFs, as shown in Tab. 11.

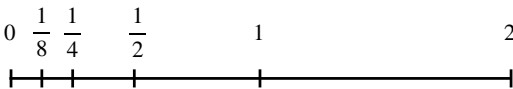

$$0 \quad \frac{1}{8} \quad \frac{1}{4} \qquad \frac{1}{2} \qquad\qquad 1 \qquad\qquad\qquad\qquad 2$$

Figure 5: Illustration of dyadic sub-bands for $r = \frac{1}{2}$ and $S = 4$.

Table 16: Graph classification performance using Graph-NLSFs with uniform sub-bands. The symbol (↑) denotes an improvement using Laplacian attention.

| | MUTAG | | PTC | | ENZYMES | | PROTEINS | | NCI1 | | IMDB-B | | IMDB-M | | COLLAB | |
|---|---|---|---|---|---|---|---|---|---|---|---|---|---|---|---|---|
| $\Phi_{\mathrm{ind}}(\mathbf{L},\cdot)$ | $82.14_{\pm1.2}$ | | $65.73_{\pm2.4}$ | | $62.31_{\pm1.4}$ | | $78.22_{\pm1.6}$ | | $\mathbf{80.51}_{\pm1.2}$ | | $63.27_{\pm1.1}$ | | $50.29_{\pm1.4}$ | | $70.06_{\pm1.3}$ | |
| $\Phi_{\mathrm{ind}}(\mathbf{N},\cdot)$ | $83.27_{\pm0.3}$ | | $66.08_{\pm2.1}$ | | $60.18_{\pm0.5}$ | | $\underline{80.31}_{\pm1.4}$ | | $78.40_{\pm0.6}$ | | $64.58_{\pm2.2}$ | | $\mathbf{52.33}_{\pm0.9}$ | | $71.88_{\pm1.7}$ | |
| $\Phi_{\mathrm{val}}(\mathbf{L},\cdot)$ | $81.73_{\pm1.4}$ | | $64.81_{\pm1.4}$ | | $\underline{64.46}_{\pm1.2}$ | | $78.31_{\pm0.9}$ | | $78.22_{\pm1.4}$ | | $70.84_{\pm2.3}$ | | $50.42_{\pm1.8}$ | | $73.49_{\pm1.4}$ | |
| $\Phi_{\mathrm{val}}(\mathbf{N},\cdot)$ | $81.12_{\pm0.6}$ | | $65.13_{\pm0.7}$ | | $60.06_{\pm1.3}$ | | $78.88_{\pm0.4}$ | | $78.48_{\pm1.3}$ | | $\mathbf{74.05}_{\pm2.4}$ | | $51.13_{\pm0.8}$ | | $\mathbf{78.16}_{\pm1.3}$ | |
| $\mathrm{att}(\Phi_{\mathrm{ind}}(\mathbf{L},\cdot),\Phi_{\mathrm{val}}(\mathbf{L},\cdot))$ | $82.76_{\pm0.9}$ | (↑) | $65.73_{\pm2.4}$ | | $\mathbf{64.58}_{\pm1.7}$ | (↑) | $79.23_{\pm1.2}$ | (↑) | $\mathbf{80.51}_{\pm1.2}$ | | $70.84_{\pm2.3}$ | | $51.04_{\pm2.2}$ | (↑) | $74.21_{\pm1.2}$ | (↑) |
| $\mathrm{att}(\Phi_{\mathrm{ind}}(\mathbf{N},\cdot),\Phi_{\mathrm{val}}(\mathbf{N},\cdot))$ | $83.27_{\pm0.3}$ | | $\mathbf{66.08}_{\pm2.1}$ | | $61.25_{\pm1.8}$ | (↑) | $\mathbf{81.07}_{\pm1.7}$ | (↑) | $\underline{79.26}_{\pm1.9}$ | (↑) | $\mathbf{74.05}_{\pm2.4}$ | | $52.33_{\pm0.9}$ | | $\mathbf{78.16}_{\pm1.3}$ | |
| $\mathrm{att}(\Phi_{\mathrm{ind}}(\mathbf{N},\cdot),\Phi_{\mathrm{val}}(\mathbf{L},\cdot))$ | $83.27_{\pm0.3}$ | | $\mathbf{66.08}_{\pm2.1}$ | | $64.46_{\pm1.2}$ | | $80.31_{\pm1.4}$ | | $79.14_{\pm1.1}$ | (↑) | $70.84_{\pm2.3}$ | | $52.33_{\pm0.9}$ | | $74.54_{\pm1.9}$ | (↑) |
| $\mathrm{att}(\Phi_{\mathrm{ind}}(\mathbf{L},\cdot),\Phi_{\mathrm{val}}(\mathbf{N},\cdot))$ | $82.53_{\pm1.9}$ | (↑) | $\underline{65.73}_{\pm2.4}$ | | $62.31_{\pm1.4}$ | | $79.54_{\pm2.1}$ | (↑) | $\mathbf{80.51}_{\pm1.2}$ | | $74.05_{\pm2.4}$ | | $51.37_{\pm0.5}$ | (↑) | $\mathbf{78.16}_{\pm1.3}$ | |

## F.7 Node Classification Using Diag-NLSFs and Lead-NLSFs

In App. C, we presented the diag-NLSFs, considering $\widetilde{d} = d$, and lead-NLSFs for leading filters that do not include orthogonal complements. We explore the leading filters, orthogonal complement, and diagonal operation in our NLSFs. The results of these investigations are summarized in Tab. 15 , which shows the node classification accuracy achieved using various configurations, including NLSFs, diag-NLSFs, lead-NLSFs, and lead-diag-NLSFs. We see that incorporating both the orthogonal complement and the multi-channel approach yields the highest classification accuracy.

## F.8 Uniform Sub-Bands

Our primary objective in this work is to introduce new GNNs that are equivariant to functional symmetries, based on a novel spectral domain transferable between graphs using analysis and synthesis. Our NLSFs in Sec. 3.4 consider filters $g_j$ that are supported on the dyadic sub-bands $\left[\lambda_N r^{S-j+1}, \lambda_N r^{S-j}\right]$. The closer to the low-frequency range, the denser the sub-bands become. We illustrate an example of filters $g_j$ supported on dyadic sub-bands with $r = \frac{1}{2}$ and $S = 4$ in Fig. 5, showing that sub-bands become denser as they approach the low-frequency range. Our primary goal is to highlight the unique aspects of our method rather than sub-band separation, which, while crucial, is an important consideration across spectral GNNs. Therefore, we also present uniform sub-bands, where filters $g_j$ are supported on the uniform sub-bands $\left[(j-1)\frac{\lambda_N}{S}, j\frac{\lambda_N}{S}\right]$. Note that the modifications required for our NLSFs are minimal, and most steps can be seamlessly applied with filters supported on the uniform sub-bands.

We evaluate our NLSFs with uniform sub-bands on graph classification tasks. We consider the graph benchmarks as in Sec. 5, including five bioinformatics datasets: MUTAG, PTC, NCI1, ENZYMES, and PROTEINS, and three social network datasets: IMDB-B, IMDB-M, and COLLAB. Note the sub-bands only affect our Value-NLSFs, where Index-NLSFs remain the same as in Sec. 3.4.

We report the graph classification accuracy of Graph-NLSFs and Pooling-NLSFs using uniform sub-bands in Tab. 16 and Tab. 17, respectively, where the $\Phi_{\mathrm{ind}}(\mathbf{L},\cdot)$, $\Phi_{\mathrm{ind}}(\mathbf{N},\cdot)$, $\Theta_{\mathrm{ind}}(\mathbf{L},\cdot)$, and $\Theta_{\mathrm{ind}}(\mathbf{N},\cdot)$ are the index-based NLSFs and therefore they are the same as in Tab. 10 and Tab. 11. Interestingly, the Laplacian attention does not yield significant improvements for either Graph-level NLSFs or Pooling-NLSFs when considering uniform sub-bands. Moreover, we observe that the graph classification performance is generally worse than when using the dyadic sub-bands reported in Tab. 10 and Tab. 11. We emphasize the importance of considering the spectral support of filters $g_j$. Empirically, we found that using the dyadic grid is more effective, which is why we focus on it and report those results in the main paper. However, exploring other sub-bands remains an important task for future work. For instance, we plan to investigate sub-bands based on $\{ar^j - b\}_{j=1}^K$ for other choices of $1 < r < 2$ and $b > 0$. In the limit when $r \to 1$, $a \to \infty$ and $b \to \infty$ this "converges" to the uniform grid.

Table 17: Graph classification performance using Pooling-NLSFs with uniform sub-bands. The symbol (↑) denotes an improvement using Laplacian attention.

| | MUTAG | PTC | ENZYMES | PROTEINS | NCI1 | IMDB-B | IMDB-M | COLLAB |
|---|---|---|---|---|---|---|---|---|
| $\Theta_{ind}^p(\mathbf{L},\cdot)$ | $\mathbf{86.41}_{\pm1.9}$ | $\mathbf{68.76}_{\pm0.9}$ | $\mathbf{69.88}_{\pm1.3}$ | $\mathbf{84.27}_{\pm1.3}$ | $80.33_{\pm0.8}$ | $60.40_{\pm1.3}$ | $51.01_{\pm1.3}$ | $71.28_{\pm0.9}$ |
| $\Theta_{ind}^p(\mathbf{N},\cdot)$ | $84.52_{\pm0.8}$ | $67.19_{\pm1.2}$ | $66.37_{\pm2.1}$ | $81.12_{\pm1.7}$ | $77.05_{\pm2.2}$ | $62.08_{\pm1.6}$ | $52.48_{\pm1.0}$ | $72.03_{\pm1.2}$ |
| $\Theta_{val}^p(\mathbf{L},\cdot)$ | $83.49_{\pm1.1}$ | $67.12_{\pm1.0}$ | $65.38_{\pm1.7}$ | $82.89_{\pm2.2}$ | $\underline{80.57}_{\pm2.4}$ | $\mathbf{72.85}_{\pm1.4}$ | $50.91_{\pm1.4}$ | $75.02_{\pm1.2}$ |
| $\Theta_{val}^p(\mathbf{N},\cdot)$ | $83.34_{\pm1.3}$ | $64.22_{\pm0.9}$ | $62.24_{\pm0.7}$ | $80.03_{\pm0.4}$ | $80.90_{\pm1.6}$ | $72.49_{\pm1.1}$ | $52.08_{\pm0.9}$ | $\underline{80.89}_{\pm2.2}$ |
| $att^p(\Theta_{ind}(\mathbf{L},\cdot),\Theta_{val}(\mathbf{L},\cdot))$ | $\mathbf{86.41}_{\pm1.9}$ | $\mathbf{68.76}_{\pm0.9}$ | $\mathbf{69.88}_{\pm1.3}$ | $\mathbf{84.27}_{\pm1.3}$ | $80.33_{\pm0.8}$ | $\mathbf{72.85}_{\pm1.4}$ | $51.01_{\pm1.3}$ | $76.13_{\pm0.7}$ (↑) |
| $att^p(\Theta_{ind}(\mathbf{N},\cdot),\Theta_{val}(\mathbf{N},\cdot))$ | $84.52_{\pm0.8}$ | $67.19_{\pm1.2}$ | $\underline{67.13}_{\pm2.2}$ (↑) | $82.53_{\pm1.1}$ (↑) | $\mathbf{80.90}_{\pm1.6}$ | $72.49_{\pm1.1}$ | $\underline{53.16}_{\pm0.8}$ (↑) | $80.89_{\pm2.2}$ |
| $att^p(\Theta_{ind}(\mathbf{N},\cdot),\Theta_{val}(\mathbf{L},\cdot))$ | $\underline{84.52}_{\pm0.8}$ | $\underline{68.01}_{\pm1.4}$ (↑) | $66.44_{\pm0.9}$ (↑) | $\underline{83.52}_{\pm2.4}$ (↑) | $80.57_{\pm2.4}$ | $\mathbf{72.85}_{\pm1.4}$ | $52.48_{\pm1.0}$ | $77.42_{\pm0.7}$ (↑) |
| $att^p(\Theta_{ind}(\mathbf{L},\cdot),\Theta_{val}(\mathbf{N},\cdot))$ | $\mathbf{86.41}_{\pm1.9}$ | $\mathbf{68.76}_{\pm0.9}$ | $\mathbf{69.88}_{\pm1.3}$ | $\mathbf{84.27}_{\pm1.3}$ | $80.90_{\pm1.6}$ | $72.49_{\pm1.1}$ | $\mathbf{53.48}_{\pm1.2}$ (↑) | $\mathbf{81.12}_{\pm1.4}$ (↑) |

# G   Additional Related Equivariant GNNs

Equivariant GNNs are designed to handle graph data with symmetries. The output of an equivariant GNN respects the same transformation applied to the input. Depending on the application, the transformation could involve translations, reflections, or permutations, to name but a few [82, 46, 78]. Due to respecting the symmetries, equivariant GNNs can reduce model complexity and improve generalization [35, 27], which can be applied to test data and produce more interpretable representations [99, 51]. When symmetry plays a critical role, such as in physical simulations, molecular modeling, and protein folding, equivariant GNNs have been demonstrated to be effective [20]. For example, [91, 33] employ spherical harmonics to handle 3D molecular structures, and [82] simplify computations for explicit rotation and translation matrices by focusing on relative distances between nodes. In addition, [30] embeds symmetry transformations by parameterizing convolution operations over Lie groups, such as rotations, translations, and scalings, through Lie algebra.

