# OpenReview forum: "Equivariant Machine Learning on Graphs with Nonlinear Spectral Filters"
_NeurIPS.cc/2024/Conference — NeurIPS 2024 poster_

### Official Review · Reviewer_6hk5 · 2024-07-04

**Soundness:** 3
**Presentation:** 3
**Contribution:** 4
**Rating:** 7
**Confidence:** 3

**Summary:**

This paper addresses the issue of inadequate modeling of graph equivariance in existing spectral GNNs due to nonlinear operations. The authors investigate the concept of domain translation in graph space as functional translations, drawing from the convolutional operations defined on images. Based on a series of in-depth analyses, they propose nonlinear spectral filters (NLSFs) that are fully equivariant to graph functional shifts and demonstrate universal approximation capabilities.

**Strengths:**

1. The research problem is highly valuable, and the ideas presented are novel.
2. The theoretical analysis is rigorous, thoroughly supporting the arguments and solutions proposed in the paper. It reflects the authors' deep understanding and insight in the field.
2. Notable experimental improvements in classification tasks.

**Weaknesses:**

1. The paper is missing important spectral GNN models such as [1,2,3,4].
2. There is a lack of discussion on equivariant GNNs, such as [5,6,7,8,9,10,11,12]. While the focus is on spectral GNNs, it is also essential to discuss works in the spatial GNN domain, especially given that the paper’s title starts with "Equivariant Machine Learning on Graphs." Positioning your work within the broader GNN field can further elucidate the significance and unique advantages of your contributions.
3. The experiments are somewhat lacking in comprehensiveness. While you have demonstrated the effectiveness of your proposed model in common node and graph classification tasks, your core contribution is enhancing equivariant representation learning in spectral GNNs. I suggest adding experiments that specifically show how the improved performance is due to enhanced equivariant learning, such as graph isomorphism tests.

[1] How powerful are spectral graph neural networks

[2] Bernnet: Learning arbitrary graph spectral filters via bernstein approximation

[3] Specformer: Spectral graph neural networks meet transformers

[4] Graph neural networks with learnable and optimal polynomial bases

[5] Universal Invariant and Equivariant Graph Neural Networks

[6] E(n) Equivariant Graph Neural Networks

[7] On the Generalization of Equivariant Graph Neural Networks

[8] Expressive Power of Invariant and Equivariant Graph Neural Networks

[9] Approximately Equivariant Graph Networks

[10] Equivariant Polynomials for Graph Neural Networks

[11] Graph Neural Networks for Learning Equivariant Representations of Neural Networks

[12] Sign and Basis Invariant Networks for Spectral Graph Representation Learning

**Questions:**

see weaknesses

**Limitations:**

yes, the authors discuss the limitations in Section 6

---

> ### Author Rebuttal · Authors · 2024-08-06
>
> We thank the reviewer for the careful and constructive review.
>
> >**Additional Comparison with Spectral GNNs:**
>
> Thank you for the comment.
>
> Following the reviewer's request, we included JacobiConv, BernNet, Specformer, and OptBasisGNN in our node classification experiment.
>
> Moreover, motivated by other comments about heterophilic graphs, we extended NLSFs to include an additional filter capturing the band that contains the remaining non-leading eigenvectors, which we call rem-NLSFs. For Index NLSFs, we incorporate $\mathbf{I} - \sum_ {j=1}^J \mathbf{P}_ j$, and for Value NLSFs, we add the $\mathbf{I} - \sum_ {j=1}^K g_ j(\mathbf{\Delta})$. This alleviates the loss of information due to projecting to the low-frequency bands. All of our proofs are trivially extended to rem-NLSFs. If the reviewer would like to see the new proofs, which are almost identical to the old ones, we can send an anonymous PDF file to the AC according to the regulations of NeurIPS.
>
> Table 2 of the global response PDF presents the node classification accuracy for these methods and rem-NLSFs. We see that rem-NLSFs outperform the other models on the Cora, Citeseer, and Chameleon datasets. For the Pubmed and Actor datasets, rem-NLSFs achieve the second-best results. Adding the orthogonal component alleviates the loss of information due to projecting to the low-frequency bands and, therefore, further improves the empirical results. The new experiments will be included in the camera-ready version if accepted.
>
> >**Additional Discussion on Equivariant GNNs:**
>
> Thank you for this feedback.
>
> While Section 2 already shortly discusses the background of Equivariant GNNs, we add an extended review in the appendix to include more works:
>
> In recent developments on equivariant GNN model design, Satorras et al. (2021) introduced a new model for learning graph neural networks (GNNs) that are equivariant to rotations, translations, reflections, and permutations, Huang et al. (2023) developed an approach for designing GNNs that respect approximate automorphism symmetries. Additionally, Lim et al. (2023) presented SignNet and BasisNet, which process eigenvectors invariant to sign and eigenspaces. Different from these methods, Kofinas et al. (2024) proposed representing neural networks as computational graphs of parameters. In a more theoretically oriented work, Keriven et al. (2019) extended the Stone-Weierstrass theorem for the algebra of real-valued functions to the equivariant case, and Karczewski et al. (2024) established a generalization bound for E(n)-Equivariant GNNs.  Azizian et al. (2021) explored the expressive power of message-passing GNNs, linear GNNs, and folklore GNNs, for both their invariant and equivariant versions. Additionally, Puny et al. (2023) introduced an alternative expressive power hierarchy to evaluate GNNs based on their ability to approximate equivariant graph polynomials.
>
> We will also include a more detailed comparison to methods that consider graph automorphisms as symmetries, like Huang et al. (2023). Let us explain the main idea of the comparison. The goal is to illustrate that functional translations are more stable than hard symmetries of the graph, namely, graph automorphisms (isomorphisms from the graph to itself). Automorphisms are another analog to translations on general graphs, which competes with our notion of functional shifts. Consider as an example the standard 2D grid on the torus. The automorphisms of the grid include all translations. However, this notion of hard symmetry is very sensitive to graph perturbations. If one adds or deletes even one edge in the graph, the automorphism group becomes much smaller and does not contain the translations anymore. On the other hand, we claim that functional shifts are not so sensitive. To visualize this, we conducted a toy experiment that illustrates that graph functional shifts are stable to graph perturbations. Specifically, we show that standard translations can be approximated by a graph functional shift on a perturbed grid (adding Gaussian noise to the edge weights), optimizing the coefficients of the functional shift so the operator is as close as possible to the classical shift. See Figure 2 in the PDF. We then train the NLSFs on a perturbed grid to do MNIST classification, following the setup in [1,2,3]. We compare the NLSF on the clean grid to the NLSF on the perturbed grid. Table 3 presents the performance of the NLSF, which is almost unaffected by the perturbation, which demonstrates that NLSFs on the perturbed grid can roughly implement CNN-like architectures (translation equivariant functions).
>
> [1] Defferrard et al. (2016). Convolutional neural networks on graphs with fast localized spectral filtering.
>
> [2] Monti et al. (2017). Geometric deep learning on graphs and manifolds using mixture model CNNs.
>
> [3] Levie et al. (2018). Cayleynets: Graph convolutional neural networks with complex rational spectral filters.
>
> >**Experiments Highlighting the Benefits of Equivariant Learning:**
>
> Thanks for this comment.
>
> It is important to note that the connection between graph isomorphism testing and GNNs is primarily theoretical. In practical applications, spectral GNNs are not commonly utilized for direct isomorphism testing to the best of our knowledge. We would appreciate it if the reviewer could provide any specific work they are referring to for further clarification.
>
> Following the reviewer's suggestion, as explained above, we added a new toy example demonstrating how a standard translation can be estimated by a graph functional shift on a perturbed grid, which shows the robustness of functional shift symmetries. We then train an NLSF MNIST classifier, which illustrates that NLSFs can roughly implement CNN-like architectures, also on a perturbed grid.
>
> Thank you again for your valuable feedback. We will include these examples in the camera-ready version if accepted.

---

> > ### Comment · Reviewer_6hk5 · 2024-08-12
> >
> > Thanks for the authors' detailed rebuttal. As all of my concerns have been addressed, I have raised my score.

---

> > > ### Author Response · Authors · 2024-08-13
> > >
> > > We warmly thank the reviewer for their comments and for acknowledging our rebuttal.

---

### Official Review · Reviewer_GXWH · 2024-07-12

**Soundness:** 3
**Presentation:** 3
**Contribution:** 3
**Rating:** 7
**Confidence:** 3

**Summary:**

This paper proposes a spectral GNN called non-linear spectral filters (NLSF), which aims to enhance GNNs with nonlinear functions. Since general GNNs with nonlinear functions do not commute with unitary operators, this paper defines Graph Functional Shifts, which is a set of unitary matrices commuting with a normal graph shift operator (GSO). It then formulates two functions for spectral index and filter bank, respectively, and concatenates these two functions as graph attention. In the experiment section, NLSF is compared with GAT, SAGE, and other spectral-like GNNs. In the node-classification task, att-Node-level NLSF shows outstanding performance among these models. In the graph classification task, att-Graph-level NLSF achieves comparable results with these models, and att-Pooling-NLSF performs better than other models in the graph classification task.

**Strengths:**

1. NLSFs have a solid mathematical foundation and proof, especially on Universal Approximation and graph expressivity.
2. The experimental results validate the effectiveness of the theory.

**Weaknesses:**

1. The method proposed in this paper requires the use of eigenvalues, hence it necessitates eigen decomposition of the GSO. The time complexity of eigendecomposition is relatively high, especially for very large graphs.

**Questions:**

None

---

> ### Author Rebuttal · Authors · 2024-08-06
>
> Thank you for your comment.
>
> Please note that in Section 4.1, we discussed the complexity of our method. We elaborated on the efficiency of the Lanczos algorithm, which is well-known for its computational efficiency. For estimating the leading $J$ eigenvectors, the Lanczos algorithm takes $O(JE)$ operations per iteration and converges very fast, where $E$ is the number of edges. Hence, the Lanczos algorithm is as efficient as a message-passing neural network that operates on features of dimension $J$.
>
> Section 4.1 already discusses the efficiency of the Lanczos algorithm, but we will extend the text and make this clearer. We will clarify that the Lanczos algorithm (for sparse graphs) can be seen as a message-passing algorithm, as it only uses the nonzero entries of the matrix, which are interpreted as edges in our context of graph machine learning.
>
> Additionally, we reported the runtime analysis in Appendix C, where the average running time per epoch(ms)/average total running time(s) of our method is 18.22/4.5 for Cora, 18.51/4.4 for Citeseer, 20.23/6.1 for Pubmed, 28.51/17.1 for Chameleon, 25.56/5.1 for Squirrel, and 17.09/4.6 for Actor. In comparison, for example, the runtime for ChebNetII is 20.53/5.9 for Cora 20.61/5.7, for Citeseer, 33.57/12.9 for Pubmed, 39.03/17.3 for Chameleon, 37.29/38.04 for Squirrel, and 40.05/9.1 for Actor. Our method is efficient and comparable with competing baselines.
>
> We appreciate the reviewer's feedback and will make sure that these points are more explicitly detailed in the camera-ready versions if accepted.

---

### Official Review · Reviewer_J5Wx · 2024-07-14

**Soundness:** 4
**Presentation:** 3
**Contribution:** 3
**Rating:** 7
**Confidence:** 3

**Summary:**

The authors introduce spectral GNNs that are equivariant to functional symmetries. Specifically, they introduce node-level, graph-level and pooling non-linear spectral filters and show that these are able to outperform standard convolutional GNNs on (semi-supervised) node classification and graph classification tasks.

**Strengths:**

- The experimental results are compelling.
- To the best of my understanding, the theory is sound
- The idea being proposed is novel and worth being investigated
- The paper is clearly written, even though it doesn't seem to be very accessible to readers unfamiliar with graphs signals processing

**Weaknesses:**

- While the authors did a good job in trying to introduce all the relevant concepts, the paper is quite dense with mathematical details and notions that will likely be unfamiliar to many GNN researchers and may therefore hinder the accessibility of the manuscript.

**Questions:**

- While it's very intuitive to understand what is meant by "shift" in the context of images and CNNs, this doesn't come across very clear in the paper in the context of graphs: what is the rationale behind the decision to "model the group of translations on graphs as the group of all unitary operators on signals that commute with the graph shift operator"? If more space in the paper can be used to make the underlying concepts more accessible (perhaps moving some of the material on the theoretical properties of the NLSFs to the appendix) I think the paper would greatly gain in accessibility, potentially increasing its impact beyond the graph signal processing community.

**Limitations:**

The limitations are clearly discussed.

---

> ### Author Rebuttal · Authors · 2024-08-06
>
> We thank the reviewer for the positive and encouraging comments.
>
> >**Enhancing Accessibility of Theoretical Contributions:**
>
> Thank you for your comment.
>
> We recognize the importance of making our manuscript accessible to a broader audience. To improve accessibility, we will provide more details on the mathematical concepts in the appendix. For instance, we will write and explain Schur’s lemma, extend the discussion on random geometric graphs, and elaborate on the background of the cut norm and Hoeffding's inequality.
>
> >**Clarifying the Concept of Functional Translation on Graphs for Enhanced Accessibility:**
>
> We will extend the text in Section 3.1 to further clarify the link between graph functional shifts and standard domain translations. We will start by showing that on the standard grid (on the 2D torus) with central difference GSO, the standard Fourier basis is the eigenbasis of the Laplacian. We will then show that any domain shift on the grid, when applied on signals, is equivalent to modulation in the frequency domain. Hence, domain shifts commute with the Laplacian (both of them are diagonal matrices in the frequency domain). Functional shifts are defined by adopting this property as a definition. We call *any* unitary operator that commutes with Laplacian a graph functional shift. For the grid, these include exactly the operators that multiply each frequency by a complex number of unit modulus.
>
> Fig. 1  (in the global response PDF) presents a comparison of a classical and a functional translation of a Gaussian signal. The classical translation $f(t-T)$ is a specific case of functional translation, which is equivalent to modulation in the frequency domain $e^{-i\omega T}\hat{f}(\omega)$. Our example of a graph functional shift is an operator that modulates low frequencies by a given rate and modulates high frequencies by another rate. This is interpreted as shifting the low-frequency content and the high-frequency content of the signal at different speeds. This illustrates that functional symmetries are more rich than domain symmetries.
>
> In addition, we add another experiment showing that functional translations are much more stable than hard symmetries of the graph, namely, graph automorphisms (isomorphisms from the graph to itself). Autonorpisms are another analogue to translations on general graphs, which competes with our notion of functional shifts. Consider again the standard 2D grid on the torus. The automorphisms of the grid include all translations. However, this notion of hard symmetry is very sensitive to graph perturbations. If one adds or deletes even one edge in the graph, the automorphism group becomes much smaller and does not contain the translations anymore. On the other hand, we propose a toy experiment that illustrates the graph functional shifts are stable to graph perturbations. Specifically, we show that standard translations can be approximated by a graph functional shift on a perturbed grid (adding Gaussian noise to the edge weights), optimizing the coefficients of the functional shift so the operator is as close as possible to the classical shift. See Figure 2 in the PDF. We then train the NLSFs on a perturbed grid to do MNIST classification, following the setup in [1,2,3]. We compare the NLSF on the clean grid to the NLSF on the perturbed grid. Table 3 shows that the performance of the NLSF is almost unaffected by the perturbation, which demonstrates that NLSFs on the perturbed grid can roughly implement CNN-like architectures (translation equivariant functions).
>
> These examples will be incorporated into the camera-ready version if accepted, including the discussion about automorphism symmetries vs functional shifts.
>
> [1] Defferrard et al. (2016). Convolutional neural networks on graphs with fast localized spectral filtering.
>
> [2] Monti et al. (2017). Geometric deep learning on graphs and manifolds using mixture model CNNs.
>
> [3] Levie et al. (2018). Cayleynets: Graph convolutional neural networks with complex rational spectral filters.

---

> > ### Comment · Reviewer_J5Wx · 2024-08-13
> >
> > I appreciate the clarifications and support the proposed updates in the camera ready version of the paper.

---

> > > ### Author Response · Authors · 2024-08-13
> > >
> > > We warmly thank the reviewer for their comments and for acknowledging our rebuttal.

---

### Official Review · Reviewer_fGo4 · 2024-07-15

**Soundness:** 3
**Presentation:** 3
**Contribution:** 2
**Rating:** 5
**Confidence:** 5

**Summary:**

The authors propose nonlinear spectral filters (NLSFs) that achieve full equivariance to graph functional shifts, demonstrating that these filters have universal approximation properties. These NLSFs are designed based on transferable spectral domain, potentially improving GNN performance in node and graph classification tasks across diverse graph structures.

**Strengths:**

1- The paper is well-written and self-contained, offering clear, didactic insights. The experiments provide valuable conclusions that future practitioners will find useful. However, a synthesis of the information could further enhance readability and understanding for the reader.

2- The use of the nonlinear spectral filters for graphs to achieve full equivariance to graph functional shifts may be a promising avenue to explore.

**Weaknesses:**

Despite these merits, I have the following concerns about the paper.

1- While the paper presents a compelling method with potential applications in graph analysis, one significant limitation is its scalability, particularly concerning large-scale graphs. The reliance on specific spectral properties, such as the leading eigenvectors, may not only limit the method's capacity to capture diverse graph dynamics but also result in computational inefficiencies when applied to extensive graph datasets.

2- The datasets used in the paper predominantly consist of mid-sized, homophilic graphs, which may not fully represent the diverse range of real-world applications, particularly in contexts involving heterophilic graphs.

3- The efficiency of the proposed models in terms of computation and resource utilization is not adequately discussed.

**Questions:**

(i) Does your theory adapt differently when applied to heterophilous graphs compared to homophilous graphs, and if so, how are these differences addressed within your methodology?

(ii) Given that your Nonlinear Spectral Filters (NLSFs) are motivated by respecting graph functional shift symmetries, similar to Euclidean CNNs, do you have any claims or observations regarding how NLSFs fit within or potentially extend the Weisfeiler-Lehman hierarchy of expressivity? Additionally, could you elaborate on how the expressivity of NLSFs, as informed by metrics from the Euclidean vector space, compares to traditional graph neural network models?

**Limitations:**

Also, the efficiency of the proposed models in terms of computation and resource utilization is not adequately addressed. For practical applications, especially in resource-constrained environments, understanding the computational overhead is essential.

---

> ### Author Rebuttal · Authors · 2024-08-06
>
> We thank the reviewer for the insightful comments and questions.
>
> >**Complexity, Efficiency, and Runtime Analysis:**
>
> In Section 4.1, we discussed our method's complexity and efficiency, highlighting the efficiency of the Lanczos algorithm. This algorithm estimates the leading $J$ eigenvectors in $O(JE)$ operations per iteration and converges quickly, making it as efficient as a message-passing neural network with $J$-dimensional features.
>
> We will extend Section 4.1 to clarify that the Lanczos algorithm can be viewed as a message-passing algorithm for sparse graphs, using only the matrix's nonzero entries, interpreted as edges in graph machine learning. In addition, we reported the runtime analysis in Appendix C, where the average running time per epoch(ms)/average total running time(s) of our method is 18.22/4.5 for Cora, 18.51/4.4 for Citeseer, 20.23/6.1 for Pubmed, 28.51/17.1 for Chameleon,  25.56/5.1 for Squirrel, and 17.09/4.6 for Actor. In comparison, for example, the runtime for ChebNetII is 20.53/5.9 for Cora 20.61/5.7, for Citeseer, 33.57/12.9 for Pubmed, 39.03/17.3 for Chameleon, 37.29/38.04 for Squirrel, and 40.05/9.1 for Actor. Our method is efficient and comparable with competing baselines. Furthermore, we will include the space overhead (MB) of our method: 79 for Cora, 163 for Citeseer, 2054 for Pubmed, 85 for Chameleon, 263 for Squirrel, and 319 for Actor. In comparison, the space overhead for ChebNetII is 67 for Cora, 159 for Citeseer, 1821 for Pubmed, 79 for Chameleon, 259 for Squirrel, and 316 for Actor.
>
> We also explain below how we addressed the limitation of computing only the leading eigenvectors. We slightly extended the NLSF method (the extension is called rem-NLSF), which now allows capturing all of the frequency contant of the graph. See the details below.
>
> >**Theoretical Analysis on Heterophilic Graphs and Large-Scale Datasets:**
>
> Thank you for your comment.
>
> We would like to clarify that in our first submission, we used three heterophilic graphs (Chameleon, Squirrel, and Actor) among the six datasets for the node classification task. We will emphasize this point more clearly in our revised version.
>
> Motivated by the reviewer's question, we extend our approach to include an additional filter in both Index and Value NLSFs. For Index NLSFs, we incorporate the $(J+1)$-th filter as $\mathbf{I} - \sum_ {j=1}^J \mathbf{P}_ j$, and for Value NLSFs, we add the $(K+1)$-th filter $\mathbf{I} - \sum_ {j=1}^K g_ j(\mathbf{\Delta})$. This alleviates the loss of information due to projecting to the low-frequency bands. Now, the full spectral range of the signal is captured by the NLSF.
>
> We denote these NLSFs with orthogonal complement as **rem-NLSFs**. This setup is particularly relevant for heterophilous graphs, which require high-frequency components to accurately represent the labels. Table 2 presents the classification performance, and we see that rem-NLSFs improve performance for heterophilic graphs. We will motivate this construction in our paper from the perspective of heterogeneous components of the target.
>
> The theory, and all of our proofs, trivially extend to rem-NLSF. If the reviewer would like to see the new version of the proofs, which are almost identical to the old proofs, we would be more than happy to send an anonymous PDF file to the AC according to NeurIPS regulations.
>
> In addition, following the reviewer's questions, we conduct additional tests on five large heterophilic graphs: Penn94, Pokec, Genius, Twitch-Gamers, and Wiki datasets from Lim et al. (2021). The experimental setup is in line with previous work by Chien et al. (2021), Lim et al. (2021), and He et al. (2022). We use the same setup for our NLSFs as reported in Appendix B. Table 1 (in the global response PDF) presents the classification accuracy. We see that rem-NLSFs outperform the competing methods on the  Penn94, Pokec, Genius, and Wiki datasets. For the Twitch-Gamers dataset, rem-NLSFs yield the second-best results. Our additional experiments show that our method could indeed scale to handle large-scale graphs effectively.
>
> Thank you again for the valuable comment.
>
> >**Expressivity:**
>
> We note that graph-level NLSFs are bounded by the expressive power of MPNNs with random positional encodings. For example, in [1], the authors showed that random features improve GNN expressivity, distinguishing certain structures that deterministic GNNs cannot. The Lanczos algorithm for computing the leading $J$ eigenvectors can be viewed as a message-passing algorithm with a randomly initialized $J$-channel signal.
>
> In Section 4.3, we discuss the expressivity in the context of a single fixed graph with variable signals. This setting is unrelated to the traditional graph isomorphism test hierarchy of expressivity.
>
> At the graph-level, our example in Appendix A.2.4 can be used to show that graph-level NLSFs (without synthesis and pooling) are not more expressive than spectral GNNs. We also have an approach for showing that spectral GNNs are more expressive than graph level NLSF (without synthesis and pooling), but it requires very high dimensional features and unstable filters. If the reviewer would like to see the analysis, we can write you a comment in OpenReview or send an anonymous PDF to the AC according to the NeurIPS regulations.
>
> Regarding pooling-NLSFs, we do not have an answer to whether standard spectral GNNs are more powerful than pooling-NLSFs with unlimited resources. A more practically useful question is comparing their expressivity within the same parameter budget. This is a theoretically challenging question. However, in practice, NLSFs typically perform better than standard spectral GNNs with equal budgets, or equivalent architecture. This comparison is also valid for graph-level NLSF (without synthesis and pooling).
>
> We will add a discussion on these points in the camera-ready version if accepted.
>
> [1] Sato et al. (2021). Random features strengthen graph neural networks.

---

> > ### Comment · Reviewer_fGo4 · 2024-08-12
> >
> > Thank the authors for their extensive rebuttal which has answered some of my concerns. After reading the rebuttal and other reviewers' comments, I decided to keep my current rating.

---

> > > ### Author Response · Authors · 2024-08-13
> > >
> > > We thank the reviewer for their comments and for acknowledging our rebuttal. We would be happy to address any remaining concerns if the reviewer has any.

---

### Official Review · Reviewer_8cKX · 2024-07-16

**Soundness:** 3
**Presentation:** 2
**Contribution:** 3
**Rating:** 7
**Confidence:** 2

**Summary:**

The paper tackles the task of Network design for graph neural networks. The suggested approach is based on spectral properties of graphs. So far in the literature spectral methods were limited in assuming that the graph domain is fixed. To address this, a relaxed version of symmetry is proposed based on band-limited projections. In addition, a nonlinear spectral filter design is suggested, suggesting node-level, graph-level, and pooling operations. The method is evaluated on several graph learning tasks, demonstrating improvement in generalization over existing spectral methods.

**Strengths:**

The paper makes a valuable contribution to the literature on Graph Neural Networks (GNNs), particularly by addressing the challenge of transferability in spectral methods, which is highlighted as a significant issue.

claims are supported by theoretical analysis.

The paper is self-contained, providing both background information and a short overview on spectral graph learning.

**Weaknesses:**

Writing Quality: Some sections of the manuscript could benefit from revision. For instance, reordering the paragraphs in the introduction could improve readability. Specifically, mentioning what was missing from previous works earlier rather than at the end would help.

More examples can be found in the method section: i) The discussion on problem the activation functions is missing some details, e.g., what rho is exactly? ii) The paper states that "It is important to note that functional shifts, in general, are not induced from node permutations. In stead, functional shifts are related to the notion of functional maps...". This sentence is too vague. Consider adding more details to make it clearer.

No qualitative results are provided. Is it possible to visualize learned features as in the illustration in figure 2? Is it possible to design a toy experiment showcasing the suggested notion of relaxed symmetry, for which the suggested network design generalizes adequately?

**Questions:**

No question other than the weakeness stated above.

**Limitations:**

Yes

---

> ### Author Rebuttal · Authors · 2024-08-06
>
> We thank the reviewer for the positive feedback.
>
> >**Enhancing Writing Quality for Improved Readability and Clarity:**
>
> Thank you for your valuable feedback.
>
> In response to the reviewer’s suggestions, we will make the following revisions:
> - Introduction Section: We will reorder the paragraphs in the introduction. Specifically, we will highlight the research gap earlier in the section.
> - Method Section:
>     - We will emphasize that $\rho$ is any non-linear activation function (such as ReLU, Sigmoid, etc.). We will include the following explanation of why non-linear activation functions break the functional symmetry. We offer a construction for complex-valued signals, which is easy to extend to the real case. Consider a 1D grid with $100$ nodes and the central difference Laplacian. Here, the Laplacian eigenvectors are the standard Fourier modes. In this case, one can see that a graph functional shift is any operator that is diagonal in the frequency domain and multiplies each frequency by any complex number with the unit norm. Consider the nonlinearity that takes the real part of the signal and then applies ReLU, which we denote in short by ReLU. Consider a graph signal $x=(e^{i\pi 10n/100} + e^{i\pi 20n/100})_{n=0}^{99}$. We consider a graph functional shift $S$ that shifts frequencies 10 and 20 at a distance of 5, and every other frequency is not shifted. Namely, frequency 10 is multiplied by $e^{-i\pi 50/100}=-i$, frequency 20 by $e^{ -i\pi = -i\pi 100/100}=-1$, and every other freequency is multiplied by 1. Consider also the classical shift $D$ that translates the whole signal by $5$ uniformly. Since $x$ consists only of the frequencies 10 and 20, it is easy to see that $Sx=Dx$. Hence, $ReLU(Sx)=ReLU(Dx)=D(ReLU(x))$. Conversely, if we apply $ReLU(x)$ and only then shift, note that $ReLU(x)$ consists of many frequencies in addition to 10 and 20. For example, by nonnegativity of ReLU, $ReLU(x)$ has a nonzero DC component (zeroth frequency). Now, $S(ReLU(x))$ only translates the 10 and 20 frequencies, so we have $S(ReLU(x))\neq D(ReLU(x))=ReLU(S(x))$.
>     We will add this example to an appendix.
>     - We will revise the sentence, "Instead, functional shifts are related to the notion of functional maps..." into "Instead, functional shifts are general unitary operators that are not permutation matrices in general. The value of the functionally translated signal at a given node can be a *mixture* of the content of the original signal at many different nodes. For example, the functional shift can be a combination of shifts of different frequencies at different speeds."
>     - We will also explain in detail why spectral filters commute with graph functional shifts. This is easily seen from the fact that both spectral filters and functional shifts are diagonal operators in the graph Fourier domain.
>
> >**Visualizing Learned Features and Demonstrating Relaxed Symmetry:**
>
> Visualizing the learned filter directly is not feasible due to its high-dimensional nature: the filter is a general MLP, which makes it challenging to visualize in a meaningful way (as is typically the case in deep learning). However, to illustrate the concepts of relaxed symmetry and functional translation, we will add a new toy example. Similarly to the above construction, one example of a functional translation on the grid, which is not a classical translation, is an operator that translates different frequency bands at different speeds. In the example, we compare the translation of a Gaussian signal on a 2D grid by a classical uniform translation and by different speeds for different frequencies. See the experiments in the rebuttal PDF.
>
> In addition, we add another experiment showing that functional translations are much more stable than hard symmetries of the graph, namely, graph automorphisms (isomorphisms from the graph to itself). Autonorpisms are another analogue to translations on general graphs, which competes with our notion of functional shifts. Consider again the standard 2D grid on the torus. The automorphisms of the grid include all translations. However, this notion of hard symmetry is very sensitive to graph perturbations. If one adds or deletes even one edge in the graph, the automorphism group becomes much smaller and does not contain the translations anymore. On the other hand, we propose a toy experiment that illustrates the graph functional shifts are stable to graph perturbations. Specifically, we show that standard translations can be approximated by a graph functional shift on a perturbed grid (adding Gaussian noise to the edge weights), optimizing the coefficients of the functional shift so the operator is as close as possible to the classical shift. See Figure 2 in the PDF. We then train the NLSFs on a perturbed grid to do MNIST classification, following the setup in [1,2,3]. We compare the NLSF on the clean grid to the NLSF on the perturbed grid. Table 3 shows that the performance of the NLSF is almost unaffected by the perturbation, which demonstrates that NLSFs on the perturbed grid can roughly implement CNN-like architectures (translation equivariant functions).
>
> These examples will be incorporated into the camera-ready version if accepted, including the discussion about automorphism symmetries vs functional shifts.
>
> [1] Defferrard et al. (2016). Convolutional neural networks on graphs with fast localized spectral filtering.
>
> [2] Monti et al. (2017). Geometric deep learning on graphs and manifolds using mixture model CNNs.
>
> [3] Levie et al. (2018). Cayleynets: Graph convolutional neural networks with complex rational spectral filters.

---

> > ### Comment · Reviewer_8cKX · 2024-08-12
> > **reply to authors**
> >
> > I appreciate the author’s rebuttal and believe that their suggestions would greatly benefit the paper. I have no further concerns.

---

> > > ### Author Response · Authors · 2024-08-13
> > >
> > > We warmly thank the reviewer for their comments and for acknowledging our rebuttal.

---

### Author Rebuttal · Authors · 2024-08-06

# General Response to All the Reviewers

We thank the reviewers for their valuable input and criticism. We highlight the main revisions to the paper below.

>**Enhancing NLSFs with Orthogonal Complements:**

Our original method projected the signal's information to the leading (low) frequencies. This projection can lose important information, especially in the context of heterogeneous graphs, which require high frequencies to represent the target/label. To mitigate this limitation, we slightly extended NLSFs to include an additional high-pass filter, denoting the new method **rem-NLSFs**. For Index NLSFs we add $\mathbf{I} - \sum_ {j=1}^J \mathbf{P}_ j$, and for Value NLSFs we add $\mathbf{I} - \sum_ {j=1}^K g_ j(\mathbf{\Delta})$. Now, the full spectral range of the signal is captured by the NLSF.

Table 2 presents classification performance on benchmark graphs, where rem-NLSFs improve performance. We will motivate this construction in our paper from the perspective of heterogeneous graphs.

The theory, and all of our proofs, trivially extend to rem-NLSF. If the reviewers would like to see the new version of the proofs, which are almost identical to the old proofs, we would be happy to send an anonymous PDF file to the AC according to the NeurIPS regulations.

>**Illustrating Functional Translations:**

Some reviewers asked for an additional discussion and illustrations focused on helping the reader to better grasp the new notions of symmetry.

We will start by showing that on the standard grid (on the 2D torus) with central difference GSO, the standard Fourier basis is the eigenbasis of the Laplacian. We will then show that any domain shift on the grid, when applied on signals, is equivalent to modulation in the frequency domain. Hence, domain shifts commute with the Laplacian (both of them are diagonal matrices in the frequency domain). Functional shifts are defined by adopting this property as a definition. We call *any* unitary operator that commutes with Laplacian a graph functional shift. For the grid, these include exactly the operators that multiply each frequency by a complex number of unit modulus.

Fig. 1  (in the global response PDF) presents a comparison of a classical and a functional translation of a Gaussian signal. While classical translation shifts the whole signal uniformly, functional translations can shift different frequency bands at different speeds. This illustrates that functional symmetries are more rich than domain symmetries.

In addition, we add another experiment showing that functional translations are much more stable than hard symmetries of the graph, namely, graph automorphisms (isomorphisms from the graph to itself).  Autonorpisms are another analogue to translations on general graphs, which competes with our notion of functional shifts. The relevant experiments are in Fig.2 and Table 3 in the PDF. See the response to Reviewer 6hk5 for more details.

These examples will be incorporated into the camera-ready version if accepted.

>**Expressivity in the Context of WL:**

We will add a discussion about the relation between our notion of expressivity and the traditional WL hierarchy. See the rebuttal of Reviewer fGo4 for more details.

>**Efficiency of Eigendecomposition:**

Some reviewers were concerned about the efficiency of eigendecomposition. We would like to point out that computing the leading eigenvectors of sparse graphs is as efficient as message-passing networks. In Section 4.1, we discussed our method's complexity and efficiency. The Lanczos algorithm estimates the leading $J$ eigenvectors in $O(JE)$ operations per iteration and converges quickly, making it as efficient as a message-passing neural network with $J$-dimensional features.

>**Scalability to Large-Scale Datasets:**

To demonstrate the scalability of our method, we conduct additional tests on five large heterophilic graphs: Penn94, Pokec, Genius, Twitch-Gamers, and Wiki datasets from Lim et al. (2021). The experimental setup is in line with previous work by Chien et al. (2021), Lim et al. (2021), and He et al. (2022). We use the same setup for our NLSFs as reported in Appendix B. Table 1 (in the global response PDF) presents the classification accuracy. We see that rem-NLSFs outperform the competing methods on the  Penn94, Pokec, Genius, and Wiki datasets. For the Twitch-Gamers dataset, rem-NLSFs yield the second-best results. Our additional experiments show that our method could indeed scale to handle large-scale graphs effectively. We will include these experiments in the camera-ready version if accepted.

>**Additional Comparison with Spectral GNNs:**

In response to the feedback about missing important spectral GNNs, we included JacobiConv, BernNet, Specformer, and OptBasisGNN in our node classification experiment following the reviewer's request.

Table 2 of the global response PDF presents the node classification accuracy for these methods and rem-NLSFs. We see that rem-NLSFs outperform the other models on the Cora, Citeseer, and Chameleon datasets. For the Pubmed and Actor datasets, rem-NLSFs achieve the second-best results. Adding the orthogonal component alleviates the loss of information due to projecting to the low-frequency bands and, therefore, further improves the empirical results. The new experiments will be included in the camera-ready version if accepted.

---

### Decision · Program_Chairs · 2024-09-25

**Decision:**

Accept (poster)

**Comment:**

The paper introduces Nonlinear Spectrum Filters (NLSFs), a novel filter operation for Graph Neural Networks (GNNs) that is equivariant to functional symmetries. The authors propose node-level, graph-level, and pooling non-linear spectral filters, demonstrating that these filters outperform standard convolutional GNNs in both semi-supervised node classification and graph classification tasks. The paper is well-supported by rigorous theoretical analysis and extensive empirical validation across various benchmarks.

While there were minor concerns about the mathematical density of the paper, which may limit accessibility, the reviewers acknowledged that the authors effectively addressed the most relevant concerns. There was limited discussion among reviewers, with the primary concern from the "borderline accept" reviewer focusing on the scalability of the method. The authors addressed this by providing results on larger datasets.

Given the overall positive feedback from the reviewers, particularly regarding the paper’s novelty, theoretical rigor, and comprehensive experimental validation, I recommend accepting the paper.